# Blocking Dectin-1 prevents colorectal tumorigenesis by suppressing prostaglandin E2 production in myeloid-derived suppressor cells and enhancing IL-22 binding protein expression

Dectin-1 (gene *Clec7a*), a receptor for β-glucans, plays important roles in the host defense against fungi and immune homeostasis of the intestine. Although this molecule is also suggested to be involved in the regulation of tumorigenesis, the role in intestinal tumor development remains to be elucidated. In this study, we find that azoxymethane-dextran-sodium-sulfate-induced and $Apc^{Min-/-}$-induced intestinal tumorigenesis are suppressed in $Clec7a^{-/-}$ mice independently from commensal microbiota. Dectin-1 is preferentially expressed on myeloid-derived suppressor cells (MDSCs). In the $Clec7a^{-/-}$ mouse colon, the proportion of MDSCs and MDSC-derived prostaglandin $E_2$ ($PGE_2$) levels are reduced, while the expression of IL-22 binding protein (IL-22BP; gene *Il22ra2*) is upregulated. Dectin-1 signaling induces $PGE_2$-synthesizing enzymes and $PGE_2$ suppresses *Il22ra2* expression in vitro and in vivo. Administration of short chain β-glucan laminarin, an antagonist of Dectin-1, suppresses the development of mouse colorectal tumors. Furthermore, in patients with colorectal cancer (CRC), the expression of *CLEC7A* is also observed in MDSCs and correlated with the death rate and tumor severity. Dectin-1 signaling upregulates $PGE_2$-synthesizing enzyme expression and $PGE_2$ suppresses *IL22RA2* expression in human CRC-infiltrating cells. These observations indicate a role of the Dectin-1-$PGE_2$-IL-22BP axis in regulating intestinal tumorigenesis, suggesting Dectin-1 as a potential target for CRC therapy.

Colorectal cancer (CRC) is the third most common type of cancer in the world, with over 1 million new cases and 500,000 deaths globally every year[1]. Most CRC cases occur in old age and dietary habit is suggested to be a possible risk factor for CRC[2]. However, underlying mechanisms that regulate intestinal tumor development have not been elucidated completely.

The C-type lectin receptor (CLR) is a family molecules consisted of more than 100 proteins with one or more lectin-like carbohydrate-recognition domains, which recognize a broad repertoire of ligands in the extracellular domain and a signal transducing motif, such as immunoreceptor tyrosine-based activation motif (ITAM) and immunoreceptor tyrosine-based inhibitory motif (ITIM), in the intracellular

e-mail: tangc7@mail.sysu.edu.cn; iwakura@rs.tus.ac.jp

domain and regulate a diverse range of immunological functions[3]. Dectin-1 (gene symbol: *Clec7a*), one of CLRs preferentially expressed in myeloid-derived cells, is the receptor for β−1, 3-linked glucans (β-glucans)[4–6]. Since β-glucans are the major cell wall components of most fungi, Dectin-1 stands on the first line of the host defense against fungal infection[5,6]. Furthermore, recent studies have revealed that this receptor is also involved in the development of allergic, autoimmune, and cancerous diseases[3,7]. Dectin-1 signaling is reported to aggravate airway hypersensitivity by inducing proallergic chemokines and mucus[8], and administration of a Dectin-1 agonistic ligand curdlan evokes arthritis, spondyloarthritis and ileitis in an autoimmune-prone mouse with a spontaneous mutation in ZAP-70[9,10]. Recently, we reported that Dectin-1-deficiency causes overexpansion of a commensal bacterium *Lactobacillus murinus*, which induces regulatory T (Treg) cell differentiation, in the mouse colon, resulting in the suppression of dextran sodium-sulfate (DSS)-induced colitis[11]. Dectin-1 signaling also induces IL-17F in the downstream and suppresses the growth of a group of commensal bacteria including *Clostridium* cluster XIVa, which also promotes Treg differentiation, by inducing specific antimicrobial proteins[12,13]. Therefore, although Dectin-1 basically activates the immune system to eradicate pathogens, this activity, on the other hand, can also trigger or enhance inflammation-related diseases.

A large number of studies were carried out about the effects of β-glucans and Dectin-1 signaling on the development of tumors, especially because we are taking lots of β-glucans as a component of daily foods such as yeasts, mushrooms, and seaweeds, and also as a herbal medicine[14,15]. However, these results are sometimes contradictory each other. Many reports suggest a protective role for Dectin-1 signaling in the development to tumors. Oral administration of high molecular weight (MW) β-glucans suppresses the growth of subcutaneously inoculated Lewis lung carcinoma[16]. Orally administrated β-glucans also enhance the effects of antibody therapy through modification of antitumor immunity[17,18]. Furthermore, β-glucan injection prevents mouse liver metastases of transferred C-26 colon carcinoma cells[19]. Although some studies suggest direct tumor cell growth inhibitory activity of β-glucans, it is now recognized that β-glucans have no such cytotoxic effects[20,21]. Dectin-1 signaling was reported to enhance tumor-killing activity of NK cells through induction of INAM on DCs by binding N-glycans on B16 melanoma cells[22]. Dectin-1 upregulates TNFSF15 and OX40L expression in DCs to promote anti-tumorigenic Th9 cell differentiation by activating Raf1 and NF-κB[23]. Dectin-1 also suppresses chemical carcinogen-induced hepatocellular carcinogenesis by suppressing Toll-like receptor (TLR) 4-dependent inflammation[24]. In contrast, a tumor promotive role for the Dectin-1 signaling was reported in pancreatic tumorigenesis. In that report, Dectin-1 recognizes noncanonical endogenous ligand galectin-9 expressed on pancreatic ductal adenocarcinoma to suppress anti-tumorigenic M1 macrophage differentiation[25]. However, until now, no study has elucidated the function of Dectin-1 in the development of intestinal tumors.

In this study, we investigate the role of Dectin-1 in colorectal tumorigenesis by analyzing mouse intestinal tumor models and CRC patients. We show that Dectin-1 signaling rather promotes the development of colorectal tumors by enhancing the production of prostaglandin E2 (PGE$_2$), which facilitates CRC development and suppresses the expression of tumor-inhibitory IL-22-binding protein (IL-22BP). These observations reveal a tumor promotion pathway in the intestinal tract and suggest Dectin-1 as a possible target for the therapy of CRC.

## Results

### Dectin-1-deficiency suppresses colorectal tumor development in a commensal microbiota-independent manner

To investigate the potential role of Dectin-1 in intestinal tumorigenesis, we first examined the influence of *Clec7a* gene deficiency

(*Clec7a$^{-/-}$*) on the development of colorectal tumor in DSS-treated *Apc$^{Min}$* (*Apc$^{Min}$*-DSS) mice, in which intestinal adenoma is induced due to enhanced β-catenin activity[26]. Body weight loss in *Apc$^{Min/+}$Clec7a$^{-/-}$* mice after DSS-treatment was significantly milder than that in *Apc$^{Min/+}$* mice (Fig. 1a), and at day 28 after DSS-treatment, polyp number in the colon was significantly reduced in *Apc$^{Min/+}$Clec7a$^{-/-}$* mice (Fig. 1b, c). Shortening of colon length, one of the symptoms of colitis, was milder in *Apc$^{Min/+}$Clec7a$^{-/-}$* mice (Fig. 1c, d), as observed in DSS-treated *Clec7a$^{-/-}$* mice[11].

We also examined the effect of Dectin-1-deficiency on spontaneous tumor development in *Apc$^{Min}$* mice. After sacrificing mice at 20–23 weeks of age, we found the number of polyps in both small and large intestines was significantly decreased in *Apc$^{Min/+}$Clec7a$^{-/-}$* mice compared with that in *Apc$^{Min/+}$* mice (Supplementary Fig. 1a–d). Histologically, complex budding of crypts and high- or low-grade dysplasia or hyperplasia in both colon and rectum were observed in *Apc$^{Min/+}$* mice, while these histopathological changes were reduced in *Apc$^{Min/+}$Clec7a$^{-/-}$* mice (Fig. 1e, f).

We further analyzed a chemically induced colorectal tumor model in which mice were treated with azoxymethane (AOM), followed by 3 cycles of DSS treatment (AOM-3DSS). *Clec7a$^{-/-}$* mice exhibited milder body weight loss than wild-type (WT) mice during 3 cycles of DSS-treatment (Fig. 1g), and the polyp number in *Clec7a$^{-/-}$* mice was significantly smaller than that in WT mice (Fig. 1h, i). Shortening of the colon length was significantly milder in *Clec7a$^{-/-}$* mice (Fig. 1i, j). When mice were treated with AOM and only one cycle of DSS, anal prolapse (Supplementary Fig. 1e, f) and shortening of colon length (Supplementary Fig. 1g, h) were also significantly milder in *Clec7a$^{-/-}$* mice. Although all the WT mice developed colorectal polyps, only half of *Clec7a$^{-/-}$* mice developed polyps with smaller sizes (Supplementary Fig. 1f, i). Histologically, complex budding of crypts, high-grade dysplasia or hyperplasia were only observed in both colon and rectum section of WT mice, and only mild levels of the lesions and crypt changes were observed in those of *Clec7a$^{-/-}$* mice (Fig. 1k, l). Submucosal invasion was not observed in these AOM-DSS-treated mice. All these results indicate that blockade of Dectin-1 signaling inhibits intestinal tumorigenesis.

We previously reported that DSS-induced colitis is suppressed in *Clec7a$^{-/-}$* mice, because colonization of a regulatory T cell (Treg)-inducing commensal bacterium *Lactobacillus murinus* is increased in the intestine of *Clec7a$^{-/-}$* mice[11]. Actually, real-time RT-PCR and flow-cytometry analysis showed that *Foxp3* (Treg marker) expression and Foxp3+CD4+ T cell proportion in polyps and non-polyp tissue were significantly increased in Dectin-1-deficient mice (Supplementary Fig. 1r, t). To examine the effects of intestinal microbiota in the regulation of tumorigenesis in *Clec7a$^{-/-}$* mice, we continuously co-housed *Clec7a$^{-/-}$* mice with WT mice after weaning to harmonize intestinal microbiota and treated them with AOM-3DSS. Body weight loss and shortening of colon length caused by DSS-treatment were similar between these co-housed mice (Supplementary Fig. 1j, l, m), but the polyp number of colon and rectum in *Clec7a$^{-/-}$* mice was still significantly fewer than those in WT mice (Supplementary Fig. 1n), especially of larger sizes (diameter >3 mm, Supplementary Fig. 1k, l), as that seen in separately housed mice.

We next examined the involvement of commensal microbiota in the effect of Dectin-1-deficiency on intestinal tumorigenesis in *Apc$^{Min}$*-DSS model. *Apc$^{Min/+}$* and *Apc$^{Min/+}$Clec7a$^{-/-}$* mice were pretreated with antibiotics (ABX) for 3 weeks, then DSS + ABX were administered for 1-week, followed by continuous ABX-treatment for another 4 weeks. *Apc$^{Min/+}$Clec7a$^{-/-}$* mice developed less number of polyps compared with *Apc$^{Min/+}$* mice, although total polyp number was decreased in ABX-treated mice (Supplementary Fig. 1o–1q). To further exclude the effects of commensal microbiota on Dectin-1-mediated colorectal tumor regulation, we generated germ-free (GF) mice and treated them with AOM-3DSS. Thirty-six weeks after AOM administration, we observed that

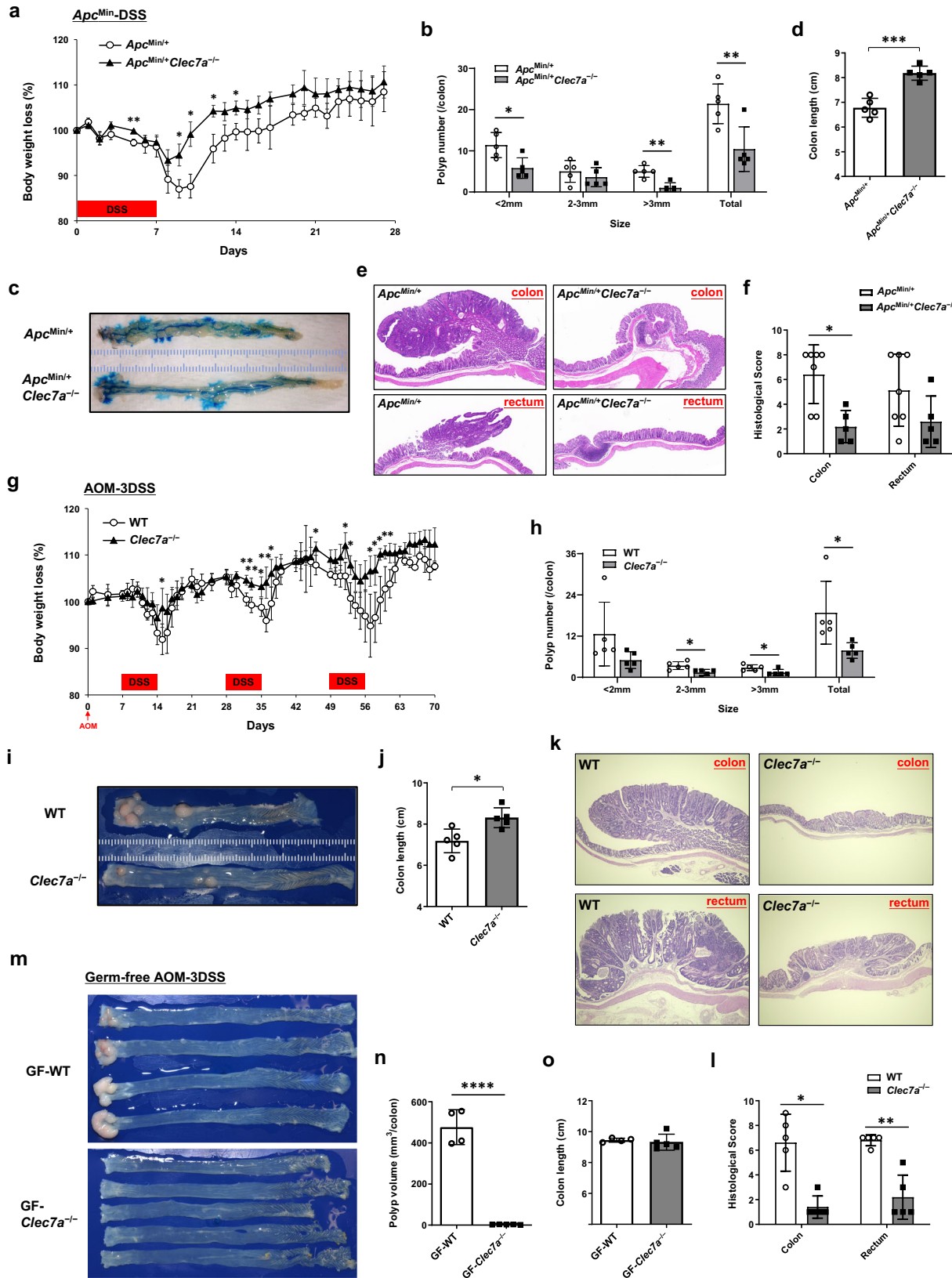

colorectal polyps were developed only in WT but not in *Clec7a*[−/−] GF mice (Fig. 1m, n), although the colon length of these mice was similar (Fig. 1m, o). Intestinal *Foxp3* expression was similar between WT and *Clec7a*[−/−] GF mice (Supplementary Fig. 1s), indicating the Treg population is similar. We also found the expression of *Ccnd1* encoding Cyclin-

D1, which promotes cell cycle progression, was downregulated in *Clec7a*[−/−] mice under both SPF and GF conditions (Supplementary Fig. 1r, s), consistent with the suppression of tumor growth in *Clec7a*[−/−] mice. These results indicate that commensal microbiota are not involved in the suppression of tumorigenesis in *Clec7a*[−/−] mice.

**Fig. 1 | Dectin-1-deficiency suppresses development of colorectal tumors without involvement of commensal microbiota. a–f** $Apc^{Min/+}$ and $Apc^{Min/+}Clec7a^{-/-}$ mice at 8–10 weeks old were treated with 1% DSS for 7 days and were sacrificed 4 weeks later. Body weight changes were observed chronologically after DSS-treatment (**a**, day5, **$*P$ = 0.0098; day9, *$P$ = 0.0297; day10, *$P$ = 0.0127, day12, *$P$ = 0.0260; day14, *$P$ = 0.0364). Number of colorectal polyps with indicated size (**b**, <2 mm, *$P$ = 0.0130; >3 mm, **$P$ = 0.00139; Total, **$P$ = 0.00948), representative gross pathology of the colon (**c**) and colon length (**d**, ***$P$ = 0.00018) are shown ($n$ = 5/group). Representative H.E.-stained histology of the polyps (**e**) and histopathological scores (**f**, $Apc^{Min/+}$ n = 7, $Apc^{Min/+}Clec7a^{-/-}$ n = 5; *$P$ = 0.0152) are shown. **g–l** WT and $Clec7a^{-/-}$ mice were i.p. administrated with AOM, and 7 days later, they were treated with 1% DSS for 3 cycles (AOM-3DSS) as described in Methods. Body weight loss was inspected chronologically during the induction of colorectal tumor (**g**, day15, *$P$ = 0.0352; day32, **$P$ = 0.00129; day35, *$P$ = 0.0237; day36, **$P$ = 0.0088; day37, *$P$ = 0.0338; day46, *$P$ = 0.0475; day52, *$P$ = 0.0339; day53, *$P$ = 0.0179; day57, *$P$ = 0.0290; day58, *$P$ = 0.0276; day59, *$P$ = 0.0119; day60, *$P$ = 0.0251;

day61, *$P$ = 0.0196). Mice were sacrificed at week 16 after AOM administration, and number of colorectal polyps with indicated size (**h**, 2–3 mm, *$P$ = 0.0150; >3 mm, *$P$ = 0.0339; Total, *$P$ = 0.0312), gross pathology of the colon (**i**), colon length (**j**, *$P$ = 0.0103), representative histology of the polyps (**k**) and histopathological scores (**l**, *$P$ = 0.0159, **$P$ = 0.0079) are shown ($n$ = 5/group). **m–o** GF WT and $Clec7a^{-/-}$ mice were treated with AOM-3DSS. Thirty-six weeks after AOM administration, these mice were sacrificed, and gross pathology of the colon (**m**) and colon length (**o**) are shown. Colorectal tumor sizes in indicated mice were measured and total polyp volume (**n**, ****$P$ < 0.0001) is shown. The polyp volume was calculated as (diameter)³ for circular polyps and as (Short diameter)² × (long diameter) for elliptic polyps (n = 4/GF WT group, $n$ = 5/GF $Clec7a^{-/-}$ group). Data in (a-f) are representatives of five, in **g–l** are representatives of six, and in **m-o** are representatives of three independent experiments. Data in **a, b, d, f, g, h, j, l, n, o** are presented as means ± SD. Data in **a, b, d, g, h, j, n, o** are analyzed using unpaired two-tailed Student's $t$-test and in **f, l** using two-tailed Mann–Whitney $U$-test. Source data are provided in the Source Data file.

## Infiltration of myeloid-derived suppressor cells (MDSCs) is decreased in polyps in $Clec7a^{-/-}$ mice

To elucidate the mechanism by which Dectin-1-deficiency suppresses intestinal tumorigenesis, we examined Dectin-1 expressing cells in tumors of $Apc^{Min}$-DSS mice. Flow cytometric analysis showed that Dectin-1 was mainly expressed on CD103⁻CD11c⁻CD11b⁺ cells in colonic polyps (Fig. 2a, 92.4%). Nearly 75% of these Dectin-1⁺ cells were MHC-II⁻ cells, in which approximately 50% were Ly6C$^{int}$Ly6G⁺CD11b⁺, 5% were Ly6C$^{hi}$CD11b⁺ and 19% were MHC-II⁻Gr1⁻CD11c⁻CD11b⁺ cells (Supplementary Fig. 2a). These tumor-infiltrating Ly6C$^{int}$Ly6G⁺CD11b⁺ and Ly6C$^{hi}$CD11b⁺ myeloid cells are commonly known as granulocytic polymorphonuclear- (PMN-) and monocytic- (M-) MDSCs[27], respectively.

We next analyzed tumor-infiltrating cell populations in the colon, and found that infiltration of T cells (including αβ & γδ T cells), B cells (including IgG⁺ B cells), and DCs (MHC-II⁺CD11c⁺ cells) was significantly increased in $Apc^{Min/+}Clec7a^{-/-}$ mice (Fig. 2b; Supplementary Fig. 2b, c), suggesting enhanced immune responses in $Apc^{Min/+}Clec7a^{-/-}$ mice compared with $Apc^{Min/+}$ mice. In contrast, MHC-II⁻CD11c⁻CD11b⁺ population was decreased in polyps of these $Clec7a^{-/-}$ mice (Fig. 2c; Supplementary Fig. 2c). We further showed that both CD11c⁻CD11b⁺Ly6C$^{int}$ and CD11c⁻CD11b⁺Ly6C$^{hi}$ population were reduced in the polyps of $Apc^{Min/+}Clec7a^{-/-}$ mice (Supplementary Fig. 2d), suggesting that Dectin-1 signaling facilitates MDSC expansion in intestinal tumors.

IL-6 and IL-1β are reported to promote MDSC expansion[28,29]. When we purified EpCAM⁺ epithelial cells, CD11b⁺ and CD11c⁺ myeloid cells, Thy1.2⁺ T cells, CD19⁺ B cells and other CD45⁺ leukocytes from intestinal polyps and non-polyp tissues, we found that $Il6$ mRNA was expressed by both myeloid cells and T cells, and $Il1b$ was exclusively detected in myeloid cells in both polyp and non-polyp tissues (Fig. 2d). When these polyp-infiltrating CD11b⁺ and CD11c⁺ cells were stimulated with a β-glucan OXCA from *Candida albicans*, $Il6$ and $Il1b$ mRNA expression and IL-6 and pro-IL-1β protein expression were induced, and this induction was inhibited by a Dectin-1 antagonist laminarin (Fig. 2e; Supplementary Fig. 2e). These results support the idea that MDSC expansion in WT mouse polyps is caused by Dectin-1 signaling-induced cytokine(s) at least in part.

## The expression of genes for PGE₂ synthesizing enzymes is decreased, while that for IL-22BP is increased in $Clec7a^{-/-}$ mice

Next, we examined gene expression by RNA-seq analysis in colonic tumors and non-tumor tissues from AOM-3DSS-treated GF mice, as we could detect the difference between WT and $Clec7a^{-/-}$ mice more clearly than using SPF mice. This is probably because the influence of microbiota on gene expression was excluded in GF mice. The expression of genes associated with T cells ($Cd3e$, $Cd28$, $Cd69$, $Ccr6$, $Ccr9$), Th1 cells ($Cd4$, $Tbx21$, $Ifng$, $Il12a$, $Cxcr3$), B cells ($Cd19$, Ig-associated, $Cxcr5$), and DCs ($Cd80$, $Cd86$, $Itgax$, $H2$-$Ab1$, $Ccr7$) was upregulated in

polyps of $Clec7a^{-/-}$ mice (Fig. 3a; Supplementary Fig. 3a, b), consistent with the results in Fig. 2b, c. The expression of cytokines such as $Il1b$, $Il17a$ and $Il23a$, and chemokines such as $Ccl1$, $Ccl2$, and $Cxcl2$ was downregulated in polyps of $Clec7a^{-/-}$ mice (Fig. 3a; Supplementary Fig. 3b). Interestingly, the expression of IL-22-binding protein (IL-22BP; gene: $Il22ra2$), which is suggested to be involved in colorectal tumor regulation[30], was upregulated in polyps of $Clec7a^{-/-}$ mice (Fig. 3a; Supplementary Fig. 3a).

By KEGG gene enrichment analysis, we found that immune related genes such as antibody production, lymphocyte activation, NK cell-mediated cytotoxicity, and antigen presentation were upregulated in tumor tissues of $Clec7a^{-/-}$ mice compared to that of WT mice (Fig. 3b). In contrast, pathways such as Wnt signaling, IL-17 signaling, Ras signaling, and arachidonic acid metabolism were downregulated in $Clec7a^{-/-}$ mouse colon (Fig. 3b). The expression of characteristic genes of MDSC[31,32] was significantly decreased in $Clec7a^{-/-}$ mouse polyps (Fig. 3c), consistent with decreased MDSC infiltration in colonic tumors in $Clec7a^{-/-}$ mice (Fig. 2c). The expression of MDSC-differentiation-promotive cytokines, such as $Csf2$, $Il6$, $Il1b$ and $Tnf$[28,29], was decreased in $Clec7a^{-/-}$ mouse polyps (Fig. 3a; Supplementary Fig. 3a, b). Interestingly, we found that the expression of genes encoding prostaglandin E₂ (PGE₂) synthases, such as $Pla2g2a/2e$ (Phospholipase A2 group IIA/IIE), $Ptgs1/2$ (Cox1/Cox2), and $Ptges2/3$ (PGE₂ synthase 2/3), was downregulated in polyps in $Clec7a^{-/-}$ mice (Fig. 3d; Supplementary Fig. 3a). The expression of $Clec7a$ was elevated in polyps compared with non-polyp tissues (Supplementary Fig. 3c), reflecting the expansion of tumor-infiltrating MDSCs in which Dectin-1 was highly expressed.

Then, we further analyzed gene expression in colonic polyp-infiltrating CD11b⁺ and CD11c⁺ cells from AOM-3DSS-treated mice by single-cell RNA-sequencing (scRNA-seq) analysis. Fifteen types (clusters) of myeloid cells were identified and they were further classified into four big groups; these are monocyte-associated (cluster #0, 1, 2), neutrophil-associated (#3, 8, 12), macrophage-associated (#5, 7, 10, 14) and DC-associated (#4, 6, 9, 11,13) (Fig. 3e; Supplementary Fig. 3d). Judging from the characteristic gene expression patten, cluster 2 was identified as M-MDSC and cluster 3 was PMN-MDSC. We found that both of these MDSC populations were decreased in $Clec7a^{-/-}$ mice compared with those in WT mice (Fig. 3e, f; Supplementary Fig. 3d). In contrast, all the DC populations consisting of 5 subgroups were increased in $Clec7a^{-/-}$ mice compared with WT mice. Other myeloid-cell populations were not changed in $Clec7a^{-/-}$ mice (Supplementary Fig. 3e).

To further investigate the influence of Dectin-1 on the differentiation of each MDSC cell type, we carried out pseudo-time trajectory analysis by focusing on the monocyte group (cluster #0, 1, 2) and the neutrophil group (cluster #3, 8, 12). We found that the gene expression profiles between neutrophils and PMN-MDSC were closely

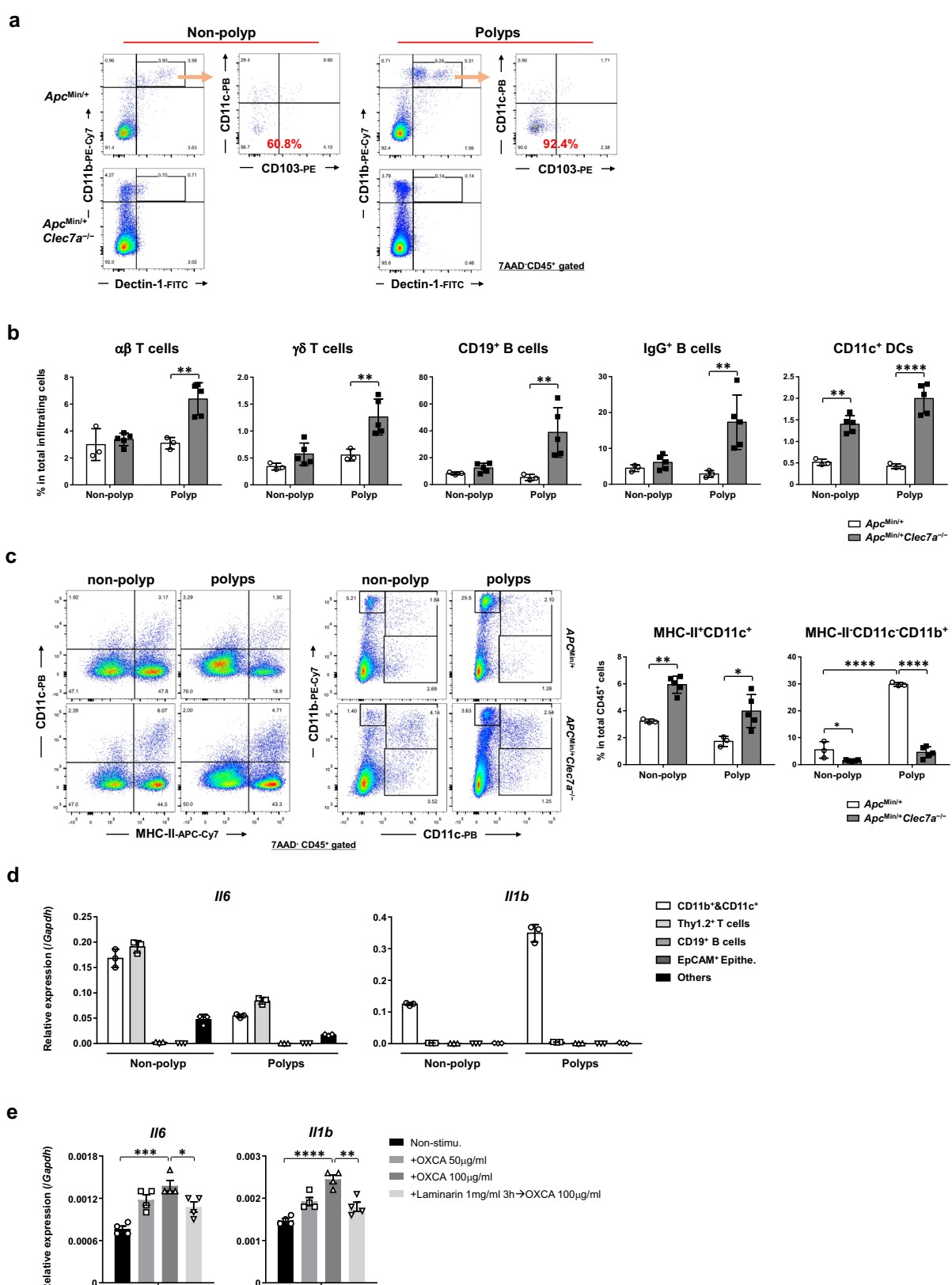

related, and the expression levels of signature genes of MDSC, such as *Arg2*, *S100a9* and *Csf3r*, were gradually increased from neutrophil to PMN-MDSC (Fig. 3g). Similar relationship was also observed between monocyte and M-MDSC (Fig. 3g). During both types of MDSC differentiation, *Il1b* expression was also upregulated, but the average level of the mRNA as well as the protein in *Clec7a−/−* mice was lower than that in

WT mice (Supplementary Fig. 3f, g). By the trajectory analysis, PMN-MDSCs were shown to be differentiated from neutrophils through neutro.→ mdsc transitional state, in which typical MDSC gene *Arg2* was not expressed yet (Fig. 3h). We found that the differentiation of PMN-MDSCs was suppressed in *Clec7a−/−* mice (WT54.8% vs. KO10.8%) due to the blockade of the transition from neutro.→mdsc transitional cells to

**Fig. 2 | MHC-II⁺CD11c⁻CD11b⁺ cell population is decreased in the colon of**
***Clec7a⁻/⁻* mice. a** $Apc^{Min/+}$ and $Apc^{Min/+}Clec7a^{-/-}$ mice were treated with 1% DSS for
7 days and were sacrificed 4 weeks later. Flow cytometric analysis was carried out to
identify Dectin-1-expressing cells in colorectal polyps and non-polyp tissues.
**b**, **c** $Apc^{Min/+}$ and $Apc^{Min/+}Clec7a^{-/-}$ mice were treated with 1% DSS for 7 days and were
sacrificed 4 weeks later. Proportions of T cells, B cells and DCs in total tissue-
infiltrating cells (**b**, αβT cells, **$P = 0.0020$; γδT cells, **$P = 0.0076$; CD19⁺ B cells,
**$P = 0.0069$; IgG⁺ B cells, **$P = 0.0058$; CD11c⁺ DCs, **$P = 0.0014$, ****$P < 0.0001$),
and proportions of MHC-II⁺CD11c⁺ and MHC-II⁺CD11c⁻CD11b⁺ cells in tissue-
infiltrating CD45⁺ cells (**c**, MHC-II⁺CD11c⁺, **$P = 0.0039$, *$P = 0.0157$; MHC-II⁻CD11c
⁻CD11b⁺, *$P = 0.0563$, ****$P < 0.0001$) in non-polyp tissues or polyps in the colon
were examined by flow cytometry ($n = 3/Apc^{Min/+}$ group, $n = 5/Apc^{Min/+}Clec7a^{-/-}$
group). **d** WT mice were treated with AOM-3DSS and were sacrificed after 12 weeks.
Indicated types of cells in polyps and non-polyp tissues were purified with auto-
MACS and *Il6* and *Il1b* expression was examined by RT-qPCR ($n = 3$ biologically
independent samples/group). **e** WT mice were administrated with AOM followed by
3 cycles of DSS. After 11 weeks of AOM treatment, these mice were sacrificed and
CD11b⁺ and CD11c⁺ cells were purified from colorectal polyps with autoMACS. The
sorted cells were stimulated with an agonistic β-glucan OXCA for 16 h with or
without pretreatment with an antagonistic β-glucan laminarin, for 3 h, then, *Il6* and
*Il1b* expression were determined by RT-qPCR ($n = 4$ biologically independent
samples/group; in *Il6* panel, ***$P = 0.0002$, *$P = 0.0382$; *Il1b* panel, ****$P < 0.0001$,
**$P = 0.0012$). Data in **a**–**e** are representatives of two independent experiments.
Data in **b**–**e** are expressed as means ± SD. Data in **b**, **c** are analyzed using two-way
analysis of variance (ANOVA) and in **e** using one-way ANOVA followed by Tukey's
multiple-comparisons test. Source data are provided in the Source Data file.

MDSCs, resulting in the accumulation of the transitional cells
(KO66.3% vs. WT10.5%) (Fig. 3h). Similarly, M-MDSC population was
differentiated from monocytes through monocyte→M1 macrophage
transitional cells, and this process was again suppressed in *Clec7a⁻/⁻*
mice (KO17.0% vs. WT39.8%). Transition from monocyte→M1 macro-
phage transitional cells to M-MDSC was significantly suppressed in
*Clec7a⁻/⁻* mice and accumulation of transitional cells were observed
(KO63.6% vs. WT37.0%), although the frequency of monocytes was not
much different between WT and *Clec7a⁻/⁻* mice (19.4% vs. 23.2%). These
results suggest that Dectin-1 signaling plays important roles in the
development of tumor-infiltrating MDSCs.

To confirm the promotive role of Dectin-1 signaling in MDSC
differentiation, we induced MDSC in vitro from bone marrow cells with
GM-CSF and IL-6[33], and examined the effect of Dectin-1 agonistic ligand
curdlan. As shown in Fig. 3i, j, curdlan significantly enhanced
Ly6C⁺CD11b⁺ cell differentiation. The expression of genes typical for
mouse MDSCs, such as *Arg2* and *Nos2*[32], was upregulated by curdlan
stimulation in a dose-dependent manner (Supplementary Fig. 3h).
When these cells were co-cultured with CD4⁺ T cells isolated from the
spleen, these Dectin-1-induced myeloid cells suppressed anti-CD3/
CD28-induced T cell proliferation and IFN-γ production in these cells
(Fig. 3j), indicating these cells represent MDSCs.

When AOM-3DSS-induced colonic polyp-derived MHC-II⁻Gr1⁺
cells were co-cultured with splenic CD4⁺ T cells, T cell proliferation
was suppressed by these tumor-infiltrating Gr1⁺ cells from WT mice,
while Gr1⁺ cells from *Clec7a⁻/⁻* mouse polyps suppressed T cell
population only weakly (Fig. 3k). Co-culturing CD4⁺ T cells with Gr1⁺
cells isolated from the spleen of AOM-3DSS-treated mice also
showed the similar impairment of immunosuppressive activity of
Dectin-1-deficient Gr1⁺ cells (Supplementary Fig. 3i, j). Drastic
impairment of NOS2 secretion by intestinal polyp-infiltrating
CD11c⁻CD11b⁺ cells from *Clec7a⁻/⁻* mice is consistent with this T cell
suppression (Fig. 3l). All these observations indicate that Dectin-1
signaling facilitates both differentiation and immune-suppressive
activity of intestinal MDSCs.

Both T cells and B cells are suggested to be involved in antitumor
immunity[34–36], and both of them are increased in the intestinal polyps
of *Clec7a⁻/⁻* mice (Fig. 3a, b; Supplementary Fig. 3a). However, *Rag2⁻/⁻*
mice treated with AOM-3DSS developed even fewer polyps compared
with WT mice (Supplementary Fig. 3k, l), suggesting that these lym-
phocytes are not the major regulator of the intestinal tumor devel-
opment in our experimental setting.

**Dectin-1 signaling directly enhances PGE₂ production in MDSCs**
As shown in Fig. 3d and Supplementary Fig. 3a, the expression of genes
involved in PGE₂ synthesis was upregulated in polyps of AOM-3DSS-
treated GF WT mice, but not in GF *Clec7a⁻/⁻* mice. Decreased expres-
sion of *Ptgs2* in colonic polyps and non-polyp tissues was also observed
in *Clec7a⁻/⁻* mice under SPF conditions (Fig. 4a), and PGE₂ concentra-
tion in colonic tissue homogenate was decreased in these mice
(Fig. 4b). Decreased *Ptgs2* expression in intestinal polyps was also

observed in $Apc^{Min/+}Clec7a^{-/-}$ mice (Fig. 4c), and its expression was only
detectable in CD45⁺ leukocytes, but not in EpCAM⁺ epithelial cells, in
both polyps and non-polyp tissues (Fig. 4d).

By utilizing scRNA-seq analysis on polyp-infiltrating myeloid
cells, we found that *Ptgs2* and *Clec7a* was highly expressed in
*Arg2⁺S100a9⁺* MDSCs (Fig. 4e, g). Since MDSC population was
decreased in *Clec7a⁻/⁻* mice, *Ptgs2⁺* cell population was also
decreased in mutant mice. We also found that *Ptgs2* expression
levels in myeloid cells were lower in *Clec7a⁻/⁻* mice than that in WT
mice (Fig. 4f). The expression of other PGE₂ synthase genes such as
*Pla2g6*, *Pla2g7*, *Ptges1-3* and *Ptgs1* was also observed in MDSCs at
high levels (Fig. 4g). MDSCs also expressed *Nos2* and *Vegfa* at high
levels. Similar results were also obtained by reanalyzing a public
NCBI GEO dataset (GSE196054, scRNA-seq) of intestinal polyp-
infiltrating CD45⁺ leukocytes from AOM-3DSS-treated C57BL/6 mice
(Supplementary Fig. 4a−c). Consistent with this analysis, we found
the major enzymes involved in PGE₂-synthesis were preferentially
expressed in tumor-infiltrating CD11c⁻CD11b⁺ MDSCs but not in
CD11c⁺ DCs from AOM-3DSS-treated mice (Fig. 4h).

When polyp-infiltrating CD11b⁺ cells from $Apc^{Min/+}$ mice were
treated with curdlan, the expression of *Ptges3*, *Ptgs1*, and *Ptgs2* was
significantly enhanced, and the induction was inhibited by laminarin
(Fig. 4i). Treatment of polyp-infiltrating leukocytes with curdlan also
induced PGE₂ synthases both at the mRNA and protein levels (Sup-
plementary Fig. 4d–f), indicating that Dectin-1 signaling directly
induces PGE₂ synthesis in intestinal tumor-infiltrating MDSCs.

Previously, we and others showed that Dectin-1 induces cytokines
through activation of Syk-CARD9-NF-κB pathway[37,38]. Furthermore,
COX2 is suggested to be induced by NF-κB p65[39–41]. Thus, we examined
the roles of the Syk-NF-κB pathway in the Dectin-1-induced PGE₂-syn-
thesizing-enzyme expression. As shown in Fig. 4j, *Ptges3*, *Ptgs1* and
*Ptgs2* expression were suppressed by the inhibitors against NF-κB
(IκBα: BAY11-7082) and SYK (R406), suggesting that these PGE₂-syn-
thesizing enzymes are induced by the activation of the Dectin-1-Syk-
NF-κB pathway.

As PGE₂ levels were reduced in *Clec7a⁻/⁻* mouse colon, we exam-
ined the effect of PGE₂ on intestinal tumorigenesis. DSS-treated
$Apc^{Min/+}Clec7a^{-/-}$ mice were i. p. administered with PGE₂ every other day
for 4 weeks, and colorectal tumor development was examined. As
shown in Fig. 5a, b, colorectal polyp number was increased to the
similar level of that in $Apc^{Min/+}$ mice. Infiltration of MHC-II⁺CD11c⁺ DCs
was decreased, whereas that of MHC-II⁻CD11b⁺Ly6C⁺/Ly6G⁺ MDSCs
was increased in polyps after PGE₂-administration (Fig. 5c; Supple-
mentary Fig. 4g, h). The expression of MDSC marker genes such as
*Itgam*, *S100a9*, and *Clec4e*, and tumor proliferation-associated genes,
*Ccnd1* (Cyclin D1) and *Myc* (c-Myc), was enhanced after PGE₂-admin-
istration (Fig. 5d), indicating that MDSC expansion and tumor growth
are promoted by PGE₂. When intestinal polyp-infiltrating cells were
treated with PGE₂ in vitro, the expression of these MDSC-associated
genes was increased, indicating that these genes were directly induced
by PGE₂ (Fig. 5e).

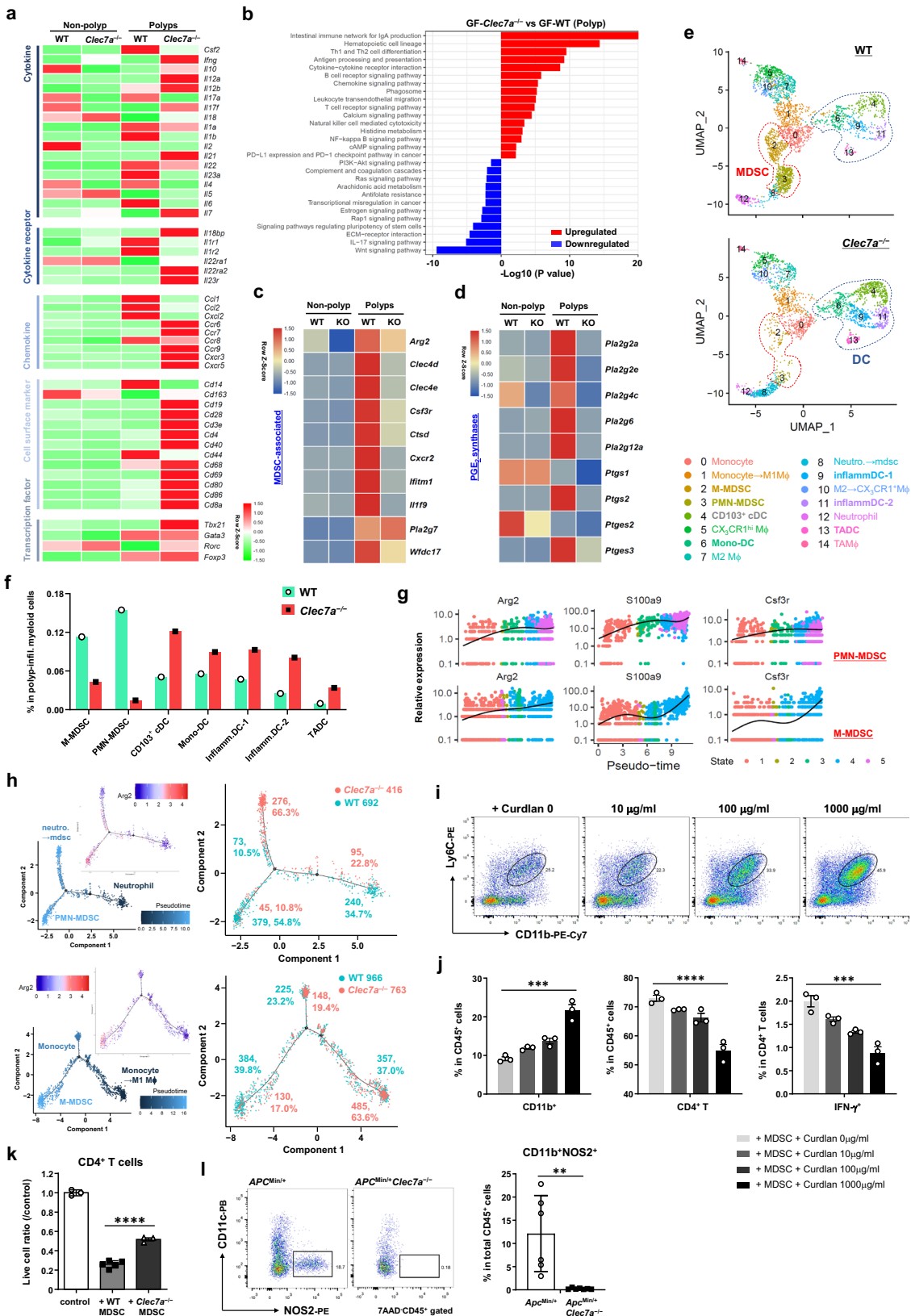

## Administration of laminarin suppresses the development of intestinal polyps

To examine therapeutic effects of Dectin-1-blockade on intestinal tumorigenesis, AOM-treated WT mice were fed with 5% laminarin-containing food during 3 cycles of DSS-treatment. Consistent with the observations in *Clec7a*[−/−] mice, significantly fewer colorectal polyps

developed in laminarin-containing-food-fed mice compared to normal-food-fed mice (Fig. 5f, g). Blocking Dectin-1 signaling suppressed intestinal PGE$_2$ production (Fig. 5h), accompanied with the reduction of CD11b gene (*Itgam*) expression (Fig. 5i). These results suggest that Dectin-1 signaling facilitates intestinal tumor development by enhancing PGE$_2$, which promotes MDSC expansion.

**Fig. 3 | The expression of genes encoding PGE₂-synthesizing enzymes and IL-22BP is regulated by Dectin-1 signaling. a–d** GF WT and *Clec7a⁻/⁻* mice treated with AOM-3DSS shown in Fig. 1m were sacrificed at 36 weeks after AOM administration, and RNA-seq analysis was carried out by using RNAs from polyps and non-polyp tissues in the vicinity (pooled 4 WT and 5 *Clec7a⁻/⁻* GF mice). Expression levels of genes for cytokine, chemokine, cell-surface marker, and T cell transcription factor in WT and *Clec7a⁻/⁻* mice are shown as heatmaps (**a**). Cell populations, signaling pathways and metabolic pathways were compared between *Clec7a⁻/⁻* and WT mouse polyps by using the KEGG database. Log2 (fold-change) > 1 is upregulated genes, while log2 (fold-change) <−1 is downregulated genes (**b**). Expression levels of MDSC-associated genes are presented as a heatmap (**c**). Expression levels of PGE₂-synthase genes in polyp and non-polyp tissues are shown as a heatmap (**d**). **e–h** SPF WT and *Clec7a⁻/⁻* mice treated with AOM-3DSS for 16 weeks, and CD11b⁺ and CD11c⁺ cells were purified from colorectal polyps and were applied to scRNA-seq analysis. UMAP showing cell type cluster distribution (**e**), proportions of MDSC and DC subsets (**f**), changes of typical MDSC-associated gene expressions during MDSC development (**g**), and the trajectory tracks of the differentiation of PMN-MDSCs and M-MDSCs (**h**) in WT and *Clec7a⁻/⁻* mice are shown (5 WT and 4 *Clec7a⁻/⁻* mice pooled). **i** Bone marrow cells from WT mice were harvested and were stimulated with rGM-CSF and rIL-6 (2 × 10⁵ cells/well) for 5 days. Curdlan with indicated doses was added on the 2nd day and flow cytometry was carried out to examine MDSC differentiation by staining with antibodies against Ly6C and CD11b. **j** After the culture in **i**, differentiated bone marrow cells were further co-cultured with splenic CD4⁺ T cells from WT mice for 2 days in the present of anti-CD3/CD28 microbeads. After the co-culture, proportions of CD11b⁺ and CD4⁺ cells in CD45⁺ cells and the proportion of IFN-γ⁺ cells in CD4⁺ cells were determined by flow cytometry (*n* = 3 biologically independent samples/group; in CD11b⁺ panel, \*\*\**P* = 0.0009; in CD4⁺ T panel, \*\*\*\**P* < 0.0001; in IFN-γ⁺ panel, \*\*\**P* = 0.0001). **k** EpCAM⁻MHC-II⁻Gr1⁺ cells were isolated from colonic polyps of AOM-3DSS-treated WT and *Clec7a⁻/⁻* mice and were co-cultured with splenic CD4⁺ T cells in the presence of anti-CD3/CD28 microbeads. Two days later, flow cytometry was carried out to determine the number of live CD4⁺ T cells (control *n* = 3, +WT MDSC *n* = 5, + *Clec7a⁻/⁻* MDSC *n* = 3, \*\*\*\**P* < 0.0001). **l** Intestinal polyp-infiltrating cells were isolated from *Apc^Min/+^* and *Apc^Min/+^Clec7a⁻/⁻* mice, and proportions of NOS2-producing CD11c⁻CD11b⁺ cells were determined by flow cytometry (*Apc^Min/+^ n* = 6, *Apc^Min/+^Clec7a⁻/⁻ n* = 7, \*\**P* = 0.0027). Data in **i**, **j** are representatives of three and in **k** is representative of two independent experiments. Data from two independent experiments are pooled in **l** and in **j**–**l** are expressed as means ± SD. Data in **j**, **k** are analyzed using one-way ANOVA followed by Tukey's multiple-comparisons test and in **l** using unpaired two-tailed Student's *t*-test. Source data are provided in the Source Data file.

## IL-22BP is critically involved in the suppression of tumor development in *Clec7a⁻/⁻* mice

We showed that the expression of *Il22ra2* was drastically upregulated in polyps of AOM-3DSS-treated GF *Clec7a⁻/⁻* mice (Fig. 3a; Supplementary Fig. 3a, 5a), and IL-22BP levels were also increased in AOM-3DSS-treated SPF *Clec7a⁻/⁻* mice (Supplementary Fig. 5b, c). The expression of *Il22ra2* mRNA was also enhanced in both polyps and non-polyp tissues in *Apc^Min/+^Clec7a⁻/⁻* mouse intestine (Fig. 6a). However, gene expression levels of IL-22 and IL-18, the ligand and the regulator of IL-22BP[30], respectively, were not changed in either AOM-3DSS-treated GF *Clec7a⁻/⁻* or *Apc^Min/+^Clec7a⁻/⁻* mice from the *Clec7a⁺/⁺* counterparts (Fig. 6a; Supplementary Figs. 3a and 5a, d).

Interestingly, we noticed that COX2 mRNA levels were negatively correlated with the *Il22ra2* expression in colonic polyps under both GF and SPF conditions (Fig. 6b). Furthermore, *Il22ra2* expression in intestinal tumors or IL-22BP expression in myeloid cells was suppressed by PGE₂ treatment both in vivo and in vitro (Fig. 6c, d; Supplementary Fig. 5e, f), suggesting that IL-22BP production is negatively regulated by PGE₂.

To identify *Il22ra2*-expressing cells in the intestine, we purified EpCAM⁺ epithelial cells and CD45⁺ leukocytes from intestinal polyps and found that *Il22ra2* mRNA was expressed only in CD45⁺ but not in epithelial cells (Supplementary Fig. 5g). *Il22ra2* was exclusively detected in CD11b⁺ and CD11c⁺ myeloid cells, but not in Thy1.2⁺ T cells, CD19⁺ B cells or other CD45⁺ cells in both colonic polyp and non-polyp tissues (Fig. 6e), and its expression was mainly detected in CD11c⁺ DCs, but not in CD11c⁻CD11b⁺ or other myeloid cells (Fig. 6f). By flow cytometry, IL-22BP protein was detected in both CD11b⁻CD103⁺CD11c⁺ and CD11b⁺CD103^int^CD11c⁺ DCs, but not in CD11b⁺CD11c⁻ cells (Fig. 6g).

IL-22BP levels in MHC-II⁺CD11c⁺ DCs detected by FACS were increased in both polyps and non-polyp tissues after treatment with 5% laminarin in diet during DSS-treatment (Fig. 6h). Furthermore, co-administration of PGE₂ impaired this laminarin-induced IL-22BP upregulation (Fig. 6i).

Since IL-22BP was reported to play an important role in the suppression of intestinal tumor development[30], we next examined the role of IL-22BP in the development of intestinal tumors by introducing *Il22ra2⁻/⁻* mutation into *Apc^Min/+^Clec7a⁻/⁻* mice. We found that the life span of *Il22ra2⁻/⁻Apc^Min/+^Clec7a⁻/⁻* mice was much shorter than that of *Apc^Min/+^Clec7a⁻/⁻* mice, and the life span of heterozygous *Il22ra2⁺/⁻* mice was still shorter than that of *Il22ra2⁺/⁺Clec7a⁻/⁻* mice (Fig. 6j). The polyp number in *Il22ra2⁻/⁻Apc^Min/+^Clec7a⁻/⁻* mice of younger age was larger than that in *Apc^Min/+^Clec7a⁻/⁻* mice (Fig. 6k, l). These results suggest that increased *Il22ra2* expression is important for the suppression of intestinal tumor development.

*Il22ra2* expression was not affected by the treatment with Dectin-1 ligands (Supplementary Fig. 5h), suggesting that IL-22BP expression is not directly regulated by Dectin-1-signaling. IL-18 has been reported to suppress IL-22BP expression[30]. We found that PGE₂-induced suppression of *Il22ra2* expression was rescued by the treatment with anti-IL-18 antibody (Supplementary Fig. 5i), suggesting that PGE₂-induced IL-18 is responsible for the suppression of IL-22BP expression. Actually, *Il18* mRNA expression was upregulated in intestinal CD11b⁺ and CD11c⁺ cells after PGE₂ stimulation (Supplementary Fig. 5j), although its expression level in whole polyp or non-polyp tissues from WT and *Clec7a⁻/⁻* mice were similar (Supplementary Fig. 5a).

Retinoic acid (RA) is also reported to enhance IL-22BP production[42,43] and the expression of enzymes involving RA synthesis such as *Adh1* and *Aldh1a1* was enhanced in polyps in *Clec7a⁻/⁻* mice (Supplementary Fig. 5k, l). By scRNA-seq analysis, we found a group of DCs, which we named tumor-associated DC (TADC) because this DC subset uniquely expresses tumor-associated genes such as *Pclaf*, Spc24 and *Top2a* (Supplementary Fig. 3d), highly expressed *Il22ra2* in *Clec7a⁻/⁻* mice but not in WT mice (Fig. 6m). The expression of these *Aldh* and *A/Rdh* family genes was highly observed in *Clec7a⁻/⁻* TADCs but not in WT cells (Fig. 6n, Supplementary Fig. 5m). Since PGE₂ was reported to suppress retinal dehydrogenases (RALDH) expression in both mouse and human DCs[44], we examined the effect of PGE₂ on the expression of RA-synthase genes in colonic CD11b⁺ and CD11c⁺ myeloid cells, and found that PGE₂ suppressed *Aldh*, but not *A/Rdh*, family gene expression (Fig. 6o; Supplementary Fig. 5n). When RA was added to colonic or splenic myeloid cells, *Il22ra2* expression was significantly recovered (Fig. 6p; Supplementary Fig. 5o). These results suggest that PGE₂ controls IL-22BP expression by regulating IL-18 expression and RA production.

## Dectin-1 signaling enhances PGE₂ synthesis in CRC patients

To evaluate the effects of Dectin-1 signaling on CRC development in humans, we firstly examined *CLEC7A* expression in CRC patients. We found that post-surgery survival rate of patients with high levels of *CLEC7A* expression (relative expression levels>2, compared with *GAPDH*) was significantly lower than those with low levels of *CLEC7A* expression (<2) (Fig. 7a). We also noticed that *CLEC7A* expression levels were positively correlated with the clinical stage of the disease (Fig. 7b, c). Disease progression rates were relatively decreased in CRC patients with *CLEC7A* mutation(s), although the tendency was not statistically significant due to the small number of CRC patients with mutant *CLEC7A* in TCGA database (Supplementary Fig. 6a). These observations suggest that DECTIN-1 signaling in humans also correlated with CRC development. Actually, we detected high levels of

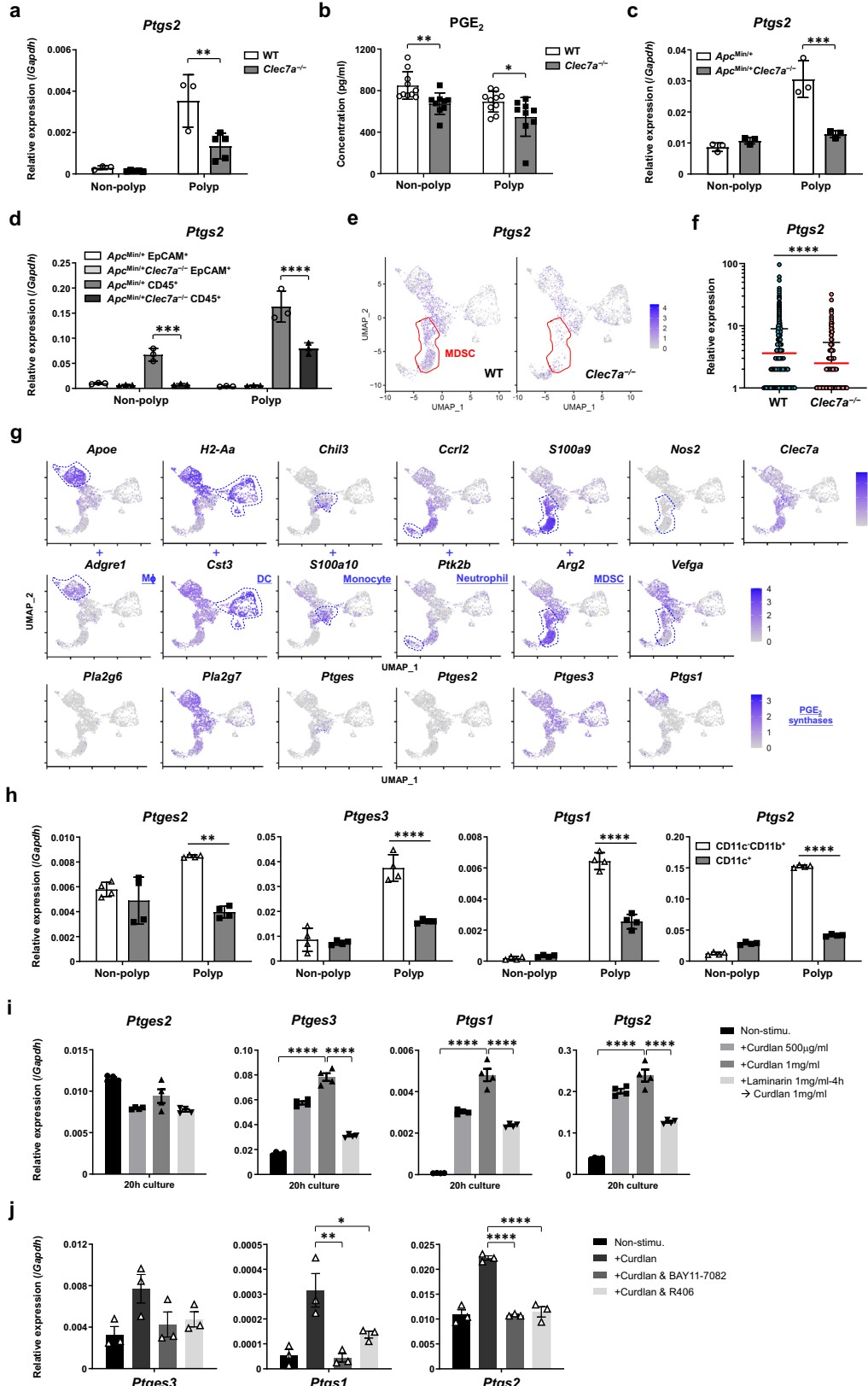

DECTIN-1 expression on CRC-infiltrating cells, while DECTIN-1-expressing cells were rarely detected in human non-tumor tissues (Fig. 7d).

To examine whether the Dectin-1-PGE₂-IL-22BP axis discovered in mice also exists in humans, we measured the *IL22RA2* and *COX2* mRNA in tumors from CRC patients. We found that the *IL22RA2* mRNA was

decreased, while *PTGS2* mRNA was significantly increased in tumors compared with non-tumor tissues (Fig. 7e). This tendency was clearer when samples from the same individuals were compared in a pairwise way (23 pairs of tumor and non-tumor tissues) (Fig. 7f).

We observed that *CLEC7A* expression was positively correlated with *PLA2G2A*, *PTGS2* or *PTGES* expression in CRC patients (Fig. 7g;

**Fig. 4 | Dectin-1 signaling activates the expression of genes involving PGE₂ synthesis. a, b** WT and *Clec7a⁻/⁻* mice under SPF conditions were treated with AOM-3DSS for 16 weeks. The expression of Cox2-encoding *Ptgs2* was determined by RT-qPCR (**a**, WT *n* = 3, *Clec7a⁻/⁻* *n* = 5; **P* = 0.0030), and the concentration of PGE₂ in tissue homogenates was determined by ELISA (**b**, WT *n* = 10, *Clec7a⁻/⁻* *n* = 9; pooled data from 2 experiments and normalized by the protein concentration in each lysate; **P* = 0.0057, *P* = 0.0449). **c, d** The expression of *Ptgs2* in intestinal polyp or non-polyp tissues (**c**, ***P* = 0.0007), or in EpCAM⁺ epithelial cells and in tissue-infiltrating CD45⁺ leukocytes (**d**, ***P* = 0.0008, ****P* < 0.0001) from *ApcMin/+* and *ApcMin/+Clec7a⁻/⁻* mice at 20 weeks old was determined by RT-qPCR (*n* = 3/group). **e–g** According to the scRNA-seq data shown in Fig. 3e–h, *Ptgs2* expressing cells in all polyp-infiltrating myeloid cells (**e**) and *Ptgs2* expression levels in *Ptgs2⁺* myeloid cells (**f**, WT *n* = 1284, *Clec7a⁻/⁻* *n* = 635; ****P* < 0.0001) in WT and *Clec7a⁻/⁻* mice are shown. The expression of indicated genes including myeloid subtype-specific genes and PGE₂ synthase genes in WT mice are shown in **g**. **h** C57BL/6J mice were treated with AOM-3DSS for 12 weeks. CD11c⁺ and CD11c⁻CD11b⁺ cells were purified from colonic polyps and non-polyp tissues with autoMACS, and the mRNA expression of indicated genes was determined by RT-qPCR (*n* = 4 biologically independent samples/group; ***P* = 0.0003, ****P* < 0.0001). **i** Purified CD11b⁺ polyp-infiltrating cells from *ApcMin/+* mice were harvested and were stimulated with curdlan for 20 h. Then, the expression of genes involved in PGE₂ synthesis was determined by RT-qPCR. Laminarin treatment was carried out for 4 h before the treatment with curdlan (*n* = 4 biologically independent samples/group, ****P* < 0.0001). **j** CD11b⁺ and CD11c⁺ myeloid cells from colonic polyps of AOM-3DSS-treated WT mice were stimulated with curdlan (1 mg/ml) in the presence of each one of inhibitors against IκBα (BAY11-7082, 2 μM)) or SYK (R406, 2 μM). After 20 h, PGE₂ synthase expression was determined by qPCR (*n* = 3 biologically independent samples/group; **P* = 0.0380, ***P* = 0.0037, ****P* < 0.0001). Data in **a, c, d, h–j** are representatives of two independent experiments and are expressed as means ± SD. Data in **a–d, h** are analyzed using two-way ANOVA followed by Tukey's multiple-comparisons test, in **f** using unpaired two-tailed Student's *t*-test, and in **i, j** using one-way ANOVA followed by Tukey's multiple-comparisons test. Source data are provided in the Source Data file.

Supplementary Fig. 6b). By analyzing the public data on TCGA, we also found that *CLEC7A* expression was correlated with *PTGS2*, *PTGES* and *PTGES3* in CRC patients (Supplementary Fig. 6c). To confirm Dectin-1-expressing cell profiles and correlation between Dectin-1 and PGE₂ synthases in human CRCs, we carried out scRNA-seq analysis by using pooled cell sequence datasets from six CRC patients (GSE178318)[45]. CRC-associated cells contained lymphocytes, NK, myeloid-derived cells, mast cells, fibroblasts, epithelial, endothelial, and tumor cells, and *CLEC7A* was mainly expressed in myeloid-derived cells (Supplementary Fig. 7a, b). We then extracted *CLEC7A⁺* myeloid cells, and after re-clustering, we found that the majority of *CLEC7A*-expressing cells were *S100A9⁺* MDSCs, and less proportion was expressed in *CD68⁺* macrophages and *FCER1A⁺* DCs (Supplementary Fig. 7c). When we further extracted *CLEC7A⁺ ᵒʳ ⁻ PTGS2⁺* populations, we found that the average expression level of *PTGS2* was significantly higher in *CLEC7A⁺* population compared with *CLEC7A⁻* ones, especially in MDSCs (Supplementary Fig. 7d). In *CLEC7A⁺* population, *PTGS2* expressing cells were largely overlapped with *CLEC4E⁺/S100A9ʰⁱ* MDSCs and only small number of *CD68ʰⁱAPOE⁺* macrophages, *HLA-DQB1ʰⁱFCER1A⁺* DCs and *FCGR3A⁺* monocytes also expressed *PTGS2* (Supplementary Fig. 7e). Other PGE₂ synthase genes such as *PTGS1* and *PTGES3* were also sub-stantially expressed in *CLEC7A*-expressing MDSC, DC and macrophage clusters (Supplementary Fig. 7f).

To further analyze the potential link between Dectin-1, MDSC and PGE₂ synthase in human CRC, we carried out pseudo-time/ trajectory analysis by using myeloid-cell population excluding fully differentiated DCs (Supplementary Fig. 7g, h). After performing the unsupervised simulation, we found that *CD68ʰⁱAPOEʰⁱHLA-DRAʰⁱ* tumor-associated macrophages (TAM) and *CD68ⁱⁿᵗAPOEⁱⁿᵗFCGR3A⁺* monocytes 'differentiate' into *CD68ˡᵒAPOEˡᵒHLA-DRAˡᵒCLEC4EʰⁱS100A9ʰⁱ* MDSCs along the pseudo-temporal trajectory (Fig. 7h left panel). At the same time, we found that in this MDSC 'differentiation' process, expression of *CLEC7A* was gradually upregulated, accompanied with the gradual upregulation of MDSC marker *S100A9* and *CLEC4E*, and with the downregulation of macrophage marker *APOE* and *CD68*. The expression of *IL1B*, *IL6* and *PTGS2* were drastically increased for over 10-100 times from the baseline (Fig. 7h), indicating that these MDSC-promoting factors are induced in the process of MDSC 'differentiation', and finally highly expressed by MDSC itself. Different from mouse tumor-infiltrating MDSCs, the expression of *ARG1* and *NOS2* was not detected in MDSCs (or in any types of tumor-infiltrating leukocytes, Supplementary Fig. 7i) from CRC patients, but the expression of other functional genes, like *IL10* and *VEGFA* that facilitate tumor development, was upregulated in the process of MDSC 'differentiation' (Supplementary Fig. 7i). To confirm these results, we analyzed another scRNA-seq dataset of CRC patients from GEO (GSE146771), and found that the transcription levels of *VEGFA* were also in parallel with *S100A9*,

*PTGS2* and *CLEC7A* in pseudotime-trajectory (Supplementary Fig. 7j), although the expression of neither *ARG1* nor *NOS2* was detectable (in MDSCs or in any other leukocytes).

When CRC tissues or cancer-infiltrating leukocytes were treated with curdlan, expression of PGE₂ synthase genes was induced in vitro (Fig. 7i; Supplementary Fig. 6d). Furthermore, curdlan also upregulated the expression of CD33, a marker of human MDSCs[46], in both tumor and normal tissues (Fig. 7j; Supplementary Fig. 6e), suggesting that Dectin-1 signaling also promotes human MDSC differentiation in CRCs. To further confirm this, we stimulated peripheral blood mono-nuclear cells (PBMCs) from healthy volunteers with curdlan under MDSC differentiation conditions. After 5 days, MHC-II⁻CD33ⁱⁿᵗCD15⁺ PMN-MDSCs were increased, while MHC-II⁺CD14⁺ antigen presenting cells and CD8⁺ T cells were decreased (Fig. 7l, m; Supplementary Fig. 6f). Addition of COX2 inhibitor partially inhibited the MDSC dif-ferentiation and immune cell suppression. Furthermore, the expres-sion of *CD33* and *CD11B* (*ITGAM*) in CRCs was directly induced by PGE₂ and *IL22RA2* was suppressed in a dose-dependent manner, suggesting that PGE₂ also promotes MDSC differentiation in humans (Fig. 7k). By examining the TCGA database, we found that the survival rate of CRC patients with high expression levels of *CLEC7A* and *PTGES* was lower than that of low expression group, and that of patients with high expression of *IL22RA2* was higher, although the relationship between *CLEC7A* expression and survival rate was not statistically significant (Supplementary Fig. 6g). These results suggest that the Dectin-1-PGE₂-IL-22BP axis is also involved in the regulation of human colorectal carcinogenesis.

## Discussion

In the present study, we showed that blocking Dectin-1 signaling sup-presses the development of colorectal tumors in both *ApcMin* familial adenomatous polyposis model and AOM-DSS-induced CRC model. Levels of PGE₂, which promotes MDSC expansion and tumor devel-opment, was reduced in *Clec7a⁻/⁻* mice due to decreased expression of PGE₂ synthesizing enzymes, including *Ptgs1*, *Ptgs2*, and *Ptges3*. In addition, the expression of IL-22BP, which can suppress the develop-ment of colorectal tumors by inhibiting IL-22, was significantly increased in *Clec7a⁻/⁻* mice. Furthermore, we found that CRC patients with low *CLEC7A* expression survive longer compared to those with high expression. A positive correlation was also observed between *CLEC7A* expression levels and colorectal cancer disease stages. This may be explained by increased infiltration of MDSCs and/or advanced differentiation of MDSCs, which express higher levels of Dectin-1 as indicated by the MDSC trajectory analysis. In CRC patients, *IL22RA2* expression was decreased and *PTGS2* expression was increased in tumors compared with normal tissues. These observations suggest that Dectin-1 plays important roles in the development of colorectal

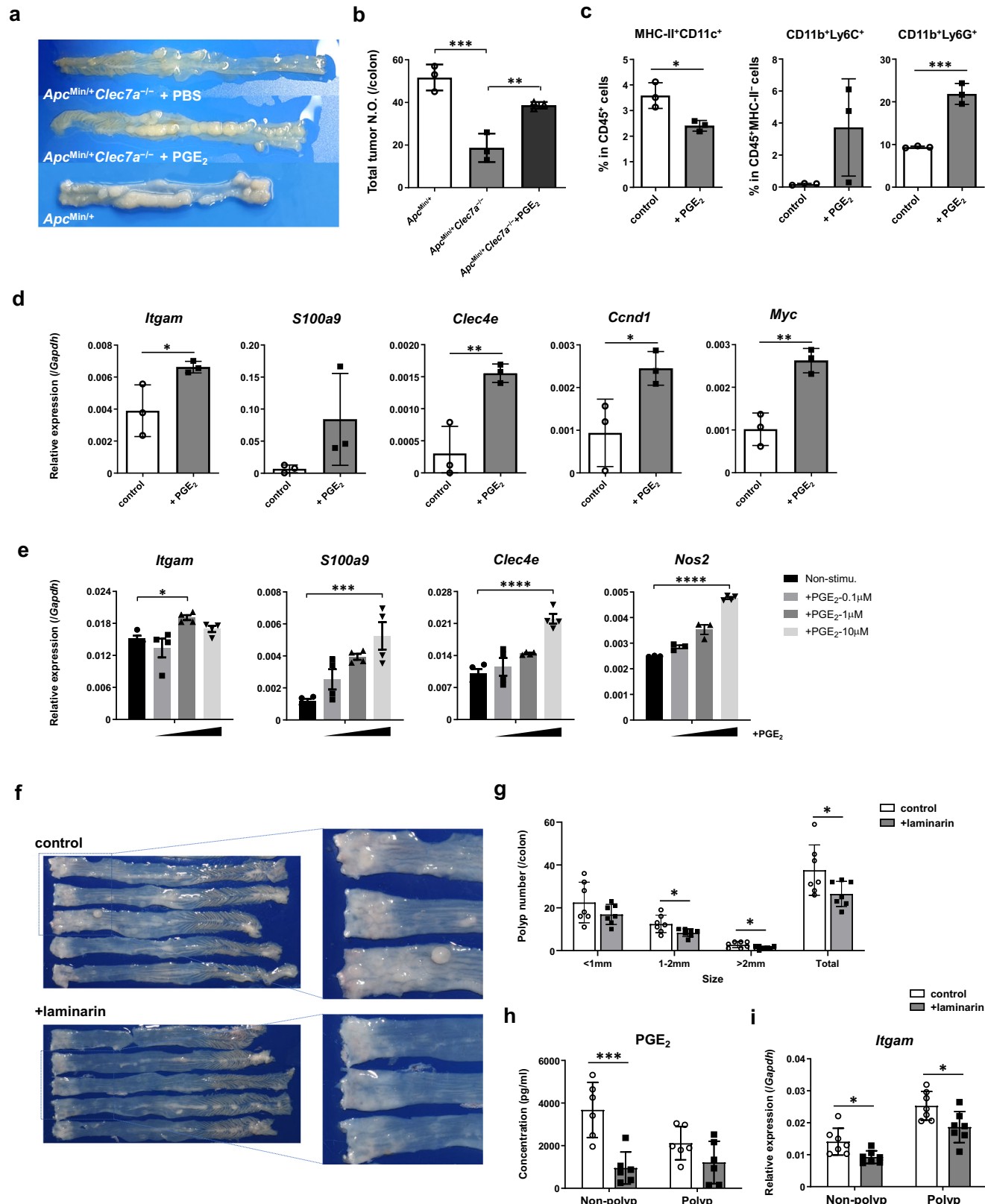

tumors in both mice and humans, through modification of PGE₂ levels and IL-22BP expression.

PGE₂, a well-known principal mediator of inflammation[47], plays a critical role in the development of colorectal tumors in AOM-DSS-induced tumor model as well as $Apc^{Min}$ mouse model by binding to the receptor EP2 to enhance the proliferation of tumor cells through

induction of genes involved in inflammation and cell growth such as *Tnf, Il6, Cxcl1*, and *Cox2*, by activating β-catenin[48,49]. Actually, intestinal polyp formation in *Apc*-deficient mice is suppressed by the *Ptgs2* null mutation, and COX-2 inhibitors such as aspirin and other NSAIDs can reduce the incidence and severity of colorectal tumors[50,51]. However, the regulatory mechanisms of PGE₂ production have not been

**Fig. 5 | Oral administration of laminarin inhibits colorectal tumorigenesis, while PGE$_2$ treatment enhances tumor development. a–d** $Apc^{Min/+}Clec7a^{-/-}$ mice (n = 3) at 8 weeks old were treated with 2% DSS for 1 week, followed by i.p. injection of PGE$_2$ (40 μg/mouse) or PBS every other day for 4 weeks. After sacrifice, colorectal polyp development was examined (**a, b**; ***P = 0.0006, **P = 0.0085). Populations of MHC-II$^+$CD11c$^+$ DCs and MHC-II$^-$CD11b$^+$Ly6C$^+$/Ly6G$^+$ cells in polyps were determined by flow cytometry (**c**; *P = 0.0193, ***P = 0.0009). Expression of indicated genes in colonic polyps was examined by RT-qPCR (**d**; in *Itgam* panel, *P = 0.0463; in *Clec4e* panel, **P = 0.0084; in *Ccnd1* panel, *P = 0.0414; in *Myc* panel, **P = 0.0043) (n = 3/group). **e** Tumor-infiltrating cells from $Apc^{Min/+}$ mice (n = 2) were collected and they were pooled. Then, cells were stimulated with PGE$_2$ for 20 h, and indicated gene expression was determined by RT-qPCR. Data are the means ± SD of 3–4 wells (*P = 0.0463, ***P = 0.0010, ****P < 0.0001). **f–i** C57BL/6J mice were administered with AOM and were treated with 5% laminarin-containing powder food for 10 days each from 3 days before the start of DSS-treatment for 3 cycles. On week 11 after AOM injection, mice were sacrificed and the gross histology of the colon (**f**) and colorectal polyp number of indicated size (**g**; 1–2 mm, *P = 0.030; >2 mm, *P = 0.0334; Total, *P = 0.0452) were examined. The concentration of PGE$_2$ in tissue homogenate was determined by ELISA (**h**, normalized by protein concentration of each lysate; ***P = 0.0006), and mRNA expression of *Itgam* (CD11b) was examined by RT-qPCR (**i**; Non-polyp, *P = 0.0174; Polyp, *P = 0.0211) (n = 7/group). Data in **a–d**, **f–i** are representatives of two, and in **e** are representatives of three independent experiments. Data in **b**, **e**, **g–i** are expressed as means ± SD. Data in **b**, **e** are analyzed using one-way ANOVA followed by Tukey's multiple-comparisons test, in **c**, **d**, **g** using unpaired two-tailed Student's t-test, and in **h**, **i** using two-way ANOVA followed by Tukey's multiple-comparisons test. Source data are provided in the Source Data file.

elucidated completely. In this study, we showed that deficiency of Dectin-1 impairs the expression of a series of enzymes involved in PGE$_2$ synthesis, including phospholipase A2 and Cox2, in MDSCs in both AOM-DSS- and $Apc^{Min}$-induced intestinal tumors. We also showed that Dectin-1 signaling directly induces the expression of these PGE$_2$-synthases in tumor-infiltrating MDSCs in vitro. In fact, intestinal levels of PGE$_2$ were decreased in $Clec7a^{-/-}$ mice, and exogenous PGE$_2$ administration increased colonic polyposis in $Clec7a^{-/-}$ mice. These results indicate that Dectin-1 signaling promotes intestinal tumorigenesis by enhancing PGE$_2$ synthesis in MDSCs. Consistent with our observations, previous reports suggested involvement of Dectin-1 in PGE$_2$ production by the stimulation with β-glucans from yeast[52], soluble egg antigen from *Schistosoma mansoni*[53], and a fungus *Paracoccidioides brasiliensis*[54]. However, the involvement of this activity in tumorigenesis has not been examined.

IL-22BP is a soluble inhibitory receptor for IL-22[55,56]. IL-22BP plays an important role in the regulation of tumorigenesis and its deficiency promotes colorectal tumor development[30]. In the present study, we showed that *Il22ra2* disruption enhanced polyp formation in $Clec7a^{-/-}$ mice, consistent with the previous reports. Although we have only compared tumor development between $Il22ra2^{-/-}$ and $Il22ra2^{+/+}$ mice on the $Clec7a^{-/-}$ background, these results suggest that enhanced expression of *Il22ra2* in $Clec7a^{-/-}$ mice protects these mice from intestinal tumor development. IL-22BP was produced by intestinal DCs but not by MDSCs. Interestingly, we found that *Il22ra2* expression in tumor-infiltrating cells was suppressed by PGE$_2$ through the suppression of the expression of RA-synthase genes and may also through the induction of IL-18, an inhibitor of *Il22ra* induction. As PGE$_2$ concentration was decreased in $Clec7a^{-/-}$ mouse intestine, this finding well explains why *Il22ra2* expression was increased in $Clec7a^{-/-}$ mice. Thus, these observations suggest that PGE$_2$ promotes intestinal tumorigenesis by suppressing *Il22ra2* expression as well as by the known tumor enhancing mechanisms of PGE$_2$[48,49]. In support for this notion, we showed that administrating $Clec7a^{-/-}$ mice with PGE$_2$ suppressed colonic *Il22ra2* expression and abrogated the tumor suppression in $Clec7a^{-/-}$ mice.

MDSC accumulation in tumors is considered to enhance cancer progression by promoting invasion of cancer cells, angiogenesis, and metastasis, and inhibiting antitumor immunity[57,58]. In this study, we found that both M-MDSC and PMN-MDSC population as well as MDSC-associated gene expression was significantly decreased in colonic polyps of $Clec7a^{-/-}$ mice. In line with our observations, Bhaskaran et al. also reported that Dectin-1-deficiency downregulates MDSC proportion in 4-Nitroquinoline 1-oxide-induced mouse tongue cancer[59]. We showed that proinflammatory cytokine production, such as IL-1β and IL-6, is reduced in $Clec7a^{-/-}$ mice. Rababi et al. also reported that Dectin-1 is critical for the infiltration of Ly6C$^{high}$ monocytes into the inflamed colon and promotes IL-1β secretion through leukotriene B4 production in DSS-colitis model[60]. These myeloid-cell-derived proinflammatory cytokines are important for MDSC differentiation and

their growth[28,29]. Trajectory analysis also suggest differentiation of macrophage and monocyte population into MDSCs. Furthermore, we showed that Dectin-1 signaling induces production of PGE$_2$ synthases in MDSCs, and PGE$_2$ enhances MDSC expansion in mice and in CRC patients. Indeed, we found that Dectin-1 signaling promoted expansion of MDSC population and enhanced their immune-suppressive activity during in vitro MDSC differentiation. Thus, these results indicate that Dectin-1 signaling facilitates MDSC differentiation and expansion by triggering the initiation of an auto-amplification loop, in which Dectin-1 induces inflammatory cytokines in monocytes and macrophages to promote MDSC differentiation and MDSCs produce PGE$_2$ to further expand themselves.

Accumulating evidence suggests that intestinal environment such as commensal microbiota affects development of colorectal tumors. Actually, development of obvious tumors was observed at 16 weeks after AOM-3DSS treatment under SPF conditions, while it took 36 weeks in GF mice in our experimental conditions. In this context, we previously showed that Dectin-1 signaling suppresses the growth of Treg-inducing bacteria such as *Lactobacillus* and *Clostridium* species by directly inducing antimicrobial protein calprotectin or by indirectly inducing antimicrobial peptides through induction of IL-17F[11–13]. Thus, blocking Dectin-1 signaling increases colonic Treg cell population in the intestine to suppress mouse colitis[11–13]. In the present study, however, we showed that AOM-3DSS-treated $Clec7a^{-/-}$ mice co-housed with WT mice or $Clec7a^{-/-}$ mice under GF conditions or ABX-treated $Apc^{Min/+}Clec7a^{-/-}$ mice still developed smaller number of polyps than WT mice, indicating that commensal microbiota are not responsible for the reduced polyp formation in $Clec7a^{-/-}$ mice. Consistent with this notion, Treg cells were not increased in GF $Clec7a^{-/-}$ mice, indicating that Treg cells are not involved in the suppression of polyp formation in $Clec7a^{-/-}$ mouse intestines.

Mouse chow used in our experiments contains over 10% of yeast extract, which contains abundant insoluble high MW β-glucans. Our daily diets also contain lots of high MW β-glucans, such as mushrooms and yeasts, as well as low MW β-glucan laminarin from seaweeds. Laminarin is water soluble and contains components with MW less than 5000. These low MW β-glucans are antagonists of Dectin-1 and compete with high MW β-glucans for the binding to Dectin-1[61]. Many reports suggest that high MW β-glucans can prevent the development of tumors[14,15]. In B16 melanoma model, for instance, intraperitoneal administration of high MW β-glucans facilitates antitumor NK cells and cytotoxic T cell responses against tumor cells by activating DCs through Dectin-1 signaling[22,23]. Tian et al. also reported that stimulation of monocyte-derive MDSCs isolated from mouse Lewis lung carcinoma with particulate β-glucans promotes the differentiation of these cells into tumor suppressive CD11c$^+$F4/80$^+$ cells and oral administration of these β-glucans promotes antitumor immunity[62]. However, it remains to be explained how orally administered particulate β-glucans are incorporated into blood stream and induce MDSC differentiation in solid tumors. In this study, we showed that oral administration of low

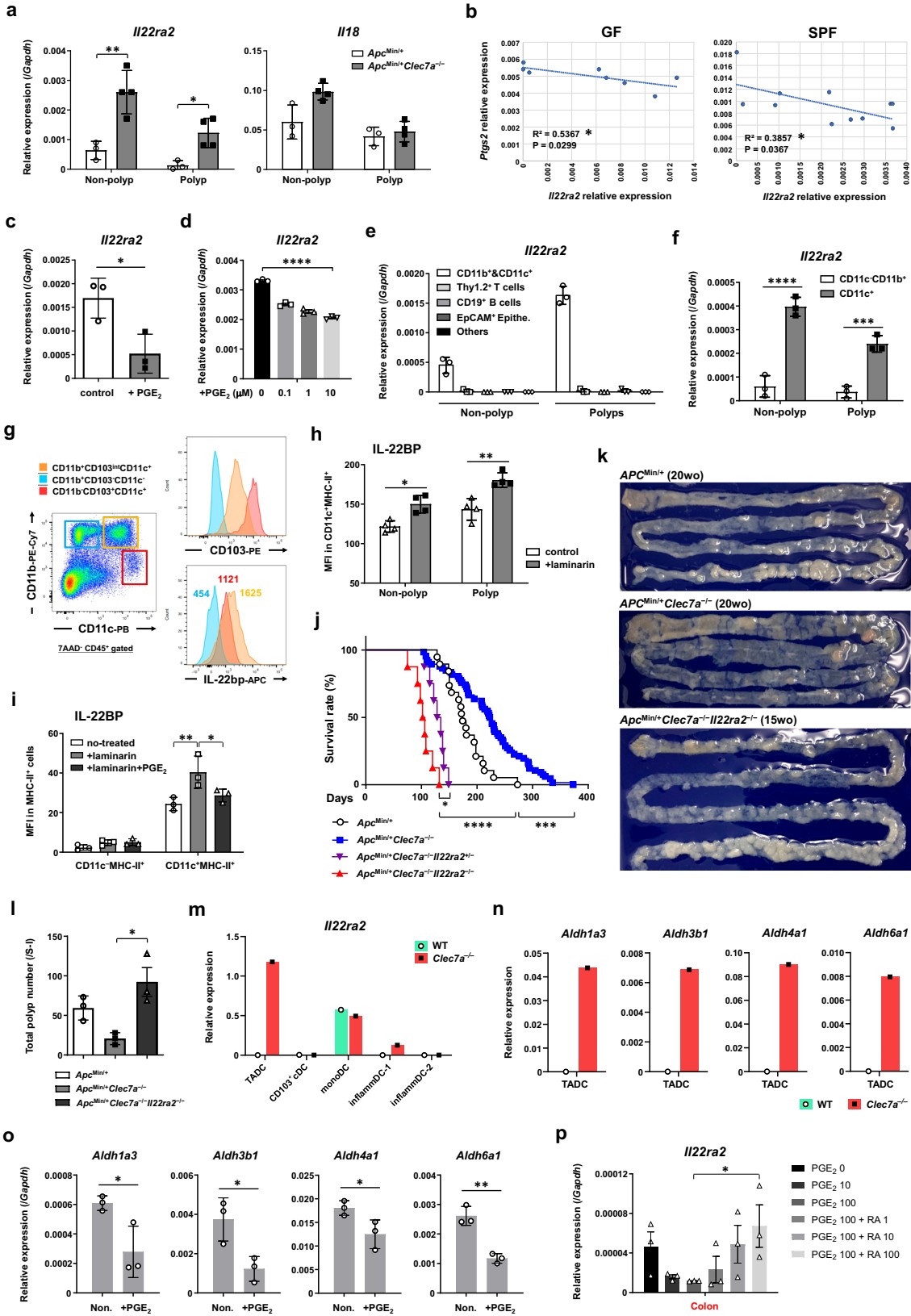

MW β-glucan laminarin, but not high MW β-glucans, suppresses colorectal polyposis. Interestingly, Hong et al. tracked β-glucans after oral uptake and found that high MW β-glucans are taken up by macrophages via Dectin-1 and are then degraded into smaller soluble β-glucan fragments[63]. Thus, it is possible that high MW β-glucans are degraded into low MW β-glucans in part after oral administration and

inhibit intestinal tumorigenesis by blocking Dectin-1 signaling rather than stimulating it.

A recent report identified an additional Dectin-1endogenous ligand, galectin-9 (*Lgals9*), in pancreatic cancer tissues[25]. In the present study, we showed that galectin-9 expression levels in colonic polyps and non-polyp tissues were similar between WT and *Clec7a*[−/−]

**Fig. 6 | *Il22ra2* expression is highly enhanced in *Clec7a*$^{-/-}$ mice and deficiency of *Il22ra2* aggravates intestinal tumor development in *Apc*$^{Min/+}$*Clec7a*$^{-/-}$ mice.**
**a** *Apc*$^{Min/+}$ and *Apc*$^{Min/+}$*Clec7a*$^{-/-}$ mice were sacrificed at 20 weeks old, and expression levels of *Il22ra2* and *Il18* in intestinal polyps and non-polyp tissues were determined by RT-qPCR (*Apc*$^{Min/+}$ *n* = 3; *Apc*$^{Min/+}$*Clec7a*$^{-/-}$ *n* = 4; \*\**P* = 0.0078, \**P* = 0.0148).
**b** Correlations between *Il22ra2* and *Ptgs2* expression levels in colorectal polyps from GF or SPF mice after treatment with AOM-3DSS are shown (*n* = 8/GF, *n* = 11/SPF group). **c** In the experiment described in Fig. 5a–d, expression of *Il22ra2* in colonic polyps was examined by RT-qPCR after PGE$_2$ in vivo treatment (*n* = 3/group, \**P* = 0.0261). **d** Tumor-infiltrating cells pooled from *Apc*$^{Min/+}$ mice were harvested and were stimulated with PGE$_2$ for 20 h, followed by the measurement of *Il22ra2* expression by RT-qPCR (*n* = 3 biologically independent samples/group; \*\*\*\**P* < 0.0001). **e** *Clec7a*$^{-/-}$ mice were treated with AOM-3DSS and were sacrificed after 12 weeks of AOM administration. Indicated types of cells in polyps and non-polyp tissues were purified with autoMACS and *Il22ra2* expression was examined by RT-qPCR (*n* = 3 biologically independent samples/group). **f** CD11c$^+$ cells and CD11c$^-$CD11b$^+$ cells were purified with autoMACS, and *Il22ra2* expression in these subsets was examined by RT-qPCR (*n* = 3 biologically independent samples/group; \*\*\*\**P* < 0.0001, \*\*\**P* = 0.0007). **g** Polyp-infiltrating cells were prepared from *Apc*$^{Min/+}$ mice, and intracellular IL-22BP expression was examined in three indicated myeloid-cell populations by flow-cytometry. Colored numbers in IL-22BP panel represent mean fluorescence intensity. **h** The experiment described in Fig. 5f was carried out, and 16 weeks after the AOM-DSS-treatment, mean fluorescence intensity (MFI) of IL-22BP expression in MHC-II$^+$CD11c$^+$ DCs in colorectal polyp and non-polyp tissues was determined by flow cytometry (*n* = 4/group; \**P* = 0.0121, \*\**P* = 0.0016). **i** WT mice were administrated with 5% laminarin in diet from 3 days before 1.5% DSS treatment and continued for 7 days until the end of DSS treatment.

During DSS-treatment, mice were also i.p. injected with PGE$_2$ every other day for 4 times. One day after the last PGE$_2$ injection, mice were sacrificed and MFI of IL-22BP expression in MHC-II$^+$CD11c$^+$ DCs was determined by flow cytometry (*n* = 3/group; \*\**P* = 0.0035, \**P* = 0.02323). **j** Survival rates of indicated mice are shown (*Apc*$^{Min/+}$ *n* = 19; *Apc*$^{Min/+}$*Clec7a*$^{-/-}$ *n* = 66; *Apc*$^{Min/+}$*Clec7a*$^{-/-}$*Il22ra2*$^{+/-}$ *n* = 8; *Apc*$^{Min/+}$*Clec7a*$^{-/-}$*Il22ra2*$^{-/-}$ *n* = 8; \*\*\*\**P* < 0.0001, \*\*\**P* = 0.0003, \**P* = 0.0179). **k** Representative pictures of the small intestine of *Apc*$^{Min/+}$ and *Apc*$^{Min/+}$*Clec7a*$^{-/-}$ mouse of 20 weeks old and *Apc*$^{Min/+}$*Clec7a*$^{-/-}$*Il22ra2*$^{-/-}$ mouse of 15 weeks old are shown. **l** Total polyp number in the intestine of *Apc*$^{Min/+}$ (*n* = 3), *Apc*$^{Min/+}$*Clec7a*$^{-/-}$ (*n* = 3), and *Apc*$^{Min/+}$*Clec7a*$^{-/-}$*Il22ra2*$^{-/-}$ (*n* = 3) mice are shown (\**P* = 0.0125). **m** In the scRNA-seq analysis described in Fig. 3e–h, the expression levels of *Il22ra2* were determined in each colorectal polyp-infiltrating DC subset. **n** As in **m**, RA synthase expression levels in TADC from WT and *Clec7a*$^{-/-}$ mice were determined by scRNA-seq. **o** CD11b$^+$ and CD11c$^+$ cells isolated from WT mouse colon were treated with 10 μM PGE$_2$ for 20 h, and expression levels of genes encoding RA synthases were examined by RT-qPCR (*n* = 3 biologically independent samples/group; *Aldh1a3*, \**P* = 0.0340; *Aldh3b1*, \**P* = 0.0261; *Aldh4a1*, \**P* = 0.0472; *Aldh6a1*, \*\**P* = 0.0023). **p** CD11b$^+$ and CD11c$^+$ cells isolated from WT mouse colon were treated with indicated doses of PGE$_2$ (μM) and RA (μM) for 20 h, and expression level of *Il22ra2* was examined by RT-qPCR (*n* = 3 biologically independent samples/group; \**P* = 0.030). Data in **a**, **c–l**, **o**, **p** are representatives of two independent experiments, and in **a**, **c**, **d–f**, **h**, **i**, **l**, **o**, **p** are expressed as means ± SD. Data in **a**, **f**, **h**, **i** are analyzed using two-way ANOVA followed by Tukey's multiple-comparisons test, in **b** using Pearson's correlation coefficient (*r*) test, in **c**, **o** using two-tailed Student's *t*-test, in **d**, **l**, **p** using one-way ANOVA followed by Tukey's multiple-comparisons test and in **j** using Mantel-Cox log-rank test. Source data are provided in the Source Data file.

---

mice (Supplementary Fig. 3m), suggesting that β-glucans in food rather than the endogenous ligand is involved in the Dectin-1-dependent regulation of colorectal tumor development.

Overall, in this report, we have demonstrated a Dectin-1-mediated colorectal tumor-promoting mechanism, in which β-glucan-induced Dectin-1 signaling promotes PGE$_2$ production and suppresses anti-tumorigenic IL-22BP expression. Our findings suggest possible influence of daily foods on the development of CRC, and suggest Dectin-1 as a possible target for the prevention and treatment of CRC.

## Methods
### Human CRC sample collection
Before the collection of human samples from CRC patients, written informed consent was obtained from each patient, and all experiments were carried out following the institutional ethical regulations and guidelines under the protocols approved by the Committee for Clinical Investigation of the First Affiliated Hospital, Sun Yat-sen University (approval number: IIT-2021-654). For tissue-infiltrating cell purification, tumor specimens and/or normal tissue around tumor were cut out and were transferred into ice-cold PBS and placed on ice until use. For the mRNA analysis, we collected 47 tumor specimens and 43 non-tumor specimens from 70 CRC patients, in which 23 pairs of tumor and non-tumor tissues from the same individuals were enrolled. In these patients, we collected the clinical information such as pathological grading, TNM stage, operation and follow-up visiting time of 36 patients, most of whom were followed up around 10 years. Except 12 samples harvested recently, most of the samples were frozen in −80 °C until use for RNA extraction. Details of patient clinical information including TMN stage, pathological grading, and tumor size are available in Supplementary Table 1.

### Mice
*Clec7a*$^{-/-}$ mice were generated as described before[5] and used after backcrossing for 9 generations to C57BL/6 J. *Apc*$^{Min/+}$ mice, kindly provided by Dr. Ryo Abe, Tokyo University of Science, were crossed with *Clec7a*$^{-/-}$ mice to generate *Apc*$^{Min/+}$*Clec7a*$^{-/-}$ mice. C57BL/6J mice,

which were originally purchased from Sankyo Lab Service (Saitama, Japan) for *Clec7a*$^{-/-}$ mouse backcrossing, and their offspring were maintained in the same animal room as that *Clec7a*$^{-/-}$ mice were housed and used as the control. GF WT mice were purchased from Sankyo Lab Service. GF *Clec7a*$^{-/-}$ mice were generated by taking out sterile babies from the uterus of a SPF full-term pregnant female by cesarean section and transferring these babies in the care of 2 to 3 GF WT mothers that were nursing their recently delivered litters in a GF isolator. Age- and sex-matched WT mice were separately housed or co-housed with *Clec7a*$^{-/-}$ mice after weaning at 4 weeks old. All mice were kept under specific pathogen-free conditions with γ-ray sterilized normal diet (Funakoshi Kagaku, Chiba, Japan), acidified (0.002 N HCl, pH 2.5) tap water, and autoclaved wooden chip bed in environmentally controlled clean rooms at the experimental animal facilities of the Center for Animal Disease Models, Research Institute for Biomedical Sciences, Tokyo University of Science, and of Zhongshan School of Medicine, Sun Yat-sen University. All animal experiments were carried out following the institutional ethical regulations and guidelines and with protocols approved by the Institutional Animal Care and Use Committee of the Tokyo University of Science, and by the Experimental Animal Manage and Use Committee of Sun Yat-sen University (approval number 2020000113, 2021001577). Since both male and female *Clec7a*$^{-/-}$ mice exhibited the same phenotypes in all disease models in the present study, mice of both genders were used.

### Generation of *Apc*$^{Min/+}$*Clec7a*$^{-/-}$*Il22ra2*$^{-/-}$ mice
*Il22ra2*$^{-/-}$ mice on *Apc*$^{Min/+}$*Clec7a*$^{-/-}$ background were generated by CRISPR-Cpf1 method. Briefly, the targeting sequence on the mouse *Il22ra2* exon 3 with the PAM sequence was determined as (TTTCATTGGTCAGGTCACAGAAGAGCGC), and the guide RNA (crRNA) was obtained from Integrated DNA Technologies (IDT, Coralville, IA, USA). Then 1.2 mM crRNA was mixed with 0.6 mM CRISPR-Cpf1 nuclease (Cat# 1081068, IDT) and was transferred into *Apc*$^{Min/+}$*Clec7a*$^{-/-}$ zygotes by micro-injection. On the next morning, normal 2-cell stage embryos were transferred into the oviducts of pseudo-pregnant female mice. The genotypes of mutant babies were determined by sequencing the RT-PCR product with the primers

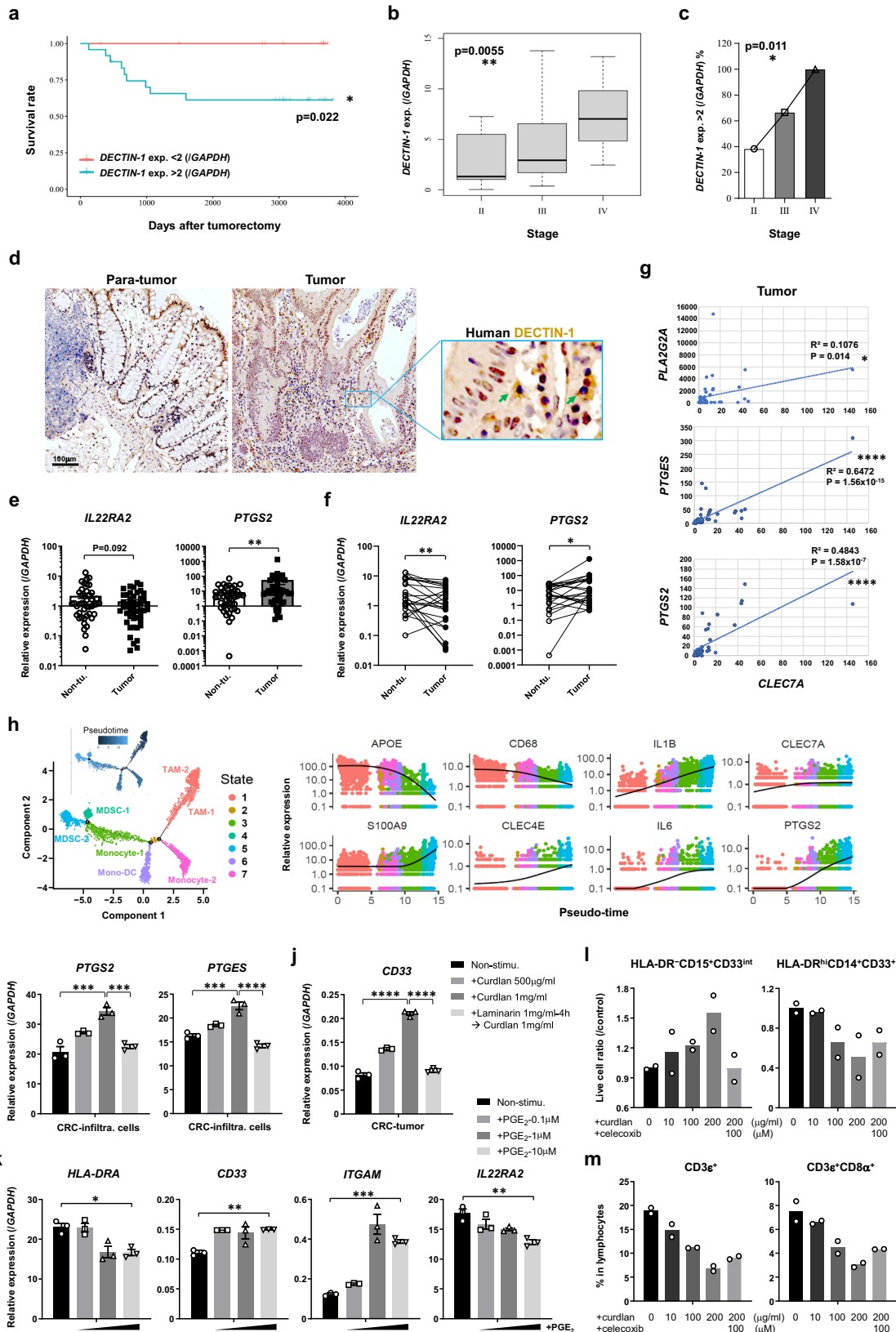

ATTTGTGTGTTTTCCCATCCTC (Forward) and TCAAAGGCAGAAG AGAAAAAGG (Reverse). Three mutant mice, all of which had 19 bp deletion that causes a frame-shift and premature termination of IL-22BP, were identified, and one of them was used to generate homozygous *Apc^Min/+Clec7a^−/−Il22ra2*-deficient mice.

## Induction of intestinal polyps in *Apc^Min* mice

*Apc^Min/+* or *Apc^Min/+Clec7a^−/−* mice were normally kept under SPF conditions for 19–23 weeks after birth until polyps in small and large intestines develop spontaneously. For the induction of colorectal tumors, same mice at 8–10 weeks old were treated with 1% DSS in drinking

**Fig. 7 | The expressions of PGE$_2$-synthesizing enzymes and *IL22RA2* are modified in CRC patients. a**–**c** Specimens from 36 CRC patients were collected and the post-surgery survival rates of the patients with *CLEC7A* mRNA expression <2 (*n* = 12) or >2 (2–20) (*n* = 24) (normalized by *GAPDH*) in tumor tissues are shown (**a**). The correlation between *CLEC7A* expression levels and tumor TNM stage (**b**, *n* = 13 in Stage II, *n* = 12 in Stage III, *n* = 8 in Stage IV), and the correlation between the proportion of patients with *CLEC7A* expression >2 in each stage (**c**, *n* = 13 in Stage II, *n* = 12 in Stage III, n = 8 in Stage IV) are shown. Tumor TNM stage was determined according to the individual clinical information. In boxplot (**b**), the central horizontal line shows the median; the upper and lower bounds of box show third and first quartile; the upper and lower horizontal lines show the maxima and minima of the data. **d** Immunohistochemical staining of human Dectin-1 in tumor and non-tumor colon tissues in CRC patients. Green arrows show Dectin-1$^+$ cells. Representative photos from two independent experiments containing nine samples of eight patients are shown. **e** Specimens from total 68 CRC patients were collected including 23 pairs of tumors and non-tumor specimens from the same individuals. *IL22RA2* and *PTGS2* expression in tumors and non-tumor tissues were examined by RT-qPCR (*n* = 47 in tumor group, *n* = 44 in non-tumor group; **P = 0.0084). **f** Twenty-three pairs of tumor and non-tumor specimens from the same individual were collected, and *IL22RA2* and *PTGS2* expression were examined (*n* = 23/group; **P = 0.0067, *P = 0.0179). **g** Correlations of the expression (normalized by *GAPDH*) between PGE$_2$ synthesizing enzymes and *CLEC7A* in CRC tissues (*n* = 47/group). **h** Pseudotime/trajectory analysis was carried out by using a scRNA-seq dataset (GSE178318) of CRC-infiltrating myeloid cells, in which fully differentiated DCs were excluded. Cell states and their annotations (left panel), and kinetics of relative

expression levels of MDSC and macrophage-associated gene markers along the MDSC 'differentiation' pseudo-time (right panels) are shown. TAM: tumor-associated macrophage; Mono-DC: monocyte-derived DC. **i**, **j** Tumor and non-tumor tissue-infiltrating cells were collected from CRC patients and were stimulated with curdlan for 20 h. In the case of laminarin treatment, cells were first treated with laminarin for 4 h, then curdlan was added to the culture. Expression of PGE$_2$ synthesizing enzymes (**i**; *PTGS2*, 1st ***P = 0.0001, 2nd ***P = 0.0004; *PTGES*, ***P = 0.0001, ****P < 0.0001) and CD33 (**j**; ****P < 0.0001) was determined by RT-qPCR. Data are the means ± SD of 3 wells. **k** Tumors from CRC patients were collected and after mincing, small pieces were cultured in the presence of PGE$_2$ for 20 h. The expression of indicated genes was determined by RT-qPCR. Data are the means ± SD of 3 wells (*HLA-DRA*, *P = 0.0122; *CD33*, **P = 0.0033; *ITGAM*, ***P = 0.0004; *IL22RA2*, **P = 0.0018). **l** Human PBMCs were harvested from healthy volunteers and were treated with curdlan and celecoxib at the indicated doses in the presence of rGM-CSF + IL-6. After the culture for 5 days, indicated cell number was determined by flow cytometry and was normalized with the average cell number in control (curdlan 0) group. **m** In the experiment described in **l**, after the culture, proportions of whole T cells and CD8$^+$ T cells were determined by flow cytometry. Data in **i**–**k** are representatives of three, and in **l**, **m** are representatives of two independent experiments and are expressed as means ± SD. Data in **a** are analyzed using Mantel-Cox log-rank test, in **b**, **c** using Spearman rank correlation test, in **e** using two-tailed Student's *t*-test, in **f** using Wilcoxon matched-pairs signed rank test, in **g** using Pearson's correlation coefficient (*r*) test, and in **i**, **j**, **k** using one-way ANOVA followed by Tukey's multiple-comparisons test. Source data are provided in the Source Data file.

water for 1 week, and they were maintained for another 4–5 weeks before sacrifice.

## Chemical induction of colorectal tumor

Six to 7-weeks-old mice were administrated intraperitoneally (i.p.) with 10 μg per gram body weight of AOM (Cat# 011-20171, Wako, Osaka, Japan). One week after the administration, these mice were treated with 1% DSS (36–50 kDa; Cat# 216011090, MP Biomedicals, Illkirch, France) in their drinking water for 7 days followed by 14 days of normal water. After 3 cycles of DSS/normal water-treatment, these mice were further maintained for additional 2–6 weeks before sacrifice.

## Cell preparation

Colonic tumor/ non-tumor tissue-infiltrating cells were prepared as follows. Briefly, tissue pieces were cut into 1 mm slices, then they were shaken for 40 min in Hanks' Balanced Salt Solution (HBSS) containing 3 mM EDTA at 37 °C. Intra-epithelial lymphocytes and colonic epithelial cells in the culture supernatant were discarded, and the gut slices were washed twice with HBSS. Then, they were shaken at 37 °C for 120 min in RPMI containing 10% FBS and 1% streptomycin+penicillin, 200 U/ml collagenase (Cat# C2139; Sigma-Aldrich), and 5 U/ml Dnase 1 (Sigma-Algrich). After the incubation, the samples were vortexed for 10 s, and single-cell suspension was harvested after sterile gauze-filtration. To prepare lymphocytes, tissue-infiltrating cells were further purified on a 45%/66.6% discontinuous Percoll (Cat# 17-0891-01, Cytiva Life Sciences) gradient at 2200 rpm for 20 min.

## Cell and tissue in vitro culture

Tissue-infiltrating cells (2 × 10$^5$) from mouse/ human colorectal tumors or non-tumor tissues, or mouse/ human colorectal tumor pieces or non-tumor tissue pieces (3 × 3 mm), were cultured in a 0.2 ml RPMI containing 10% FBS and 1% streptomycin plus penicillin at 37 °C under 5% CO$_2$ in a 96-well flat-bottomed plate (Falcon, Becton Dickinson). To induce MDSC differentiation in vitro, mouse bone marrow cells or human PBMCs were harvested from thigh and shin bones of mice or peripheral blood of healthy volunteers, respectively, and were cultured with recombinant mouse GM-CSF and IL-6 (Cat# 315-03, 216-16, Peprotech Inc., Rocky Hill, NJ, USA) in 48-well flat-bottomed plate (Falcon) for 120 h. For T cell activation, CD4$^+$ or total T cells were isolated from the spleen, and were cultured with the stimulation of

anti-CD3/CD28 Dynabeads (Cat# 11456D, Thermo Fisher Scientific) for 3-4 days. In some experiment, anti-mouse IL-18 antibody (Cat# 210-401-323, Rockland Immunochemicals, Inc, Philadelphia, PA, USA) was added to the culture with or without PGE$_2$ stimulation. In other experiments, inhibitors of NF-κB (IκBa: BAY11-7082, Cat# SF0011, Beyotime, China) or SYK (R406, Cat# SC1022, Beyotime) was added to the culture after stimulation with curdlan. After the culture, cells or tissues were collected and mRNA expression was measured by quantitative reverse transcription PCR (RT-qPCR) or cell population was examined by flow cytometry.

## Purification of epithelial cells and immune cells from lamina propria

Cells from mouse whole intestines or from polyps were labeled with biotin-conjugated anti-mouse CD326/EpCAM (G8.8) (Cat# 118203, Biolegend), followed by microbeads-conjugated anti-biotin CD326$^+$ epithelial cells were positively sorted by using autoMACS (Miltenyi Biotec, Bergisch Gladbach, Germany). The CD326 negative fraction was then sorted with anti-mouse CD45 (Cat# 130-052-301) microbeads for leukocyte purification, or was sorted with anti-mouse CD11b (Cat# 130-049-601) and CD11c (Cat# 130-125-835) microbeads for myeloid-derived cells. Anti-Thy1.2 (Cat# 130-121-278) microbeads were used for T cells and anti-CD19 (Cat# 130-121-301) microbeads were used for B cell sorting. All the antibody-conjugated microbeads were obtained from Miltenyi Biotec. Colonic polyp-infiltrating myeloid cells for scRNA-seq analysis were stained with anti-mouse CD11b and CD11c antibodies for flow cytometry and were sorted by FACS Aria Fusion (BD Biosciences).

## Flow-cytometry

Antibodies against mouse CD4 (GK1.5, Cat# 100411), CD8α (53-6.7, Cat# 100707), TCRγδ (GL3, Cat# 118105), CD45 (30-F11, Cat# 103115, 103113), CD19 (6D5, Cat# 115505), CD3ε (KT3.1.1, Cat# 155607), Ly6C (HK1.4, Cat# 128007, 128025), Ly6G (1A8, Cat# 127605, 127633), CD11b (M1/70, Cat# 101215), CD11c (N418, Cat# 117321), I-A/I-E (M5/114.15.2, Cat# 107613, 107627), CD103 (2E7, Cat# 121405), IgG1 (RMG1-1, Cat# 406607), NOS2 (W16030C, Cat# 696805), IFN-γ (XMG1.2, Cat# 505805), IL-17 (TC11-18H10.1, Cat# 506917), Foxp3 (MF-14, Cat# 126403), CD326 (G8.8, Cat# 118203), Thy1.2 (30-H12, Cat# 105304) and anti-human CD33 (HIM3-4, Cat# 303303), CD14 (M5E2, Cat# 301807),

CD15 (W6D3, Cat# 323005), HLA-DR (L243, Cat# 307645), CD3ε (HIT3a, Cat# 300305) and CD8α (SK1, Cat# 344721) were purchased from Biolegend (San Diego, U.S.A.). Antibody against mouse Dectin-1 (2A11, Cat# ab21646) was purchased from Abcam (Cambridge, UK). Anti-mouse IL-22BP polyclonal antibody was purchased from R&D systems (Cat# IC2376A, Biotechne, Minnesota, U.S.A.). 7-AAD (7-amino-actinomycin D) staining solution were obtained from eBioscience (Cat# 00-6993-50, San Diego, CA, USA). The anti-CD16/CD32 (2.4G2, anti-FcγRIII/II) mAb was obtained from Tonbo Biosciences (Cat# 70-0161-U500, San Diego). All antibodies were used at a 1:250 dilution. Cells isolated from mouse intestine were washed twice with FACS Hanks buffer (HBSS containing 2% FCS) and then treated with 2.4G2 to block non-specific FcR binding. Cells were then surface-stained with mAbs for 25-30 min. For the analysis of cytokine expression by FACS, cells were first stimulated with 50 ng/ml PMA (Cat# P8139, Sigma Aldrich) + 500 ng/ml ionomycin (Cat# 407952, Sigma Aldrich) and 1 mM monensin (Cat# 475897, Sigma Aldrich) for 4 h. After the stimulation, cells were washed twice with FACS Hanks buffer and then treated with 2.4G2 to block FcR binding. Cells were then stained with mAbs against cell-surface markers for 30 min and were fixed and permeabilized with Cytofix/Cytoperm solution (Cat# 554722, BD Biosciences) for 20 min. After washing with washing buffer, cells were incubated with anti-mouse IFN-γ and IL-17 mAbs for another 30 min. The fluorescence was detected by a Canto II or FACSCalibur flow cytometer (BD Biosciences) and FlowJo v10.0 & v10.1 software (BD Biosciences) was used for the analysis.

## Quantitative reverse transcription PCR (RT-qPCR)

Total RNA was extracted with a Mammalian Total RNA Miniprep kit (Cat# RTN350-1KT, Sigma Aldrich). RNA was denatured in the presence of an oligo dT primer and then reverse transcribed with a High Capacity cDNA Reverse Transcription kit (Cat# 4368814, Applied Biosystems, San Francisco, U.S.A.). qPCR was performed with a SYBR Green qPCR kit (Cat# 639676, Takara Bio., Shika, Japan) and C1000 Thermal Cycler system (Bio-Rad, Hercules, USA) with the sets of primers described in Supplementary Table 2, and relative mRNA expression levels were calculated with the comparative cycle threshold (Ct value) method and were normalized with *Gapdh* mRNA expression level.

## Bulk RNA sequencing and analysis

Total RNA was isolated from colorectal tumor or non-tumor tissues using Sigma Aldrich total RNA-prep kit. RNA was prepared using Contech SmartSeq2 library prep kits and sequenced with NovaSeq6000. Data were processed with Galaxy (usegalaxy.org). Quality filtering and adapter trimming of the resulted reads were performed by Trimmomatic[64], aligned to GRCm38 (mm10) whole-genome data by HISAT2[65], assembled to transcripts by StringTie[66]. Data were normalized by total lead count correction and transformed to log2 (TPM + 1) for further analysis. Genes with log2 (fold-change) >1 or <−1 were considered as upregulated and downregulated genes, respectively. Gene with log2 (fold-change) >1 or <−1 was input for KEGG pathway analysis with Benjamini and Hochberg method. *P*-value <0.05 and *q*-value <0.1 are considered as significant. Heatmaps were plotted using Tbtools software[67], and the *z*-score was calculated by transcripts per million (TPM), as (TPM-average of TPM)/Standard deviation.

## PGE₂ and IL-1β concentration measurement

Polyps and non-polyp tissues from mouse colon were harvested and were put into 2 ml tubes containing 500 µl lysis buffer for protein [RIPA buffer: 50 mM Tris-HCl (pH 8.0) + 100 mM NaCl + 0.1% Triton X-100] and 50 µl protease inhibitor cocktail (Takara Bio, Shika, Japan). Tissue was homogenized with beads and Micro Smasher (MS-100R, TOMY). After centrifugation, supernatant of tissue homogenate was collected and PGE₂ and IL-1β concentration was measured by ELISA Development Kit for PGE₂ (Cat# KGE004B, R&D systems) and ELISA

MAX™ Standard Set for IL-1β (Cat# 432601, Biolegend). The concentration of PGE₂ was normalized to total protein concentration of each individual.

## Antibiotics treatment

$Apc^{Min/+}$ mice were pretreated with an antibiotics (ABX) mixture containing 500 mg/l vancomycin (Cat# V820413), 1 g/l ampicillin (Cat# A6265), 1 g/l metronidazole (Cat# M813526) and 1 g/l neomycin (Cat# N814740) (VAMN, Macklin Inc., Shanghai, China) for 3 weeks with addition of 75 g/l sucrosein drinking water, and then were co-administrated with DSS for 1 week, and were continuously treated with ABX for further 4 weeks. In some experiment, ABX cocktail containing 0.4 g/l kanamycin (Cat# K812216), 0.035 g/l gentamycin (Cat# G6064), 0.0570 g/l colistin (Cat# C805491), 0.215 g/l metronidazole, 0.045 g/l vancomycin HCl and 0.025 g/l erythromycin (Cat# E808820)[68] (6 mixture, Macklin) were used to treat mice.

## β-glucan treatment

Agonistic Dectin-1 ligand curdlan was purchased from Wako Pure Chemical Corp. and NaClO-oxidized *Candida albicans* (OXCA) was prepared by Prof. Ohno at Tokyo Pharmaceutical University[69]. Colonic polyp or non-polyp-infiltrating cells were treated with curdlan or OXCA for 16 or 20 h, and the expression of cytokine and PGE₂ synthesizing enzyme was examined. Dectin-1 antagonist laminarin (Cat# L9634, 1 mg/ml, Sigma-Aldrich, St. Louis, MO, USA) treatment was carried out for 4 h before curdlan treatment. For laminarin treatment in vivo, mice were firstly i.p. administrated with AOM and then were treated with 5% laminarin (Cat# L0088, Tokyo Chemistry Industry, Tokyo, Japan) mixed in powder food from 3 days before the start of DSS-treatment for 10 days, and this laminarin-DSS treatment were carried out for 3 cycles.

## PGE₂ treatment

PGE₂ was purchased from TopScience Co. Ltd, Shanghai, China (Cat# T5014). For in vitro treatment, colonic polyp or non-polyp-infiltrating cells were treated with PGE₂ (0.1–10 µM) for 20 h, and the expression of myeloid-cell marker genes was examined. For in vivo treatment, mice at 8 weeks old were firstly treated with 2% DSS for 1 week, then were i.p. injected with 40 µg/mouse PGE₂ or PBS every other day for 4 weeks.

## Single-cell RNA-seq library construction and sequencing

After colonic polyp-infiltrating myeloid-derived cells were harvested, cell count and viability was estimated using fluorescence Cell Analyzer (Countstar® Rigel S2) with AO/PI reagent. Fresh cells were resuspended at $1 × 10^6$ cells/ml in PBS containing 0.04% bovine serum albumin. Single-cell RNA-Seq libraries were prepared using SeekOne® MM Single Cell 3′ library preparation kit (SeekGene, Cat# SO01V3.1). Briefly, appropriate number of cells were loaded onto the flow channel of SeekOne® MM chip which had 170,000 microwells and allowed to settle in microwells by gravity. After removing the unsettled cells, sufficient Cell Barcoded Magnetic Beads (CBBs) were pipetted into flow channel and were also allowed to settle in microwells with the help of a magnetic field. Next, excess CBBs were rinsed out and cells in MM chip were lysed to release RNA, which was captured by the CBB in the same microwell. Then all CBBs were collected and reverse transcription were performed at 37 °C for 30 min to label cDNA with cell barcodes on the beads. Further Exonuclease I treatment was performed to remove unused primer on CBBs. Subsequently, barcoded cDNA on the CBBs was hybridized with random primer, which had read 2 SeqPrimer sequence on the 5′ end and could extend to form the second strand DNA with cell barcode on the 3′ end. The resulting second strand DNA were denatured off the CBBs, purified and amplified in PCR reaction. The amplified cDNA product was then cleaned to remove unwanted fragment and added to full length sequencing adapter and sample

index by indexed PCR. The indexed sequencing libraries were cleanup with SPRI beads, quantified by quantitative PCR (KAPA Biosystems KK4824) and then sequenced on illumina NovaSeq 6000 with PE150 read length.

## Preprocessing of single-cell RNA-seq data analysis

Single-cell sequencing data obtained from the present study or pre-processed data of AOM-3DSS-treated C57BL/6 mice (GSE196054) and CRC patients (GSE178318[45] and GSE146771[70]) from NCBI GEO database were converted into a Seurat object by the R package (version 4.1.3) using the software Rstudio Desktop v2022.02.3 (Posit Software, PBC) and then were filtered to remove cells with a number of unique gene counts over 4000 or less than 200. Cells with more than 5% of mito-chondrial gene counts were also filtered. After unwanted cells were removed from the dataset, a global-scaling normalization method "LogNormalize" was carried out to normalize the feature expression measurements for each cell by the total expression, multiply this by a scale factor of 10,000, and log-transform the result. Over 95% of high-quality cells with total more than 25,000 protein-coding genes were remained after the filtering and normalization for the downstream processing.

## Identification of cell types and gene expression by dimensional reduction

After the normalization, subsets of feature genes were calculated that exhibit high cell-to-cell variation in the dataset. Next, a linear trans-formation (scaling data) and following linear dimensional reduction were performed. Then, cells were clustered by feature gene profiles. The annotation of each cell cluster was confirmed by the expression of canonical marker genes such as: T cells (*CD3D, TRAC*), B cells (*CD19, MS4A1*), plasma cells (*JCHAIN, IGHA1, CD79A*), macrophages (*CD68, APOE, MERTK*), MDSCs (*S100A8, S100A9, CLEC4E, CD33*), DCs (*FCER1A, CST3, HLA-DQB1*), mast cells (*TPSAB1, MS4A2*), fibroblasts (*COL1A1, COL3A1*), epithelial cell (*EPCAM, KRT8, KRT18*) and tumor cells (*TK1, STMN1, TUBB*).

## Analysis for pseudo-temporal trajectory of MDSC differentiation

Tumor-infiltrating myeloid cells, in which $PTPRC^{lo}$ non-leukocytes, $CD3D^{+}TRAC^{+}$ T cells and $CST3^{hi}HLA-DRA^{hi}$ DCs were excluded by Seurat in R, were extracted from scRNA-seq dataset of CRC (GSE196054). Pseudotime/ trajectory analysis for MDSC differentiation was then performed by using the R package Monocle and Monocle3. Simulation of MDSC differentiation that orders myeloid cells along a pseudo-temporal trajectory was output through unsupervised Monocle pro-cessing. Expression of *S100A9* and *CLEC4E* was used as the differentiation marker for MDSCs and that of *CD68* and *APOE* was used for macrophages.

## Histopathological analysis

Colon and rectum with polyps were removed and fixed with 10% neutral buffered formalin and embedded in paraffin. After being cut into 5 μm slices, tissue sections were stained with haematoxylin and eosin. Histopathological score was evaluated according to a previous report[71]: for number of lesions (*L*), 1 = 0–2, 2 = 3–5, 3 = 6–8, 4 = 9–; for crypts (*C*): 0 = normal, 1 = goblet cell depletion, 2 = branching, 3 = complex budding; for Epithelium (*E*): 0 = normal, 1 = hyperplasia, 2 = low-grade dysplasia, 3 = high-grade dysplasia; for Submucosal invasion (*S*): 0 = absent, 1 = present. The total histological score was calculated as $L + C + E + S$.

## Immunohistochemistry (IHC)

For immunohistochemistry, tumor and non-tumor tissues from CRC patients after surgery were rinsed with PBS and fixed with 10% for-malin. After fixation, tissues were dehydrated in ethanol overnight for paraffin embedding. After paraffin embedding, the paraffin block was cut into 5 μm slices. After deparaffinization, the tissue slices were rehydrated in 3,3′-diaminobenzidine solution and were treated with $H_2O_2$ to close the endogenous peroxidase. Then the slices were incubated in retrieval solution at 60 °C overnight for heat-induced epitope retrieval, followed by treatment with 1% BSA for 2 h to block non-specific antibody binding. Then, the slices were incubated with 1 μg/ml of primary antibody against human Dectin-1 (Cat# ab140039, Rabbit polyclonal to human and Rat Dectin-1, Abcam Inc, Cambridge, UK) overnight at 4 °C, followed by HRP-conjugated anti-rabbit IgG (Cat# GB23303, Servicebio co. Wuhan, China) for 15 min at room temperature. Tissue slices were then treated with DAB substrate solution for 1-3 min, and with hematoxylin counterstain for additional 10 seconds at room temperature. Photos of IHC were taken by Leica DM6FS with imaging scanning system (Leica Camera AG, Wetzlar, Germany).

## Western blot analysis

For the detection of IL-22BP protein levels in mouse intestinal polyps tissues, a piece of tissue (200 mg) was firstly lysed with 0.1 ml RIPA buffer (25 mM Tris-HCl, pH 7.5, 50 mM NaCl, 0.5% NP-40, 0.1% SDS, 0.5% DOC, 1 mM DTT, 20 mM NEM supplied with fresh Protease inhibitors). A syringe with sonicator was used to mechanically crush the tisuses. For the detection of IL-22BP, PTGES3, COX1 and COX2, in CD11b$^+$ and CD11c$^+$ cells ($1–3 \times 10^5$) isolated from mouse intestinal polyps, after PGE$_2$ or curdlan sti-mulation for 20 h, cells were washed with 1 ml PBS and were directly lysed in 100 μl RAPI buffer. Lysate was centrifuged at 12,000 rpm for 30 min at 4 °C. The supernatant was transferred to a new tube and the protein concentration was measured with BCA protein assay kit (Bioteke, Cat# PP1001). The cell lysate was denatured for 10 min at 99 °C in a loading buffer (NuPAGE 4×, Reducing Agent 10×, Beyo-time, Cat# P0015L) and was loaded on a ExpressPuls™ 4–20% PAGE Gel (Genscript, Cat# M42012C). After electrophoresis, proteins in the gel were transferred to a PVDF membrane (Millipore, Cat# IPVH00010) for 60 min with 200 mA. PowerPa™ Basic power supply and Mini-PROTEAN Tetra Cell 4-Gel System (Bio-Rad) were used for both electrophoresis and membrane transfer. The membrane was then sunk in a TBST solution (50 mM Tris-Cl, 150 mM NaCl, 0.1% Tween-20) with 5% of non-fat milk (Biofroxx, Cat# 1172gr100), and was then incubated with specific primary antibodies (listed below) overnight at 4 °C. In the next morning, the membrane was washed and was incubated with a horseradish peroxidase-conjugated sec-ondary antibody for 1 h. The membrane was then washed, and protein bands on the membrane were visualized by the Pierce™ ECL Western Blotting Substtrate (Thermo, Cat# XA338096) and were detected by Amersham™ Imager 600. Chemiluminescence inten-sity of each band was calculated by the imaging analysis software ImageJ version 1.53t.

Antibodies used:

Mouse IL-22BP Antibody (Bio-techne, Cat# AF2376, 0.1 μg/ml)

Human/Mouse p23/PTGES3 Antibody (Bio-techne, Cat# MAB100391, 2 μg/ml)

Human/Mouse COX-2 Antibody (Bio-techne, Cat# AF4198-SP, 1 μg/ml)

Human/Mouse COX-1 Antibody (Bio-techne, Cat# MAB37401, 2 μg/ml)

THETM beta-Actin mouse Antibody (GenScript, Cat# A00702-100, 1:3000)

HRP-conjugated Affinipure Donkey Anti-Goat IgG (H + L) (Pro-teintech, Cat# SA00001-3, 1:1000)

HRP-conjugated Affinipure Rabbit Anti-Sheep IgG (H + L) (Pro-teintech, Cat# SA00001-16, 1:1000)

Goat Anti-Mouse IgG Antibody (H&L) [HRP], pAb (GenScript, Cat# A00160, 1:3000).

**Article** https://doi.org/10.1038/s41467-023-37229-x

## Statistical analysis

Differences in parametric data were evaluated by the unpaired two-tailed Student's *t*-test. For experiments containing more than two relative groups, one-way ANOVA followed by Tukey's multiple-comparisons test was performed. Statistical significance of survival rate between groups was assessed by the log-rank test. Statistics were computed with the PRISM v8 & v9.0 software (GraphPad Software). For CRC patient study, categorical variables were compared by the Fisher tests, correlation was analyzed by the Spearman's correlation test and statistical analysis was performed using the R software (version 4.0.3, R Foundation). Difference of *P*-value < 0.05 were considered statistically significant.

## Reporting summary

Further information on research design is available in the Nature Portfolio Reporting Summary linked to this article.

## Data availability

Source data are provided with this paper. Barcodes, features and matrix files for scRNA-seq analysis and raw sequencing files for bulk RNA-seq analysis have been deposited in GEO with the accession number GSE221206. The publicly available data of single-cell RNA sequencing used in this study are also available in the GEO database under accession code GSE196054, GSE178318, and GSE146771. The remaining data are available within the Article, Supplementary Information or Source Data file. Source data are provided with this paper.

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

## Acknowledgements

We thank Sachiko Kubo for her excellent technical assistance. We thank Dr. Xiaoxing Li, Dr. Wu Song, Dr. Ji Cui, Dr. Lixia Xu, and Dr. Shuishen Zhang (The First Affiliated Hospital, Sun Yat-sen University) for kindly providing tissue samples from CRC patients after colorectal tumor-ectomy, and we thank Dr. Fang Wang and Min Zhang (The First Affiliated Hospital, Sun Yat-sen University) for their kind help on scRNA-seq analysis. We also thank Takao Kuge at Adeka Corporation for his kind gift of short chain β-glucans. This work was supported by General Program of National Natural Science Foundation of China (82070564, C.T.), the Foundation of 100 Talents Program of Sun Yat-Sen University (Y61231, C.T.), the Fundamental Research Funds for the Central Universities, Sun Yat-sen University (C.T.), the Grants-in Aid from the Ministry of Education, Culture, Sports, Science and Technology of Japan [Scientific

Research (B), (20H03176, C.T. & 18H02671, Y.I.), and JSPS KAKENHI Grant-in-Aid for Scientific Research on Innovative Areas 18H05045 and 2R.(Y.I.)], and was carried out as a collaboration research with Boehringer Ingelheim International GmbH (Y.I.).

## Author contributions

C.T. mainly contributed to work and wrote the manuscript in collaboration with H.S. and M.K. W.H., Y.M., and X.Y. helped laminarin-administrating experiments. J.D., B.F., D.Q., Y.T., X.W., and Z.G. performed in vivo and in vitro $PGE_2$-stimulation experiments and CRC tissue and cell culture experiments. C.H., S.P., and M.C. supported in gathering tissue samples from CRC patients after colorectal tumorectomy. Y.A. and N.O. provided the OXCA for in vitro experiments. S.T. provided advice on designing experiments. Y. I. and C.T. organized and supervised the project and edited the manuscript.

## Competing interests

The authors declare no competing interests.

## Additional information

Ce Tang ®[1,2,3] ✉, Haiyang Sun[2,3], Motohiko Kadoki[3], Wei Han[3], Xiaoqi Ye[1,3], Yulia Makusheva[3], Jianping Deng[2], Bingbing Feng[2], Ding Qiu[2], Ying Tan[2], Xinying Wang[2], Zehao Guo[1], Chanyan Huang[4], Sui Peng[1,2], Minhu Chen[1], Yoshiyuki Adachi[5], Naohito Ohno[5], Sergio Trombetta[6] & Yoichiro Iwakura ®[3] ✉

[1]Department of Gastroenterology and Hepatology, The First Affiliated Hospital, Sun Yat-sen University, No.58, Zhong Shan Er Lu, 510080 Guangzhou, Guangdong Province, China. [2]Institute of Precision Medicine, The First Affiliated Hospital, Sun Yat-sen University, No.58, Zhong Shan Er Lu, 510080 Guangzhou, Guangdong Province, China. [3]Center for Animal Disease Models, Research Institute for Biomedical Sciences, Tokyo University of Science, Yamazaki 2669, Noda-shi, Chiba 278-0022, Japan. [4]Department of Anesthesiology, The First Affiliated Hospital, Sun Yat-sen University, No.58, Zhong Shan Er Lu, 510080 Guangzhou, Guangdong Province, China. [5]Laboratory for Immunopharmacology of Microbial Products, School of Pharmacy, Tokyo University of Pharmacy and Life Sciences, Hachioji, Tokyo 192-0392, Japan. [6]Boehringer Ingelheim USA, 900 Ridgebury Rd, Ridgefield, CT 06877, USA. ✉e-mail: tangc7@mail.sysu.edu.cn; iwakura@rs.tus.ac.jp

