## [Peer Review File · Nature Communications]

Blocking Dectin-1 prevents colorectal tumorigenesis by suppressing prostaglandin E2 production in myeloid-derived suppressor cells and enhancing IL-22 binding protein expressionREVIEWER COMMENTS

Reviewer #1 (Remarks to the Author): with expertise in colorectal cancer, prostaglandins

Here Tang et al. showed that inhibition of Dectin-1 by genetic deletion of *Clec7a* or by an antagonist reduced tumor burden accompanied by a reduction of the accumulation of myeloid cells in colorectal tumor tissues, downregulation of *Ptgs2* and *PGE2* levels in tumor-infiltrating myeloid cells, and upregulation of *Il22bp* in tumor-infiltrating DCs in two mouse models of CRC. They also revealed that activation of Dectin-1 by its ligands induced *PTGS2* and *PTGES* expression in tumor-infiltrating myeloid cells, and *PGE2* was downregulated by *Il22bp* in tumor-infiltrating DCs in vitro. This study is quite interesting. However, some key conclusions are not fully supported by the experimental results, as outlined in more detail below. In addition, an important limitation of the study is the lack of a mechanism to explain how Dectin-1 induces *PTGS2* and *PTGES* expression and how *PGE2* downregulates *Il22bp*.

1. MDSCs are known to suppress effector T cell activation, proliferation, trafficking, viability, inhibit NK cell function, and induce Treg activation/expansion. The authors present data showing that Dectin-1 promoted colorectal tumorigenesis accompanied by induction of MDSC accumulation in tumor tissues, indicating that Dectin-1 can promote colorectal tumorigenesis by suppressing CD8⁺ T cell cytotoxicity and NK cell function via MDSCs. However, the authors state that T and B cells are NOT the major regulators of colorectal tumor development in the AOM/DSS-treated mouse model. Thus, how does Dectin-1 promote tumor development in their models?

2. Since CD11b⁺Ly5C⁺Ly6G⁺ are markers of both neutrophils and PMN-MDSCs, is it possible that this population is made up of neutrophils and not MDSCs?

3. Based on the results from Fig. 1g-k, it is clear that gut flora is required for AOM-DSS-induced colonic tumorigenesis. Indeed, the results showed that the effect of Dectin-1 on rectal tumorigenesis is independent of microbiota in AOM/DSS-treated mice. However, it is unclear whether the effect of Dectin-1 on colon tumorigenesis is independent of microbiota in the AOM/DSS-treated mice. In addition, it would be helpful to determine the effect of *Clec7a* deletion on colorectal tumorigenesis in GF-ApcMin mice.

4. The results from Fig. 4a, c, d, and g showed that *Ptgs2* was much higher in polyps, mainly in polyp-infiltrating myeloid cells than in normal tissues in WT mice. However, the results from Figs. 4b and 5c showed that *PGE2* levels in polyps were much lower than in normal tissues in WT mice. These results are not consistent. In general, *PGE2* levels are higher in tumor tissues than in matched normal tissues. In addition, the difference in *PGE2* levels in polyps between the control and the laminarin-containing diet does not appear significant in Fig. 5c.

5. It would also be helpful to confirm the q-PCR results as shown in Figs. 2e, 4h, and 6d by Western blotting assays. Does β -glucan OXCA fail to induce *Il-6* and *Il1b* in *Clec7a*-deficient mouse myeloid cells? Does β -glucan curdlan fail to induce expression of *Ptgs1*, *Ptgs2*, and *Ptgs3* in *Clec7a*-deficient mouse polyps-infiltrating myeloid cells?

6. The authors defined MHCII-CD11c-CD11b⁺ as MDSCs in Figs. 5g and S4d. However, this population could also include macrophages, MDSCs, and neutrophils. Please confirm.

7. Does laminarin treatment induce *Il22ra2* expression in polyp-infiltrating DCs in the AOM/DSS-treated mice? Does *PGE2* treatment inhibit the effect of laminarin on *Il22ra2* in vivo?

8. Is there any correlations between *CLEC7* and *PTGS2/PTGES* in TCGA and other human databases? Are *CLEC7* levels associated with prolonged survival times in these databases?

Reviewer #2 (Remarks to the Author): with expertise in Dectin, intestinal immunology

Comment for the authors

In this manuscript, Ce Tang et al. report that intestinal tumorigenesis induced by sodium azoxymethane dextran sulphate and ApcMin- is strongly suppressed in Clec7a-/- mice independently of commensal microbiota. The authors demonstrate in vitro and in vivo that Dectin-1, preferentially expressed in myeloid-derived suppressor cells (MDSCs), induced PGE2 synthesis and consequently suppressed the expression of tumor protective IL-22 binding protein which promotes tumorigenesis of colorectal cancer. Finally, this study reveals a central role of the Dectin-1-PGE2-IL-22BP axis in the regulation of intestinal tumorigenesis, suggesting Dectin-1 as a potential target for CRC therapy.

In general, although the involvement of Dectin-1 in establishment of an intra-tumoral immunosuppressive milieu is already described (PMID: 33968777, ref 51), this is an interesting study and well thought. However, I feel that more work is needed to support the conclusions of the paper. Indeed, immune cells populations are not enough characterized to advance the conclusions discussed by the authors. In conclusion, although this work is significant, it is necessary to improve the quality of the approach to increase the robustness of the study.

Major comments:

1. In murine experimental models of colorectal cancer (sodium azoxymethane dextran sulphate and ApcMin-), the tumorigenesis index is only assessed by colon length and polyp volume. The authors cannot only rely on these parameters to determine colorectal tumor development. A histological analysis is therefore necessary.
2. The authors characterize and identify by flow cytometry the MDSCs only on CD11b+, CD11c-, MHC-2-, LY6C and LY6G markers. The identification and characterization of MDSCs is too approximate. Indeed, in mice, no specific marker for MDSCs has been identified yet. The only way to identify them is the analysis of their functional characteristics; otherwise it is impossible to differentiate them from monocytes and granulocytes. Thus, it is essential to characterize their functions in order to assert that this population represents the MDSCs. Moreover, the functionality of the MDSCs in figure 4 and 7 (scRNA-seq analysis) was also not studied. Only the expression of certain genes is not sufficient to identify MDSCs. Further gene characterization is also needed. One of the characteristics of MDSCs is the co-expression of Arg-2 and NOS2. For example, although the expression of Arg-2 is shown, the expression of NOS2 is missing.
3. There are two subpopulations of MDSCs, G-MDSCs and M-MDSCs, which exhibit distinct phenotypes and functions. Since the authors show different proportions of G-MDSCs and M-MDSCs, thus, treating them uniformly as MDSCs is inappropriate. The authors have to consider them individually to determine their contribution to the effects mediated by Dectin-1.
4. Although the authors show that dectin-1 appears to act as a receptor involved in the MDSC/DCs balance in favor of MDSCs in tumorigenesis, there is no data showing that dectin-1 is involved in the differentiation of monocytes/granulocytes into MDSCs. To strengthen these data, it is necessary to directly show in vitro the involvement of Dectin-1 in the differentiation of MDSCs in humans and mice. An adoptive transfer approach should be also performed to confirm the role of dectin-1 in vivo.
5. The identification of cytokines is only represented in mRNA. Given the importance of IL-1 and IL-22 in the manuscript, it is necessary to determine them at the protein level.
6. The data acquired with laminarin are sometimes less straightforward than those obtained with curdlan. Given that laminarin can be either a Dectin-1 antagonist or agonist, the use of a more specific antagonist would be appreciated.

Minor comments:

- All flow cytometry assays are represented in % of cells. It is necessary to add the quantification of the number of cells for each analysis.
- The characterization of dendritic cells is weak. This characterization should be further investigated.

Responses to Reviewers' comments (NCOMMS-22-19011)

Reviewer #1:

Remarks to the Author:

Here Tang et al. showed that inhibition of Dectin-1 by genetic deletion of *Clec7a* or by an antagonist reduced tumor burden accompanied by a reduction of the accumulation of myeloid cells in colorectal tumor tissues, downregulation of *Ptgs2* and PGE2 levels in tumor-infiltrating myeloid cells, and upregulation of *Il22bp* in tumor-infiltrating DCs in two mouse models of CRC. They also revealed that activation of Dectin-1 by its ligands induced *PTGS2* and *PTGES* expression in tumor-infiltrating myeloid cells, and PGE2 was downregulated by *Il22bp* in tumor-infiltrating DCs in vitro. This study is quite interesting. However, some key conclusions are not fully supported by the experimental results, as outlined in more detail below. In addition, an important limitation of the study is the lack of a mechanism to explain how Dectin-1 induces *PTGS2* and *PTGES* expression and how PGE2 downregulates *Il22bp*.

Thank you very much for reviewing our manuscript. We really appreciate your time and thoughtful comments. According to your comments, we have carried out necessarily experiments to address your concerns and now we believe the revised manuscript is much improved. Our new data provide clear evidence to address every comment.

Previously, we and others showed that Dectin-1 induces cytokines through activation of Syk-CARD9-NF- κ B pathway (Saijo et al., *Immunity*; Drummond et al., *Eur J Immunol*, 2011). Furthermore, COX2 is suggested to be induced by NF- κ B p65 (Schmedtje et al., *J Biol Chem*, 1997; Duque et al., *Cell Signal*, 2006; Charalambous et al., *Br J Cancer*, 2009). Thus, we examined the roles of the Syk-NF- κ B pathway on the Dectin-1-induced PGE₂-synthesizing-enzyme expression. As shown in the figures below, we harvested CD11b⁺ and CD11c⁺ myeloid cells from colonic polyps of AOM-3DSS-treated WT mice, and stimulated these cells with Dectin-1 agonistic ligand curdlan for 20 h, and examined the effects of inhibitors against NF- κ B (κ B α : BAY11-7082) or SYK (R406) on the expression of *Ptgs3*, *Ptgs1* and *Ptgs2*. We found that both *Ptgs1* and *Ptgs2* expression were suppressed by these inhibitors, suggesting that

these PGE₂-synthesizing enzymes are induced by the activation of the Dectin-1-Syk-NF- κ B pathway. We added these results to new Fig. 4j and added related statements in line 282-287

in the revised manuscript.

Regarding the mechanisms by which PGE₂ downregulates IL-22BP, IL-22BP is reported to be suppressed by IL-18 (Huber et al., Nature. 2012) and be enhanced by retinoic acid (RA) (Martin et al., Mucosal Immunol 2014). Although *I18* expression in myeloid cells from polys of ApcMin mice was upregulated by Dectin-1 stimulation *in vitro* (left panel below), we could not find difference in IL-18 mRNA expression in mouse colon between WT and Dectin-1 KO mice (middle panel), probably because high levels of *I18* is widely expressed in lymphocytes and even in epithelial cells in a Dectin-1-independent manner (right panel).

Actually, we did find that *I18* expression was upregulated by the stimulation with PGE₂ (left panel below). Furthermore, *I22ra2* expression was suppressed by PGE₂, and this suppression was recovered by the treatment with anti-IL-18 antibody, indicating that PGE₂-induced IL-18 is potentially responsible for the PGE₂-induced *I22ra2* suppression (right panel below). We added these two panels to new Fig. S5i, S5j and added related statements in line

342-348 and 470-474 in the revised manuscript.

Another potentially important regulatory factor for IL-22BP is retinoic acid (RA). We found that expression levels of genes encoding RA synthesizing enzymes were increased in Dectin-1 KO mouse colon compared with WT mice (new Fig. S5k, l). Using single cell RNA-seq analysis, we found that *I22ra2* is mainly expressed in tumor-associated DCs (TADC) in Dectin-1 KO mice but not in WT mice (new Fig. 6m, shown in left panel below). We then identified that a series of genes in *Aldh* family and *A/Rdh* family (encoding RA synthases) were also exclusively expressed in KO polyp-infiltrating TADCs but not in these cells in WT mice (new Fig. 6n and new Fig. S5m, shown in right panels below). It was reported that PGE₂ suppresses retinal dehydrogenases (RALDH) expression in both mouse and human DCs (Stock et al., J Exp Med. 2011). We also observed that PGE₂ mainly suppressed the expression of *Aldh* family genes but not of *A/Rdh* family upon treatment of colonic CD11b⁺ and CD11c⁺ myeloid cells

with PGE₂ for 20 h (middle panels below). When we further added RA in both colonic and splenic myeloid cells to compensate the suppression of RA synthase genes by PGE₂, we found that PGE₂-suppressed *I/22ra2* expression was completely recovered by the treatment with RA

(shown in lower panels above), suggesting that suppression of RA is involved in the regulation of *I/22ra2* expression by PGE₂, at least in part. We added these data as new Fig. 6m-p and new Fig. S5m-o respectively, and also added related statements in line 349-360 and 470-474 in the revised manuscript.

1. MDSCs are known to suppress effector T cell activation, proliferation, trafficking, viability, inhibit NK cell function, and induce Treg activation/expansion. The authors present data showing that Dectin-1 promoted colorectal tumorigenesis accompanied by induction of MDSC accumulation in tumor tissues, indicating that Dectin-1 can promote colorectal tumorigenesis by suppressing CD8⁺ T cell cytotoxicity and NK cell function via MDSCs. However, the authors state that T and B cells are NOT the major regulators of colorectal tumor development in the AOM/DSS-treated mouse model. Thus, how does Dectin-1 promote tumor development in their models?

Thanks for this insightful comment. As shown in original Fig. S3c, d (new Fig. S3k, l), development of polyps after AOM-3DSS-treatment in Rag2 KO mice was greatly suppressed compared with WT mice, clearly indicating that adaptive immunity is rather promotive for the tumorigenicity, and not required for the anti-tumor immunity in this mouse model. Therefore, the only reasonable explanation for its tumorigenesis regulation is that the tumor induced by AOM-DSS depends on innate immunity-induced IL-22BP production, and MDSCs are important as the source of PGE₂ to suppress IL-22BP secretion by highly expressing PGE₂ synthesizing enzymes.

2. Since CD11b⁺Ly6C⁺Ly6G⁺ are markers of both neutrophils and PMN-MDSCs, is it possible that this population is made up of neutrophils and not MDSCs?

Thanks for the critical comment. As shown in the figure below, after AOM-DSS-treatment for 16 weeks, CD11b⁺CD11c⁻Ly6C⁺Ly6G⁺ cell population in the colonic polyps is expanded over 14 times compared with normal intestinal tissue. These tumor-infiltrating myeloid cells are

commonly considered as MDSCs and their immune-suppressive function was described by ample studies (Hegde et al., Immunity 2021; Veglia et al., Nat Rev Immunol 2021).

In this study, we also used single cell-RNA sequencing (scRNA-seq) analysis to distinguish MDSCs from neutrophils in mouse intestinal tumors (new Fig. S4a-c, NCBI GEO dataset GSE196054). Neutrophils express their unique markers such as *Cd117* and *Adpgk*, while MDSCs have a unique marker *Arg2*, the gene encoding arginase 2 that suppresses T cell activation and expansion, and share *S100a8* and *S100a9* with neutrophils. As shown in the panels below, cell cluster 3 which represents MDSCs can be distinguished from cluster 7 that represents neutrophils.

Next, we compared these populations between WT and *Clec7a*^{-/-} mice by scRNA-seq analysis using CD11b⁺ and CD11c⁺ myeloid cells isolated from colonic polyps after 16w AOM-3DSS treatment. As shown in the heatmaps below, we can clearly distinguish all myeloid-derived cell populations, including M-MDSCs, PMN-MDSCs and neutrophils. Compared with WT mice, Dectin-1 KO mice have greatly reduced number of MDSCs (red square), while the populations of all their DC subsets are expanded, consistent with our FACS data (new Fig. 2b,

and new Fig. S2c).

As shown in the UMAPs in upper panels below, reduction of the MDSC population in *Clec7a*^{-/-} mice is also exhibited (cluster 2: M-MDSC; cluster 3: PMN-MDSC), while the neutrophil (cluster 12) population is normal. By gating on cluster 3, 8 and 12, we carried out pseudo-time trajectory analysis to compare the ‘transition’ from neutrophils to PMN-MDSCs.

As shown in lower panels above, most of the neutrophils in WT colonic polyps 'transitioned' into PMN-MDSCs (WT54.8% vs KO10.8%, lower right panel), whereas those in *Clec7a*^{-/-} mice remain as neutro.→mdsc transitional cells which do not express *Arg2*, of which expression level changes in the process of cell 'transition' (lower left panel). These scRNA-seq data clearly show that MDSCs in mouse intestinal tumors can be distinguished from neutrophils and are much decreased in Dectin-1-deficient hosts. We added these data to new Fig. 3e-h and new Fig. S3d-f and related statement in line 205-234 in the revised manuscript.

3. Based on the results from Fig. 1g-k, it is clear that gut flora is required for AOM-DSS-induced colonic tumorigenesis. Indeed, the results showed that the effect of Dectin-1 on rectal tumorigenesis is independent of microbiota in AOM/DSS-treated mice. However, it is unclear whether the effect of Dectin-1 on colon tumorigenesis is independent of microbiota in the AOM/DSS-treated mice. In addition, it would be helpful to determine the effect of *Clec7a* deletion on colorectal tumorigenesis in GF-ApcMin mice.

We appreciate for your detailed comment. We also recognized that the colon and the rectum should be treated as different organs. As the reviewer mentioned, original Fig. 1g-k (new Fig. 1i-n) show that AOM-DSS induce tumors mostly in the rectum and colonic polyps rarely develop under germ-free conditions. To distinguish the development of polyps in the colon

and in the rectum, WT and Dectin-1KO mice were co-housed for 3 weeks under SPF conditions, then they were treated with AOM-3DSS for 16 weeks under continued co-housing conditions. As shown in the left panels above, after co-housing for total 19 weeks, polyp numbers in Dectin-1 KO mice were decreased in both the rectum and colon compared to WT mice. The right panels show the polyp number in the rectum and the colon, respectively.

These results indicate that the effect of Dectin-1 is independent of microbiota in both AOM-3DSS-induced rectal and colonic tumorigenesis.

Regarding the GF ApcMin mouse experiments, we are very sorry but we could not generate GF ApcMin mice because we have to shrink the mouse colony due to the COVID-19 pandemic. Therefore, instead of GF mice, we examined the effects of microbiota using antibiotics treatment. ApcMin mice were pre-treated with AVMN-combined antibiotics for 3

weeks and then were treated with DSS for 1 week. After another 4 weeks, tumor development in the colon and the rectum was examined. As shown in the panels above, development of tumor was significantly decreased in Dectin-1 KO mice compared with WT mice in two independent experiments, although only small number of polyps developed after treatment with antibiotics. These results suggest that commensal microbiota is not involved in the suppression of colonic polyp formation in *Apc^{Min/+} Clec7a^{-/-}* mice. We added these data as new Fig. S1n-q and the related statement in line 135-136 and 138-143 in the revised manuscript.

4. The results from Fig. 4a, c, d, and g showed that *Ptgs2* was much higher in polyps, mainly in polyp-infiltrating myeloid cells than in normal tissues in WT mice. However, the results from Figs. 4b and 5c showed that PGE₂ levels in polyps were much lower than in normal tissues in WT mice. These results are not consistent. In general, PGE₂ levels are higher in tumor tissues than in matched normal tissues. In addition, the difference in PGE₂ levels in polyps between the control and the laminarin-containing diet does not appear significant in Fig. 5c.

Ptgs2 mRNA levels relative to the *Gapdh* mRNA are shown in new Fig 4a, c, d, and h, while PGE₂ concentration per ml lysate which contains a constant amount of protein is shown in new Fig. 4b and new Fig. 5h. Here, mRNA levels and PGE₂ concentration were represented as relative values to reference mRNA and to total protein respectively. Relative contents of mRNA and PGE₂ may change depending on tissues, because *Gapdh* expression level and total protein concentration of lysate are different among different tissues. Probably, the *Gapdh*

levels are higher and/or the protein content is lower in normal tissue compared to those of polyp tissue. To avoid misunderstanding, we added the related statement of normalizing method in figure legends (line 953 and 987) in revised manuscript.

Regarding the effect of laminarin in original Fig. 5c, we have tried to measure the PGE₂ concentration for several times, but the data seems fluctuated more in polyp groups than in normal tissue groups, although the average level of PGE₂ was two-times lower in laminarin group than in control group. These data still suggest that PGE₂ production in the intestine is inhibited by laminarin in food, although the effect in polyps is not statistically significant.

5. It would also be helpful to confirm the q-PCR results as shown in Figs. 2e, 4h, and 6d by Western blotting assays. Does beta-glucan OXCA fail to induce Il-6 and Il1b in Clec7a-deficient mouse myeloid cells? Does beta-glucan curdlan fail to induce expression of Ptgs1, Ptgs2, and Ptgs3 in Clec7a-deficient mouse polyps-infiltrating myeloid cells?

We thank for the comments. According to the reviewer's concern, we measured IL-1 β and IL-6 expression at the protein levels. We used flow cytometry instead of Western blot assay, because flow cytometry is more sensitive in our experience. As shown in the panels below,

after stimulation of CD11b⁺ and CD11c⁺ intestinal polyp-infiltrated myeloid cells with OXCA for 6 h, both IL-1 β and IL-6 expression were upregulated and this upregulation was inhibited by laminarin. These results are consistent with the original mRNA expression data. We added these results as new Fig. S2d and added related statement in line 174-176 in the revised manuscript.

Although we already showed in previous reports that NaClO-oxidized *Candida albicans* (OXCA) is a pure Dectin-1 agonistic ligand (Saijo et al., Nat. Immunol., 2007; Tang et al., Cell Host & Microbe, 2015), we confirmed this using Dectin-1-deficient myeloid cells. As shown in panels below, no induction or enhancement of IL-1 β or IL-6 expression was detected in Dectin-1 KO CD11b⁺ and CD11c⁺ intestinal cells upon stimulation with OXCA.

Following the comment, we have also examined the expression of PGE₂ synthesizing enzymes in myeloid cells from intestinal polyps of ApcMin-Dectin-1 KO mice. As shown in the panels below, expression levels of *Ptgs1/2* or *Ptges2/3* were not upregulated in these cells

after stimulation with curdlan.

To confirm the protein levels of *Ptgs1*, *Ptgs2* and *Ptges3* genes in tumor-infiltrating myeloid cells after curdlan stimulation in original Fig. 4h (new Fig. 4i), CD11b⁺ and CD11c⁺

cells were harvested from ApcMin mouse intestines, and after stimulated with curdlan for 20 h, the expression of COX1, COX2 and PTGES3 was detected by Western blot assay. As shown in the panels above, we found that protein levels of all these three PGE₂ synthases were also upregulated by curdlan in a dose-dependent manner similarly as the mRNA levels. We added these data as new Fig. S4e and f and added related statement in line 279-280 in the revised manuscript.

Finally, we confirmed the protein levels of *Il22ra2* in myeloid cells after PGE₂ stimulation in original (and also new) Fig. 6d. Myeloid cells from intestinal polyps of ApcMin mice were stimulated with PGE₂ for 20 h, and IL-22BP was detected by Western blot assay. We found that, consistent with the mRNA levels, IL-22BP protein expression in tumor-infiltrating

myeloid cells was also suppressed by PGE₂ in a dose dependent manner (panels above). We also added these data to new Fig. S5e and f in the revised manuscript.

6. The authors defined MHCII-CD11c-CD11b+ as MDSCs in Figs. 5g and S4d. However, this population could also include macrophages, MDSCs, and neutrophils. Please confirm.

Thank you for the important comment. We noticed this, and because many researchers defined MDSCs as tumor-infiltrating MHC-II⁻CD11c⁻CD11b⁺ cells, we also followed that definition. However, as your present comment as well as comment #2 are very important, we

have further analyzed this population by FACS. As shown in the FACS dot plot panels above, when MHC-II⁻CD45⁺ cells were gated, we can further distinguish Ly6C^{hi}CD11b⁺ cells as M-MDSCs, Ly6C^{int}Ly6G⁺CD11b⁺ as PMN-MDSCs, and Ly6C⁻CD11b⁺ as macrophages. It was

difficult to distinguish PMN-MDSCs from neutrophils by FACS, as we mentioned in the response to comment #2. However, since MDSCs expanded for over 10 times more in tumor tissues compared with normal tissues (related data in response to comment #2), over 90% of

these Ly6G⁺CD11b⁺ polyp-infiltrating cells are considered as PMN-MDSCs. These MDSC populations were greatly expanded after PGE₂ administration (bar graphs above). We replaced original Fig. 5g to new Fig. 5c and S4g, with adding new MDSC panels and FACS dot plot panels and related statement in line 292-293 in the revised manuscript.

7. Does laminarin treatment induce IL22ra2 expression in polyp-infiltrating DCs in the AOM/DSS-treated mice? Does PGE2 treatment inhibit the effect of laminarin on IL22ra2 in vivo?

To address these comments, we firstly treated WT mice with AOM, followed by 5% laminarin administration in diet during 3 cycles of DSS-treatment. Then, colonic polyp- and nonpolyp-infiltrating cells from these mice at 13 weeks after the first AOM-administration were harvested, and IL-22BP expression in CD11c⁺MHC-II⁺ DCs was examined by FACS, since we

observed that IL-22BP is expressed only on intestinal DCs. As shown in the panels above, the mean fluorescence intensity (MFI) of IL-22BP on DCs from both polyp and non-polyp tissues was significantly increased in laminarin-administrated group compared with control group, indicating that laminarin treatment can induce or enhance IL-22BP expression in intestinal DCs after AOM-3DSS treatment. We added this result to new Fig. 6h and related statement

in line 329-330 in revised manuscript.

Regarding the effect of PGE₂, as we have no sufficient ApcMin mice, the effect was examined on WT mice. Mice were administered with 5% laminarin in diet from day 0 during

the whole experimental period, and were treated with 1.5% DSS at day 3 for 1 week. PGE₂ was i.p. injected 4 times during DSS treatment at 3 days interval. Then, colonic cells were harvested at day 12 and IL-22BP expression was analyzed by flow cytometry. As shown in the panels above, laminarin treatment significantly enhanced IL-22BP expression level in colonic DCs, and PGE₂ additional administration suppressed the laminarin-induced enhancement. These data indicate again that PGE₂ treatment inhibits the promotive effect of laminarin on IL-22BP expression. We added this result as new Fig. 6i and added related statement in line 331-332 in the revised manuscript.

8. Is there any correlations between CLEC7 and PTGS2/PTGES in TCGA and other human databases? Are CLEC7 levels associated with prolonged survival times in these databases?

Following the reviewer's comment, we have checked TCGA, the most popular and biggest database, and also several other public databases related to human cancer and gene expression, such as GDC (<https://gdc.cancer.gov/>), GEPIA (<http://gepia.cancer-pku.cn/>) or UALCAN (<http://ualcan.path.uab.edu/>), but we found these databases use the cancer-gene expression data from original TCGA database. Therefore, we just focused on TCGA and checked the expression correlations between *CLEC7A* and *PTGS2* or *PTGES* in human CRC cancer. As shown in the upper panels above, the expression of *CLEC7A* is significantly correlated with *PTGS2*, *PTGES* and *PTGES3* gene expression, similar to our own data in original (and also new) Fig. 7g. Although the correlation between *CLEC7A* expression and shortened survival is not statistically significant, CRC patients with low expression level of *CLEC7A* tended to live longer. Interestingly, consistent with our observations in the present study, *PTGES* expression is significantly negatively correlated with prolonged survival time and *IL22RA2* level is positively correlated (lower panels above). We added these data to new Fig. S6c and g and added related statement in line 380-382 and 425-429 in revised manuscript.

Reviewer #2:

Remarks to the Author:

In this manuscript, Ce Tang et al. report that intestinal tumorigenesis induced by sodium azoxymethane dextran sulphate and ApcMin- is strongly suppressed in Clec7a^{-/-} mice independently of commensal microbiota. The authors demonstrate in vitro and in vivo that Dectin-1, preferentially expressed in myeloid-derived suppressor cells (MDSCs), induced PGE2 synthesis and consequently suppressed the expression of tumor protective IL-22 binding protein which promotes tumorigenesis of colorectal cancer. Finally, this study reveals a central role of the Dectin-1-PGE2-IL-22BP axis in the regulation of intestinal tumorigenesis, suggesting Dectin-1 as a potential target for CRC therapy.

In general, although the involvement of Dectin-1 in establishment of an intra-tumoral immunosuppressive milieu is already described (PMID: 33968777, ref 51), this is an interesting study and well thought. However, I feel that more work is needed to support the conclusions of the paper. Indeed, immune cells populations are not enough characterized to advance the conclusions discussed by the authors. In conclusion, although this work is significant, it is necessary to improve the quality of the approach to increase the robustness of the study.

Thank you very much for reviewing our manuscript. We appreciate very much your time and efforts to read our manuscript and your favorable comments. To address your concerns, we have carried out additional experiments including single cell RNA-seq analysis and now we believe all the concerns are resolved.

Major comments:

1. In murine experimental models of colorectal cancer (sodium azoxymethane dextran sulphate and ApcMin-), the tumorigenesis index is only assessed by colon length and polyp volume. The authors cannot only rely on these parameters to determine colorectal tumor development. A histological analysis is therefore necessary.

Following this important comment, we have analyzed the histopathology of tumors. Tumors were cut from both colon and rectum of AOM-3DSS-treated mice, and tissue sections were stained with HE. As show in the tables below, the histological score was evaluated according to the reported method (Kargl J. et al., Mol Med (Berl). 2013). All 5 WT mice developed

Histologic criteria of tumors	Score	
Number of lesions	0-2	1
	3-5	2
	6-8	3
	9-	4
Crypts	normal	0
	goblet cell depletion	1
	branching	2
	complex budding	3
Epithelium	normal	0
	hyperplasia/ACF	1
	low-grade dysplasia	2
	high-grade dysplasia	3
Submucosal invasion	absent	0
	present	1

ACF, aberrant crypt foci

colon				rectum			
	lesion	score		lesion	score		score
wt1	6	3		2	1		
wt2	3	2		2	1		
wt3	2	1		1	1		
wt4	2	1		1	1		
wt5	1	1		2	1		
ko1	0	1		1	1		
ko2	1	1		0	1		
ko3	0	1		0	1		
ko4	0	1		0	1		
ko5	0	1		1	1		
	Crypts			Crypts			
wt1	cb	3		cb	3		3
wt2	cb	3		cb	3		3
wt3	cb	3		cb	3		3
wt4	cb	3		cb	3		3
wt5	gcd	1		cb	3		3
ko1	n	0		br	2		2
ko2	gcd	1		n	0		0
ko3	n	0		n	0		0
ko4	n	0		n	0		0
ko5	n	0		gcd	1		1

colon			rectum		
	Epithelium score		Epithelium score		
wt1	hgd	3	hgd	3	
wt2	hgd	3	hgd	3	
wt3	lgd	2	hgd	3	
wt4	hgd	3	hgd	3	
wt5	acf	1	lgd	2	
ko1	n	0	lgd	2	
ko2	acf	1	n	0	
ko3	n	0	n	0	
ko4	n	0	n	0	
ko5	n	0	acf	1	
	Sub-invas. score		Sub-invas. score		
wt1	ab	0	ab	0	
wt2	ab	0	ab	0	
wt3	ab	0	ab	0	
wt4	ab	0	ab	0	
wt5	ab	0	ab	0	
ko1	ab	0	ab	0	
ko2	ab	0	ab	0	
ko3	ab	0	ab	0	
ko4	ab	0	ab	0	
ko5	ab	0	ab	0	

intestinal tumors, while only 1 out of 5 KO mice developed tumors. Complex budding of crypts (cb) was seen in 4/5 colon and 5/5 rectum of WT mice, but none in KO mice. Most of the WT mice exhibited high-grade dysplasia or hyperplasia, but only low-grade dysplasia or hyperplasia was observed in 2 KO mice. Submucosal invasion was observed in these AOM-DSS-induced tumors. Representative photos of tumors from colon and rectum are shown in new Fig. 1k, and combined scores of 4 histologic criteria are shown in new Fig.1l (panels shown below).

We also examined the development of tumors in *Apc*^{Min} mice following the comment. We found that DSS-treated *Apc*^{Min} mice developed both colonic and rectal tumors with high-grade dysplasia, but histological score was greatly suppressed in *Apc*^{Min}-Dectin-1 KO mice (panels shown below). Representative photos of tumors from colon and rectum are

shown in new Fig. 1e and combined scores of 4 histologic criteria are shown in new Fig.1f. We added related statements in line 111-113 and 122-125 in the revised manuscript.

2. The authors characterize and identify by flow cytometry the MDSCs only on CD11b⁺, CD11c⁻, MHC-2⁻, LY6C and LY6G markers. The identification and characterization of MDSCs is too approximate. Indeed, in mice, no specific marker for MDSCs has been identified yet. The only way to identify them is the analysis of their functional characteristics; otherwise it is impossible to differentiate them from monocytes and granulocytes. Thus, it is essential to characterize their functions in order to assert that this population represents the MDSCs. Moreover, the functionality of the MDSCs in figure 4 and 7 (scRNA-seq analysis) was also not studied. Only the expression of certain genes is not sufficient to identify MDSCs. Further gene characterization is also needed. One of the characteristics of MDSCs is the co-expression of

Arg-2 and NOS2. For example, although the expression of Arg-2 is shown, the expression of NOS2 is missing.

Thanks for the important comment. To address the reviewer's concern, we firstly carried out scRNA-seq analysis by using purified CD11b⁺ and CD11c⁺ myeloid cells from colonic polyps of AOM-3DSS-treated WT and Dectin-1 KO mice. As shown in the heatmap and UMAPs below, we could clearly distinguish different myeloid-derived cell types, such as M-MDSCs,

PMN-MDSCs (cluster 2&3, in yellow square on the heatmaps) and neutrophils. The MDSC

populations were greatly decreased in Dectin-1 KO mice compared with WT mice, while DC populations were expanded in Dectin-1 KO mice (the UMAPs above), consistent with the original FACS data (in new Fig. 2b, c). Following the reviewer's suggestion, we also examined the expression of characteristic genes of MDSCs including *Nos2* and *Arg2*. We found that although *Arg2* was highly expressed in M-MDSC and PMN-MDSC, *Nos2* expression level was relatively low and was found in M-MDSC, PMN-MDSC and monocyte clusters with relatively small cell proportions (the heatmap shown above).

Following the comment, we re-analyzed the scRNA-seq data in original Fig. 4f (new Fig. S4c) for the *Nos2* expression, and found that although *Nos2*-positive cells are in small number, most of them were gathered in *S100a9*+*Arg2*+*Clec4d* triple-positive MDSCs (Umap panels below). Meanwhile, we found that *Vegfa* gene, which is important for tumor angiogenesis, is highly expressed in MDSCs. We added these data in new Fig. S4c and added related statement

in line 271-274 in the revised manuscript.

In human CRC-associated MDSCs in original (and also new) Fig. 7h, *NOS2* expression was not detectable (only positive in one MDSC cell), and *ARG2* was detected in only minor population (right panels below). Meanwhile, MDSCs expressed *VEGFA* at high levels, accompanied with increased expression of *IL1B*, *CLEC4E* and *S100A9*, of which expression was

low in other myeloid cells other than MDSCs. These observations indicate that MDSC population is clearly distinguishable by the expression of these genes from other myeloid cells. We confirmed this by using another human CRC scRNA-seq dataset (GSE146771). We found that the transcription levels of *VEGFA* were also in parallel with the expression levels of *S100A9*, *IL1B*, *PTGS2* and *CLEC7A* in the pseudotime-trajectory (right panels below), suggesting that

VEGF is another Dectin-1-dependent tumor-promoting factor of MDSCs. We added these data to new Fig. S7i and j and added related statement in line 407-414 in the revised manuscript.

Since *Nos2* expression was difficult to detect in transcription level, we tried to detect NOS2 protein in intestinal polyp-infiltrating CD11b⁺ cells by using flow cytometry. As shown in the FACS panels below, NOS2 was clearly detected in WT cells by the intracellular staining, but its

concentration was greatly reduced in *Clec7a*^{-/-} cells in both AOM-3DSS and ApcMin models, supporting our notion that MDSC population is reduced in Dectin-1 KO mice. The inconsistency between the mRNA and protein levels of NOS2 may be caused by multiple factors that influence the stability of *Nos2* mRNA (Nathan C et al., J Biol Chem 1994; Park SK et al., J Neurochem 1996). We added the ApcMin data as new Fig. 3l and added related statement in line 248-249 in the revised manuscript.

Next, we examined functional characteristics of these apparent MDSC populations. EpCAM⁺MHC-II⁺Gr1⁺ cells were purified from colonic polyps or spleen from WT and Dectin-

1 KO mice after AOM-3DSS treatment, and co-cultured with splenic CD4⁺ T cells isolated from a WT mouse in steady state. Then, T cells were stimulated with anti-(CD3+CD28) beads. After co-culture for 72 h, we found that T cell population was significantly decreased by being cultured with the colonic polyp-infiltrating Gr1⁺ cells from WT mice, but these cells from Dectin-1 KO mice only weakly reduced the T cell number (lower panels above). Thus, these apparent MDSC population can actually suppress T cell population, indicating these cells represent MDSCs.

We also found the similar phenomenon that both CD4⁺ T cell population and IFN- γ -producing T cells proportion was significantly suppressed after co-culture with splenic MHC-II⁺Gr1⁺ cells isolated from AOM-3DSS-treated WT mice, and those myeloid cells from *Clec7a*^{-/-} mice suppressed less efficiently (panels below), indicating the impairment of immune

Spleen

suppressive activity of Dectin-1-deficient MDSCs. We added these data as new Fig. 3k and S3i, j and added related statement in line 243-248 in the revised manuscript.

3. There are two subpopulations of MDSCs, G-MDSCs and M-MDSCs, which exhibit distinct phenotypes and functions. Since the authors show different proportions of G-MDSCs and M-MDSCs, treating them uniformly as MDSCs is inappropriate. The authors have to consider them individually to determine their contribution to the effects mediated by Dectin-1.

Thank you for the important comment. We agree that PMN-MDSCs and M-MDSCs may have distinct phenotypes and functions. However, in our present study, we focused our elaboration to show the role of Dectin-1 signaling in the development of MDSCs during tumor development. In this study, we have shown that the differentiation/expansion of both PMN-MDSCs and M-MDSCs is promoted by the Dectin-1 signaling, but we think that to elucidate the functional roles of these MDSC subpopulations in the development of tumors is beyond the focus of our present study.

Regarding your comment, we carried out single cell RNA-seq analysis to further elucidate the influence on PMN- and M-MDSCs by Dectin-1 deficiency using tumor-infiltrating myeloid cells from AOM-3DSS-treated WT and Dectin-1 KO mice. As shown in the very first heatmaps and UMAPs in the response to comment #2, we found that both M-MDSCs and PMN-MDSCs are markedly decreased in Dectin-1 KO mice. In the UMAPs, cluster #2 representing M-MDSCs and cluster #3 representing PMN-MDSCs are nearby each other, although M-MDSCs form a group with monocytes (cluster #0, 1, 2) and PMN-MDSCs form a group with neutrophils (cluster #3, 8, 12), respectively. Gene expression patterns of these two types of MDSCs are closely resembled each other (the square with red point line in the left 'WT' heatmap), suggesting their functional similarity, although monocyte-macrophages and

neutrophils exhibit distinct gene profiles.

Then, to demonstrate the influence of Dectin-1 signaling on M- and PMN-MDSC development, we analyzed transition of M-MDSCs and PMN-MDSCs from monocyte- and neutrophil-progenitor, respectively in WT and Dectin-1 KO mice. By gating on cluster #0, 1 and 2 (monocyte group) or on cluster #3, 8, 12 (neutrophil group), we carried out pseudo-

time trajectory analysis. As shown in panels above, relative expression of characteristic gene of MDSCs like *S100a9*, *Csf3r* and *Arg2* was gradually upregulated during MDSC-transition process, and this *Arg2* expression change can be observed more visually in the trajectory panels below. As shown in the upper trajectory panels, macrophages first ‘transitioned’ into

monocytes, then monocytes to M-MDSCs. In this process, although monocyte content was similar between WT and Dectin-1 KO mice (19.4% vs 23.2%), macrophage content was increased (WT37.0% vs KO63.6%) and M-MDSC content was decreased (WT39.8% vs KO17.0%) in Dectin-1 KO mice, suggesting that monocyte→M-MDSC transition is impaired in colonic polyps of Dectin-1 KO mice. As shown in the lower trajectory panels, although most of the neutrophils in Dectin-1 KO mice first fully ‘transitioned’ into neutro.→mdsc transitional cells (which do not express *Arg2*), the final ‘transition’ of these cells into PMN-MDSCs was greatly impaired in Dectin-1 KO mice (WT54.8% vs KO10.8%), suggesting that Dectin-1 signaling also promotes the PMN-MDSC development. We added these data to new Fig. 3e-h and new Fig. S3d-f and added related statement in line 205-234 in the revised manuscript.

4. Although the authors show that dectin-1 appears to act as a receptor involved in the MDSC/DCs balance in favor of MDSCs in tumorigenesis, there is no data showing that dectin-

1 is involved in the differentiation of monocytes/granulocytes into MDSCs. To strengthen these data, it is necessary to directly show *in vitro* the involvement of Dectin-1 in the differentiation of MDSCs in humans and mice. An adoptive transfer approach should be also performed to confirm the role of dectin-1 *in vivo*.

Thank you for the important comment. Accordingly, we examined MDSC differentiation *in vitro* by culturing bone marrow cells from WT mice with GM-CSF and IL-6 for 5 days (Curr Protoc Immunol. 2019, 124: e61; Methods in Molecular Biology, MIMB, volume 2236). On the second or the fourth day of the culture, curdlan was added for 3 day- or 1 day-stimulation, and MDSC differentiation was analyzed by flow cytometry using CD11b, Ly6C and MHC-II as

the markers. As shown in the FACS patterns above, CD11b⁺Ly6C⁺ cells, which are also MHC-II⁺, were greatly increased after treatment with curdlan for 1 or 3 days in a dose-dependent manner. We further analyzed typical markers for MDSCs by qPCR, and found that immune

suppressive genes such as *Arg2* and *Nos2*, and MDSC-differentiation promotive genes such as *Il1b*, *Csf3* and *Ptgs2*, were greatly upregulated after Dectin-1 stimulation for 3 days (bar graphs above).

Next, we examined the immune suppressive function of Dectin-1-induced MDSCs. After stimulation of bone marrow cells with curdlan under the MDSC-differentiation conditions for 5 days, these cells were co-cultured with splenic CD4⁺ T cells, and stimulated with anti-CD3+CD28 beads for 72 h. As shown in the panels below, CD11b⁺Ly6C⁺ cells were increased after Dectin-1 stimulation, and CD4⁺ T cells co-cultured with curdlan-stimulated MDSCs were

decreased in a dose-dependent manner. We also found the IFN- γ ⁺ frequencies were decreased in CD4⁺ T cells. These data indicate that Dectin-1 signaling can directly promote the differentiation or expansion of MDSCs in mice. We added these results to new Fig. 3i, j and new Fig. S3h and added related statement in line 235-242 in the revised manuscript.

Regarding human MDSC differentiation *in vitro*, due to the limitation of the institutional ethical regulations in Sun Yat-sen University (approval number: IIT-2021-654), we were not allowed to use human bone marrows. Instead, we could use PBMCs isolated from a healthy volunteer for the induction of MDSCs according to previous reports (Methods in Molecular Biology, volume 2236; J Transl Med. 2011, 9: 90). Because the cell number was not enough, we just made duplicates for each group and stimulated these PBMCs with curdlan together with GM-CSF and IL-6. To inhibit COX2, celecoxib was added to the culture to see the effect

on MDSC differentiation. As shown in the panels above, the expression of CD33, a marker for human MDSC, could be graded into three levels together with the expression of CD14: CD33^{hi}CD14^{hi/int}, CD33^{int}CD14⁻ and CD33^{lo}CD14⁻. More than 70% of CD33^{hi}CD14^{hi/int} population were CD11B⁺MHC-II⁺ and only 6% are CD11B⁺MHC-II⁻ (M-MDSCs), while over 80% of CD33^{int}CD14⁻ cells were CD15⁺MHC-II⁻ (PMN-MDSCs). We found that the proportion of CD11B⁺CD33⁺CD15⁺ PMN-MDSCs in myeloid cells was increased after curdlan stimulation

and COX2 inhibitor suppressed this increase (FACS panels and bar graphs above). The curdlan stimulation also enhanced the population of CD33^{int}HLA-DR^{low}CD14⁺ M-MDSCs and the

blockade of COX2 also suppressed this enhancement (panels above). In contrast, CD33^{hi}HLA-DR^{hi}CD14⁺ antigen-presenting cells were decreased after curdlan treatment. When the effect of curdlan on MDSC-mediated T cell suppression was examined, the proportions of both CD3ε⁺ T cells and CD3ε⁺CD8α⁺ T cells were decreased after curdlan treatment in a dose-dependent manner (panels below). By using PBMCs from another healthy volunteer, we

repeated the experiment above and confirmed the similar results that stimulation with curdlan enhances HLA-DR⁻CD15⁺ and HLA-DR⁻CD14⁺ cell expansion and suppresses CD3⁺ and CD8⁺

T cell proliferation (bar graphs above). We added the representative data above to new Fig. 7l, m and S6f and added related statement in line 418-423 in the revised manuscript.

The *in vivo* transfer experiment is a difficult question for us. We first considered to try to transfer KO mouse bone marrow cells (BMCs) to WT mice and induce AOM-DSS tumors in these recipients. But we noticed this experiment is not meaningful, because Dectin-1 KO BMC-transferred mice should behave like KO mice because Dectin-1 is only expressed in myeloid cells. Then, we thought to transfer 1:1 mixed BMCs from WT and Dectin-1 KO mice. However, because Dectin-1 can induce PGE₂ production and PGE₂ promotes MDSC differentiation *in vivo* (new Fig. 5c, d), co-transferred WT BMC-derived MDSCs may 'help' Dectin-1 KO MDSC expansion by producing PGE₂, resulting in the effect of Dectin-1 deficiency obscure. Therefore, we stopped to carry out these *in vivo* experiments. But to confirm our hypothesis about the potential 'help' from WT cells to the KO cells, we co-cultured CD45.1⁺ WT BMCs with CD45.2⁺ *Clec7a*^{-/-} BMCs under the condition of MDSC differentiation and then with curdlan or PGE₂ stimulation (method in the panel below). As

Co-cultured

shown in the FACS panels above, under the co-culture and MDSC differentiation condition,

proportions of Ly6C⁺CD11b⁺ cells were increased in both WT and KO groups after both curdlan and PGE₂ stimulation ('*Co-cultured*' panels). In contrast, under the separated-culture condition, the frequency of Ly6C⁺CD11b⁺ cells from only WT but not KO group was increased after curdlan stimulation, but those in both WT and KO groups were increased after PGE₂

Separated

stimulation ('*Separated*' panels above). Ratios of live cell numbers normalized with control group clearly showed that co-cultured WT cells 'helped' KO MDSC expansion when Dectin-1 ligand curdlan was added, but without this 'help' when KO cells were cultured separately,

these cells could not further expand with curdlan stimulation (bar graphs above). Both WT and KO cells further expanded after PGE₂ stimulation, and COX2 inhibitor celecoxib cancelled this expansion (right panel above). All these results indicate that the development of MDSCs can be promoted by Dectin-1 and PGE₂ signaling. We are very sorry for giving up the adaptive-transfer experiment, but we believe that our present *in vivo* and *in vitro* results should be sufficient to show Dectin-1 signaling causes MDSC expansion and promotes colorectal tumor development. We appreciate for your understanding.

5. The identification of cytokines is only represented in mRNA. Given the importance of IL-1 and IL-22 in the manuscript, it is necessary to determine them at the protein level.

According to the reviewer's suggestion, we first examined IL-1β protein expression in intestinal polyp-infiltrating CD11b⁺ and CD11c⁺ cells after Dectin-1 antagonist OXCA stimulation using flow cytometry (mRNA expression was measured by qPCR in original Fig.

2e). As shown the data above, OXCA-stimulation enhanced pro-IL-1β production in CD11b⁺ cells and Dectin-1 antagonist laminarin treatment suppressed this IL-1β production.

We then measured IL-1β protein concentration in colonic polyps and non-polyp tissues after 16 weeks AOM-3DSS treatment. Tissues were harvested and lysed with a lysing buffer,

and IL-1β concentration/tissue weight was measured by ELISA. As shown in the data above, IL-1β concentration in polyps of Dectin-1 KO mice was significantly decreased compared with that in WT mice, consistent with the mRNA levels in RNA-seq and qPCR analysis.

Regarding IL-22, we found IL-22 mRNA levels were similar between WT and Dectin-1 KO mice, but the expression levels of its antagonistic receptor, IL-22BP (gene *I22ra2*), markedly increased in Dectin-1 KO mice. Compared with IL-22, we paid more attention to IL-22BP in our present study, so we confirmed its protein level here. We firstly measured IL-22BP protein levels in colonic tumors. After AOM-3DSS treatment, colonic polyps were harvested and IL-22BP protein was detected by Western blotting. As shown in the panels below, levels of IL-22BP protein was significantly higher in Dectin-1 KO mice compared with that in WT mice,

consistent with the *I22ra2* mRNA levels in RNA-seq or qPCR analysis.

We also determined IL-22BP levels in polyp-infiltrated myeloid cells that were isolated from ApcMin-Dectin-1 KO mice and were treated with PGE₂ for 20 h. As shown in the panels below, PGE₂ treatment suppressed the IL-22BP protein expression in a dose-dependent manner, consistent with the mRNA level change in original (and also new) Fig. 6d. We added

all these new data to new Fig. S2d, S3g, S5b, c, e, f and added related statement in line 174-176, 220-222 and 311-312 in the revised manuscript.

6. The data acquired with laminarin are sometimes less straightforward than those obtained with curdlan. Given that laminarin can be either a Dectin-1 antagonist or agonist, the use of a more specific antagonist would be appreciated.

We agree with the reviewer's comment that laminarin sometimes acts as a Dectin-1 agonist. We found that laminarins from different makers have different molecular weight and those with high MW (over 30,000) have Dectin-1 agonist activity. Laminarin with low MW (average 4400), produced by Sigma-Aldrich (St. Louis, MO), only acts as Dectin-1 antagonist, but laminarins from other companies with MW over 34,000 stimulate RAW264.7 cells to produce TNF- α and are recognized as Dectin-1 agonists (J Immunol 2018, 200: 788-799). The laminarin we used is the one with low MW from Sigma, and our data also showed that this laminarin can block OXCA or curdlan to activate Dectin-1.

We thank for the reviewer's advice that we should use a more Dectin-1-specific antagonist for the present study. We actually have already tried to generate anti-mouse Dectin-1 neutralizing antibodies and have examined the effects both *in vitro* and *in vivo*. Splenocytes

3H8

were obtained from WT mice and stimulated with curdlan (1,000 $\mu\text{g}/\text{ml}$). Then, after treatment with anti-mouse Dectin-1 antibody (clone number 3H8) for 20 h, *Il1b* and *Il6* mRNA expression were examined by qPCR. As shown in the left qPCR panels above, 3H8 suppressed curdlan-induced cytokine production *in vitro*. Then, we administrated mice with this antibody. This antibody could cause antibody-dependent cell-mediated cytotoxicity (ADCC), by which Dectin-1-expressing CD11b⁺ cells in blood were eliminated after i.p. administration of this antibody (200 $\mu\text{g}/\text{mouse}$) for 3 days (right FACS panels above). However, when we checked colonic cells, we found that this 3H8 had no effect on Dectin-1⁺ cell depletion, even at higher dose (500 $\mu\text{g}/\text{mouse}$) either by i.p. or i.v. (FACS panels above). We also tried another clone 2F3, but this antibody was less effective to suppress Dectin-1 signaling (qPCR panels below),

2F3

and when this antibody was administrated to mice, only Dectin-1⁺ myeloid cells in blood, but not those cells in the colon, were eliminated (FACS panels above).

We also examined the effect of these antibodies on Dectin-1 signaling *ex vivo*. After i.p. administration of one of these antibodies (300 $\mu\text{g}/\text{mouse}$) for 3 days, we harvested cells from mouse blood, and treated them with curdlan (500 $\mu\text{g}/\text{ml}$) for 20 h and then cytokine expression was determined by qPCR. As shown in the bar graphs below, 2F3 or 3H8 antibody

treatment did not inhibit curdlan-induced *Il1b* and *Il6* expression, suggesting that these antibodies mainly cause ADCC by binding to Dectin-1, but only weakly inhibit Dectin-1

signaling *in vivo*.

We also examined the ADCC-ability of one commercially available anti-Dectin-1 antibody (clone# R1-8g7, InvivoGen, cat# mabg-mduct), and found that *in vivo* administration of this antibody again only partially reduced Dectin-1-expressing cells in blood but not in the colon (FACS panels above). Thus, different from our expectation, all the Dectin-1 antibodies we examined could not suppress Dectin-1 signaling in mouse intestines and we gave up to use them for further *in vivo* experiments. In place of these antibodies, we used low MW laminarin (5% in food) to examine the effects of Dectin-1 inhibition on intestinal tumor development.

Minor comments:

- All flow cytometry assays are represented in % of cells. It is necessary to add the quantification of the number of cells for each analysis.

Thanks for the comment. Following this concern, we exhibited the absolute numbers of each immune cell subset shown in original (and new) Fig.2b-c, and found the differences in cell

numbers of each cell type between *ApcMin* and *ApcMin-Dectin-1* KO mice are consistent with the cell proportions originally shown (panels above). We added these data to new Fig. S2c in the revised manuscript.

Another experiment in which the flow cytometry was carried out is PGE₂ *in vivo* treatment in original Fig. 5g (and new Fig. 5c). Since we have changed the original CD11b⁺ cell % to CD11b⁺Ly6C⁺ and CD11b⁺Ly6C⁻ cell % according to the Reviewer 1's comment (new Fig. 5c and left panels above), we added absolute numbers of these two populations, as well as DCs and T cells (right panels above) to new Fig. S4h.

- The characterization of dendritic cells is weak. This characterization should be further investigated.

Following the reviewer's comment, we carried out the flow cytometry to check other markers for intestinal dendritic cells such as CD103, CX₃CR1 and Ly6C after AOM-3DSS treatment.

Meanwhile, we also carried out scRNA-seq analysis by using colonic tumor-infiltrating CD11b⁺ and CD11c⁺ myeloid cells. As shown in the panels above, CD11c⁺MHC-II⁺ DC

population was expanded in Dectin-1 KO colonic polyps (left panels), while the proportion of CD11b⁺Ly6C⁺ monocyte-derived DC subset in this population was not changed (right panels). Proportions of CD11b⁺CD103⁺, CD11b⁺CD103⁻ and CD11b⁻CD103⁺ subsets in whole DCs were also not changed in Dectin-1 KO mice compared with WT mice, in either polyps or non-polyp tissues (panels above). These data indicate that Dectin-1 deficiency facilitates the expansion of whole DC population but does not influence the proportion of DC subset(s).

By scRNA-seq analysis, we also confirmed that the whole DC population in Dectin-1 KO colonic polyps was increased, accompanied by the decrease of MDSCs (UMAPs and bar graphs below). We found that these polyp-infiltrating DCs are classified into subsets of

CD103⁺ cDC (highly expressing *Itgae*), monocyte-derived DC (highly expressing *Fcer1g*, *Csf1r*, *Cd14*), inflammatory DC-1 (highly expressing *Ccl5*, *Tnfrsf9*), inflammatory DC-2 (highly expressing *Ccl22*, *Il12b*), and tumor-associated (or -facilitating) DC (highly expressing *Pclaf*, *Spc24*, *Top2a* etc) (left heatmap above). All these DC subsets were similarly increased in Dectin-1 KO mice compared with WT mice (middle heatmap, UMAPs and bar graph above).

REVIEWER COMMENTS

Reviewer #1 (Remarks to the Author):

The authors addressed all of my questions completely and adequately.

Reviewer #2 (Remarks to the Author):

The authors have responded satisfactorily to many of my concerns and made a significant effort to improve this manuscript. No further comments.

Reviewer #3 (Remarks to the Author):

Tang et al. present a manuscript that inhibition of Dectin-1 signaling suppresses the colorectal tumors development in both ApcMin familial adenomatous polyposis model and AOM-DSS-induced CRC model. The authors begin by exploring the function of Dectin-1 on MDSCs. They found that Dectin-1 signaling promotes intestinal tumorigenesis by enhancing PGE2 synthesis in MDSCs. The authors then revealed that blocking Dectin-1 prevents colorectal tumorigenesis by enhancing IL-22BP expression. Moreover, they analyze the data on CRC patients and confirm that Dectin-1-PGE2-IL-22BP axis is also involved in the regulation of human colorectal carcinogenesis.

All told, I think the authors are tracking an interesting phenotype and have collected a fair amount of supporting data. However, I am a little apprehensive about the molecular mechanism and specificity in which Dectin-1 is involved in intestinal tumor development. Moreover, in several instances, more rigorous experimental validation would also bolster the interpretation and presentation of their findings. More detailed comments are included below.

Major comments:

1. The sc-RNAseq experiment is interesting, I would like to see the expression of Clec7a in different cell types by U-map at single cell level.
2. The authors found that infiltration of T cells (including $\alpha\beta$ & $\gamma\delta$ T cells), B cells (including IgG+ B cells), and DCs (MHC-II+CD11c+ cells) was significantly increased in ApcMin/+Clec7a-/- mice (Fig. 2b, S2b, S2c), suggesting enhanced immune responses in ApcMin/+Clec7a-/- mice compared with ApcMin/+ mice. In contrast, MHC-II-CD11c-CD11b+ MDSC population was greatly decreased in polyps of these Clec7a-/- mice (Fig. 2c, S2c), suggesting that Dectin-1 signaling facilitates MDSC expansion in intestinal tumors. However, as shown in Fig. 2b, the percentage of $\alpha\beta$ T cells, $\gamma\delta$ T cells and CD11c+ DC cells in total infiltrating cells were extremely low apart from CD19+ and IgG+ B cells. In Fig. 2c, only MHC-II-, CD11c-, CD11b+ were used to characterize MDSCs. To me, this result is not sufficient to conclude that Dectin-1 signaling facilitates MDSC expansion in intestinal tumors. The authors should better organize this part in their manuscript.
3. The bulk RNA-seq data in this study was performed by using RNAs from polyps and non-polyp tissues in the vicinity (pooled 4 WT and 5 Clec7a-/- GF mice). In Fig. 3, it is not clear how this data was analyzed. It seems as if only one replicate of each genotype was analyzed for the RNA-Seq. It would be better to present all the replicates in the figures (heatmap, bar plot, etc.) in order to obtain a more convinced result. The similar issue also happened in single-cell RNA-seq. Given the variabilities that can arise in a given genotype and for experimental rigor, two and ideally three distinct biological replicates of each genotype should be analyzed.
4. The authors indicate that Dectin-1 prevents colorectal tumorigenesis by enhancing IL-22BP expression, but it was already known that IL22-BP promotes intestinal tumorigenesis in mice. The molecular mechanism by which Dectin-1 specifically regulates IL-22BP expression is missing. Are there any target genes involved in intestinal tumor development which are directly influenced by Dectin-1? Similarly, How Dectin-1 regulate expression of PGE2 synthesizing enzymes?
5. Fig. 4a-c showed that decreased expression of Ptgs2 in colonic polyps and non-polyp tissues was also observed in Clec7a-/- mice under SPF conditions (Fig. 4a), and PGE2 concentration in colonic tissue homogenate was decreased in these mice (Fig. 4b). Decreased Ptgs2 expression in intestinal polyps was also observed in ApcMin/+Clec7a-/- mice (Fig. 4c). However, it looks that PGE2

concentration was not significantly decreased. How does this result support the authors' conclusion?

Minor comments:

Foxp3 plays an essential role in maintaining homeostasis of the immune system. In this study, the expression level of Foxp3 (Treg marker) in polyps and non-polyp tissue in Dectin-1-deficient mice was shown only by RT-PCR (Fig. S1r). An immunostaining is also essential to show the phenotypical difference.

Responses to Reviewers' comments (NCOMMS-22-19011A)

Reviewer #1 (Remarks to the Author):

The authors addressed all of my questions completely and adequately.

Thank you very much for your appreciation on our study. We greatly appreciate your kind and precious comments that led us to improve the revised manuscript.

Reviewer #2 (Remarks to the Author):

The authors have responded satisfactorily to many of my concerns and made a significant effort to improve this manuscript. No further comments.

Thank you so much for evaluating our finding and additional work in the present study. We greatly appreciate your kind and precious comments that led us to improve the revised manuscript.

Reviewer #3 (Remarks to the Author):

Tang et al. present a manuscript that inhibition of Dectin-1 signaling suppresses the colorectal tumors development in both ApcMin familial adenomatous polyposis model and AOM-DSS-induced CRC model. The authors begin by exploring the function of Dectin-1 on MDSCs. They found that Dectin-1 signaling promotes intestinal tumorigenesis by enhancing PGE2 synthesis in MDSCs. The authors then revealed that blocking Dectin-1 prevents colorectal tumorigenesis by enhancing IL-22BP expression. Moreover, they analyze the data on CRC patients and confirm that Dectin-1-PGE2-IL-22BP axis is also involved in the regulation of human colorectal carcinogenesis.

All told, I think the authors are tracking an interesting phenotype and have collected a fair amount of supporting data. However, I am a little apprehensive about the molecular mechanism and specificity in which Dectin-1 is involved in intestinal tumor development. Moreover, in several instances, more rigorous experimental validation would also bolster the interpretation and presentation of their findings. More detailed comments are included below.

Major comments:

1. The sc-RNAseq experiment is interesting, I would like to see the expression of *Clec7a* in different cell types by U-map at single cell level.

Thank you for your suggestion to examine the cell type-specificity of Dectin-1 expression using scRNA-seq. Dectin-1 is known to be expressed in myeloid-derived cells, and our flowcytometric analysis also showed that Dectin-1 is mainly expressed on CD11b⁺ cells, most of which are Ly6C⁺/Ly6G⁺MHC-II⁻ cells, in colorectal tumors (Fig. 2a & S2a). We

analyzed *Clec7a* expression in the scRNA-seq data shown in Fig. 3e according to your suggestion. As shown in the panels above, the *Clec7a* RNA was highly expressed in both monocyte clusters (cluster 0, 1, 2) including M-MDSCs and neutrophil clusters (cluster 3, 8, 12) including PMN-MDSCs, and was also expressed in macrophage clusters (cluster 5, 7, 10, 14) and DC clusters (cluster 4, 6, 9) at lower levels. These data are consistent with the data analyzed by FACS (Fig. 2a). The UMAP of WT mice is added to Fig. 4g.

2. The authors found that infiltration of T cells (including $\alpha\beta$ & $\gamma\delta$ T cells), B cells (including IgG⁺ B cells), and DCs (MHC-II⁺CD11c⁺ cells) was significantly increased in *ApcMin*⁺/*Clec7a*^{-/-} mice (Fig. 2b, S2b, S2c), suggesting enhanced immune responses in *ApcMin*⁺/*Clec7a*^{-/-} mice compared with *ApcMin*⁺ mice. In contrast, MHC-II⁻CD11c⁻CD11b⁺ MDSC population was greatly decreased in polyps of these *Clec7a*^{-/-} mice (Fig. 2c, S2c), suggesting that Dectin-1 signaling facilitates MDSC expansion in intestinal tumors. However, as shown in Fig. 2b, the percentage of $\alpha\beta$ T cells, $\gamma\delta$ T cells and CD11c⁺ DC cells in total infiltrating cells were extremely low apart from CD19⁺ and IgG⁺ B cells. In Fig. 2c, only MHC-II⁻, CD11c⁻, CD11b⁺ were used to characterize MDSCs. To me, this result is not sufficient to conclude that Dectin-1 signaling facilitates MDSC expansion in intestinal tumors. The authors should better organize this part in their manuscript.

Thank you for the comment. As pointed out by the reviewer, infiltration of T, B, and DC was also significantly changed in *Clec7a*^{-/-} mice. However, since we found that these T cells or B cells did not play important roles in the regulation of colorectal tumor development in our experimental setting (Rag2KO AOM-3DSS data in Fig. S3k-l), we did not further analyze their functions in tumorigenesis.

In Fig. 2c, we just used the markers MHC-II⁻CD11c⁻CD11b⁺ to identify MDSCs. But we further characterized these MDSC population using scRNA-seq analysis as shown in Fig. 3e, in which we showed that this population can be divided into M-MDSC (cluster 2) and PMN-MDSC (cluster 3). The data shown in Fig. 3e clearly show that these MDSC populations are decreased in *Clec7a*^{-/-} mice. We have also analyzed the relationship between neutrophils and PMN-MDSC and monocyte and M-MDSC by trajectory analysis (Fig. 3h).

At the same time, we further analyzed these MHC-II⁻CD11c⁻CD11b⁺ population by FACS by staining the cells with Ly6C. As shown in the panels below, intestinal tumor-infiltrating CD11c⁻CD11b⁺Ly6C⁺ cell population contains Ly6C^{hi} and Ly6C^{int} cell

subsets, which represents M-MDSC and PMN-MDSC, respectively. Both the proportions and the absolute cell numbers of these MDSC populations were decreased in *Clec7a*^{-/-} *Apc^{Min}* mice compared with *Apc^{Min}* mice. We added these results to new Fig. S2d with representative FACS panels, and additional statement in line 168-169 in the revised manuscript.

3. The bulk RNA-seq data in this study was performed by using RNAs from polyps and non-polyp tissues in the vicinity (pooled 4 WT and 5 *Clec7a*^{-/-} GF mice). In Fig. 3, it is not clear how this data was analyzed. It seems as if only one replicate of each genotype was analyzed for the RNA-Seq. It would be better to present all the replicates in the figures (heatmap, bar plot, etc.) in order to obtain a more convinced result. The similar issue also happened in single-cell RNA-seq. Given the variabilities that can arise in a given genotype and for experimental rigor, two and ideally three distinct biological replicates of each genotype should be analyzed.

Thanks for the comment and we agree with the reviewer's concern about replicates in RNA-seq analysis in Fig. 3a. Since we used pooled samples from 4 or 5 mice for an experiment, we believe that our data represent the average characteristics of the experimental group and thus exclude the possibility that we analyze an exceptional individual. Furthermore, we usually confirm these RNA-seq data by real-time qPCR or by flow cytometry, of which data were confirmed by several independent experiments. For example, the expression levels of *Ptgs2*, *Il22ra2* and *H2-Ab1* was confirmed by qPCR (panels below, data of *Il22ra2* expression is also shown in Fig. S5a). Also, some critical

gene expression such as IL-22BP (Fig. S5b, c) was confirmed at the protein level and PGE₂ synthesizing enzyme expression was confirmed by the end product PGE₂ levels. Furthermore, scRNA-seq analysis of AOM-DSS-induced polyps of our experiment shown in Fig. 3e-h was confirmed using public data (GSE196054) (Fig. S4a-c). Since we always checked consistency of the data among RNA-seq, scRNA-seq, qPCR, protein and FACS analysis, we are confident about the results in our present study.

4. The authors indicate that Dectin-1 prevents colorectal tumorigenesis by enhancing IL-22BP expression, but it was already known that IL22-BP promotes intestinal tumorigenesis in mice. The molecular mechanism by which Dectin-1 specifically regulates IL-22BP expression is missing. Are there any target genes involved in intestinal tumor development which are directly influenced by Dectin-1? Similarly, How Dectin-1 regulate expression of PGE2 synthesizing enzymes?

As the reviewer pointed out, suppression of intestinal tumorigenesis by IL-22BP is already known, but here we showed that IL-22BP production is inhibited by Dectin-1 signaling through induction of PGE₂. As shown in Fig. 4i, Dectin-1 signaling (stimulation

with Dectin-1 agonistic ligand curdlan *in vitro*) directly induces the PGE₂ synthesizing enzymes, suggesting these enzymes are the direct downstream of Dectin-1 signaling. Previously, we and others showed that Dectin-1 induces cytokines through activation of Syk-CARD9-NF-κB pathway (Saijo et al., *Immunity*, 2007; Drummond et al., *Eur J Immunol*, 2011). Furthermore, COX2 is induced by NF-κB p65 (Schmedtje et al., *J Biol Chem*, 1997; Duque et al., *Cell Signal*, 2006; Charalambous et al., *Br J Cancer*, 2009).

Thus, we examined the roles of the Syk-NF-κB pathway in the Dectin-1-induced PGE₂-synthesizing-enzyme expression. As shown in Fig. 4j (also shown in the panels above), *Ptg3*, *Ptg1* and *Ptg2* expression were suppressed by the inhibitors against NF-κB (IκBα: BAY11-7082) and SYK (R406), indicating that these PGE₂-synthesizing enzymes are directly induced by the activation of the Dectin-1-Syk-NF-κB pathway.

We have also analyzed the mechanism by which Dectin-1 regulates IL-22BP

expression since another reviewer also raised similar comment. We found that PGE₂ downregulates IL-22BP expression (Fig. 6c, 6d, S5e, S5f). Then, we further asked the mechanisms how PGE₂ regulates IL-22BP. IL-22BP expression is suppressed by IL-18 (Huber et al., Nature, 2012), while retinoic acid (RA) enhances the expression (Martin et al., Mucosal Immunol., 2014). We found that *Iil8* expression was upregulated by the stimulation with PGE₂ (left panel above). Furthermore, *Ii22ra2* expression was suppressed by PGE₂, and this suppression was recovered by the treatment with anti-IL-18 antibody, suggesting that PGE₂-induced IL-18 is responsible for the PGE₂-induced *Ii22ra2* suppression (right panel above and also in Fig. S5i, j).

Retinoic acid (RA) is also reported to enhance IL-22BP production (Martin et al., Mucosal Immunol, 2014; Lim et al., Sci Signal, 2016) and we found the expression of enzymes involving RA synthesis such as *Adh1* and *Aldh1a1* was enhanced in polyps in *Clec7a*^{-/-} mice (Fig. S5k, l). We also found that PGE₂ mainly suppressed the expression of *Aldh* family genes upon treatment of colonic myeloid cells with PGE₂ (middle panels below). When colonic and splenic myeloid cells were treated with RA, PGE₂-suppressed

Ii22ra2 expression was completely recovered (lower panels above). These observations suggest that suppression of RA is also involved in the regulation of *Ii22ra2* expression by PGE₂, at least in part (Fig. 6m-p, Fig. S5m-o).

5. Fig. 4a-c showed that decreased expression of *Ptgs2* in colonic polyps and non-polyp

tissues was also observed in *Clec7a*^{-/-} mice under SPF conditions (Fig. 4a), and PGE₂ concentration in colonic tissue homogenate was decreased in these mice (Fig. 4b). Decreased *Ptgs2* expression in intestinal polyps was also observed in *ApcMin*^{+/+}*Clec7a*^{-/-} mice (Fig. 4c). However, it looks that PGE₂ concentration was not significantly decreased. How does this result support the authors' conclusion?

By using another group of AOM-DSS-treated mice, we repeated the experiment described in Fig. 4b, and found the similar tendency that PGE₂ levels were decreased in *Clec7a*^{-/-} mice, although the difference was again not statistically significant (shown in the left panel below). But by pooling the data from two experiments, the difference of the PGE₂ levels in polyp between WT and KO mice becomes statistically significant (right panel below). We replaced the original Fig. 4b to this new panel in the revised manuscript.

What we want to show here is that PGE₂ levels are decreased in *Clec7a*^{-/-} mice and are correlated with the decrease of PGE₂ synthesizing enzymes.

Minor comments:

Foxp3 plays an essential role in maintaining homeostasis of the immune system. In this study, the expression level of Foxp3 (Treg marker) in polyps and non-polyp tissue in Dectin-1-deficient mice was shown only by RT-PCR (Fig. S1r). An immunostaining is also essential to show the phenotypical difference.

Because we have shown that adaptive immune response is not involved in the regulation of intestinal tumor development in the setting of our study (Fig. S3k-l, Rag2KO AOM-3DSS), we just showed the real-time RT-PCR data about Foxp3 expression in Fig. S1r.

Following the comment, we show here the Foxp3 staining by flowcytometry analysis, which exhibits the over-expansion of colonic Foxp3⁺CD4 T cells in Dectin-1 KO mice

under SPF condition (panels above), consistent with the observation in RT-PCR analysis. We added the histogram above to new Fig. S1t and added related statement in line 129-131 in the revised manuscript.

REVIEWERS' COMMENTS

Reviewer #3 (Remarks to the Author):

The author answered and responded to my questions and requests appropriately.

Responses to Reviewers' comments (NCOMMS-22-19011B)

Reviewer #3 (Remarks to the Author):

The author answered and responded to my questions and requests appropriately.

Thanks very much for your appreciation on our study. We greatly appreciate your precious comments that led us to improve our manuscript.